# Validation of the PALM model system 6.0 in a real urban environment; case study on Prague-Dejvice, Czech Republic

Jaroslav Resler[1], Kryštof Eben[1], Jan Geletič[1], Pavel Krč[1], Martin Rosecký[1], Matthias Sühring[2], Michal Belda[3], Vladimír Fuka[3], Tomáš Halenka[3], Peter Huszár[3], Jan Karlický[3], Nina Benešová[4], Jana Ďoubalová[3,4], Kateřina Honzáková[4], Josef Keder[4], Šárka Nápravníková[4], and Ondřej Vlček[4]

[1]Institute of Computer Science, Czech Academy of Sciences, Prague, Czech Republic
[2]Institute of Meteorology and Climatology, Leibniz University Hannover, Hannover, Germany
[3]Department of Atmospheric Physics, Faculty of Mathematics and Physics, Charles University, Prague, Czech Republic
[4]Czech Hydrometeorological Institute, Prague, Czech Republic

**Correspondence:** Jaroslav Resler (resler@cs.cas.cz)

**Abstract.** In recent years, The PALM 6.0 modelling system has been rapidly developing its capability to simulate physical processes within urban environments. Some examples in this regard are energy-balance solvers for building and land surfaces, a radiative transfer model to account for multiple reflections and shading, a plant-canopy model to consider the effects of plants on flow (thermo)dynamics, and a chemistry transport model to enable simulation of air quality. This study provides a thorough evaluation of modelled meteorological, air chemistry, and ground and wall-surface quantities against dedicated in-situ measurements taken in an urban environment in Prague, Dejvice, Czech Republic. Measurements included monitoring of air quality and meteorology in street canyons, surface temperature scanning with infrared cameras, and monitoring of wall heat fluxes. Large-eddy simulations (LES) using the PALM model driven by boundary conditions obtained from a mesoscale model were performed for multiple days within two summer and three winter episodes characterized by different atmospheric conditions.

For the simulated episodes, the resulting temperature, wind speed, and chemical compound concentrations within street canyons show realistic representation of the observed state, except that the LES did not adequately capture nighttime cooling near the surface for certain meteorological conditions. In some situations insufficient turbulent mixing was modelled, resulting in higher near-surface concentrations. At most of the evaluation points the simulated surface temperature reproduces the observed surface temperature reasonably well for both absolute and daily amplitude values. However, especially for the winter episodes and for modern buildings with multi-layer walls, the heat transfer through walls is partly not well captured, leading to discrepancies between the modelled and observed wall-surface temperature. Furthermore, the study corroborates model dependency on the accuracy of the input data. In particular, the temperatures of surfaces affected by nearby trees strongly depend on the spatial distribution of the leaf area density, land-surface temperatures at grass surfaces strongly depend on the initial soil moisture, wall-surface temperatures depend on the correct setting of wall material parameters, and concentrations on detailed information on spatial distribution of emissions, all of which are often unavailable at sufficient accuracy. The study also points out some current model limitations, particularly the implications of representing topography and complex heterogeneous facades on a discrete Cartesian grid, and glass facades that are not fully represented in terms of radiative processes.

Our findings are able to validate the representation of physical processes in PALM, while pointing out specific shortcomings. This will help to build a baseline for future developments of the model and improvements of simulations of physical processes in an urban environment.

## 1 Introduction

A majority of the world's population live in large cities (55 % as of 2018) and this percentage is expected to grow (UN, 2019). At the same time, global climate change, especially global temperature increases, will be influencing nearly every natural ecosystem and human society, with potentially severe impacts worldwide. The high level of attention currently being paid to the impact of climate change on urban areas is therefore amply justified, and is supported by many important studies and reports of global standing (IPCC, 2014a, b). This intensifying urbanization heightens the awareness that control of the microclimate in the urban environment, which can reduce heat stress among other general environmental improvements, is crucial for the well-being of city inhabitants (Mutani and Fiermonte, 2017). The problem of increased heat stress in urban areas as a consequence of what has become known as the urban heat island (UHI) is therefore of direct concern to municipal authorities, who are well aware that the physical well-being of their inhabitants is vital to the well-being of the whole city. Moreover, the UHI effect is often followed by secondary processes, such as air quality issues. Researchers have responded to, or anticipated, such concern and the requirement for modelling of urban climate processes, and several small-grid scale models and frameworks for numerical climate modelling have recently been developed (Geletič et al., 2018).

The health and well-being of the urban population is influenced by the conditions of the urban environment. The local microclimate, exposure to pollutants, and general human comfort depends strongly on the local conditions driven by the urban environment. The turbulent flow, exchange of latent and sensible heat, and radiative transfer processes play an important role in the urban microclimate and need to be considered in modelling approaches. The implementation of important microclimate processes (e.g. turbulence, heat fluxes and radiation) in street-level scale models is typically partially or fully parameterized. The most exhaustive approach consists of a group of computational fluid dynamics (CFD) models. The explicit simulation of turbulent flow is computationally demanding; thus, various techniques have to be adapted to make calculations feasible, usually based on limiting the range of the length scales and time scales of the turbulent flow to be resolved.

This study uses the PALM model system 6.0 (Maronga et al., 2020), which is an atmospheric modelling system. The core of the system contains model dynamics based on the LES (Large Eddy Simulation) and RANS (Reynolds-Averaged Navier-Stokes) techniques with additional modules for modelling of various atmospheric processes, e.g. interaction of atmosphere with earth surface or cloud microphysics. This system core is complemented with a rich set of *PALM-4U* (PALM for urban applications) modules related to modelling of physical phenomena relevant for urban climate, such as the interaction of solar radiation with urban surfaces and with urban vegetation, sensible and latent heat fluxes from the surfaces, storage of heat inside buildings and in pavements, or dispersion and chemical reaction of air pollutants (see Maronga et al., 2020). The first version of the PALM urban components represented the urban surface model (PALM-USM) which has been validated using data from a short experimental campaign in the centre of Prague (Resler et al., 2017). The new set of modules in PALM is

more general and is divided according to the physical processes they cover. The most relevant for urban climate are the land surface model (LSM), the building surface model (BSM), the radiative transfer model (RTM), the plant-canopy model (PCM), and the chemistry transport model (CHEM). The human biometeorology module (BIO) then allows evaluation of the impact of simulated climate conditions on the human population.

Validation of the urban model requires a dataset of measurements of the urban meteorological and air quality conditions, the properties of the urban canopy elements, and the energy exchange among parts of the urban canopy. Several campaigns of comprehensive observations and measurements of the urban atmospheric boundary layer, covering more than one season, have been done in the past. The Basel Urban Boundary Layer Experiment (BUBBLE) dataset containing observations from Basel is specifically targeted for validation of urban radiation models, urban energy balance models, and urban canopy parameterizations (Rotach et al., 2005). The MUSE experiment (Montreal Urban Snow Experiment) aimed at the thermoradiative exchanges and the effect of snow cover in the urban atmospheric boundary layer (Lemonsu et al., 2008). The CAPITOUL (Canopy and Aerosol Particles Interaction in TOulouse Urban Layer) project (Masson et al., 2008) concentrated on the role of aerosol particles in the urban layer.

Results of urban measurement campaigns have already been used for validation of several micrometeorological models, models of radiative transfer, and microscale chemical transport models. Micro-scale model validation brings difficulties due to high heterogeneity of the urban environment and the modelled variables, uncertainty in the detailed knowledge of urban canopy properties, as well as local irregularities caused by domain discretization. Important examples of such validation studies were published by Qu et al. (2013), Maggiotto et al. (2014) and Toparlar et al. (2015). These validation studies most frequently analyze micrometeorological models of the RANS type. Early examples of LES validation studies that include thermal conditions within cities were presented by Nozu et al. (2008) and Liu et al. (2012). Due to our previous experience with a limited validation of surface temperatures simulated by the PALM model (Resler et al., 2017), the aim of this study was to design a comprehensive experiment for complex model validation, including air velocity, air pollution, and surface temperature analysis. The focus on collection of detailed temporally and spatially localized observations in various urban canopy and meteorological conditions was dictated by the intention to use these observations to assess the performance of the newly developed or updated PALM modules RTM, BSM, LSM, PCM, and CHEM. This focus of the study also complied with its additional purpose, which was assessment of the utility of the PALM model performance for detailed urban studies (Geletič et al., 2021).

These considerations influenced the selection of the study area. The Prague-Dejvice quarter is an urbanized area typical of others in Prague and similar Central European cities with various types of urban environment. Further, the realization of the street level observation campaign was technically and organizationally easier in this area than areas such as the historical centre of Prague. Moreover, this area represents one of the pilot areas for urban adaptations studies carried out in cooperation with Prague Municipality and their organizations (e.g. Prague Institute of Urban Planning and Development). Their interest in the results of this study and their plans for subsequent modelling studies of urban heat island and air quality adaptation and mitigation strategies for this quarter also influenced our selection of this area.

Section 2 gives a detailed overview of the observation campaign, followed by a description and an evaluation of the numerical setup in Sect. 3 and 4. In Sect. 5 results from the numerical experiment and the observation campaign are presented

and compared. Finally, Sect. 6 closes with a summary, outlines the current limitations of the model, and gives ideas for future improvements.

## 2   Observation campaign

The observation campaign was designed with two main aims: 1) to evaluate PALM's capability, with its newly developed or improved thermal capability from the radiative transfer model (RTM), land and building surface modules (LSM, BSM), and plant canopy model (PCM), to reproduce surface temperatures; 2) to evaluate its capability to reproduce pollutant concentrations and meteorological quantities in different types of street canyons, with special focus on the impact of trees located in streets on both types of quantities. The campaign was carried out in a warm part of the year (10–23 July 2018, further referred to as summer campaign) and a cold part of the year (23 November–10 December 2018, further referred to as winter campaign). Measurement locations are shown in Fig. 1 and the measurements themselves are described in Sect. 2.3.1–2.3.5. More details on the campaign are available in ČHMÚ (2020).

### 2.1   Study area

The study area is located in the north-west centre of Prague, capital city of the Czech Republic. The position and a map of this area are presented in Fig. S1 in the supplements. This figure also marks the extent of the PALM modelling domains; for more information about model domain setup see Sect. 3.1. The study area includes complex terrain mainly in the western part of the outer domain (further referred as the parent domain), with altitude ranging from 175 to 346 meters above sea level. The altitude variability of the inner domain (further referred as the child domain) is up to 30 m (see Fig. S2). The observations were located inside the child domain (blue square in Fig. S2). This is a densely built-up area with specific conditions created by the roundabout (Vítězné náměstí) in combination with west-east (Evropská / Čs. armády) and north-south (Jugoslávských partyzánů / Svatovítská) oriented boulevards. The eastern and southern parts of the child domain represent a typical historical residential area in Prague-Dejvice, with a combination of old and new buildings and a variety of other urban components, such as gardens, parks and parking places. The north-west quarter has the larger buildings of the Czech Technical University campus. The south-western and north-eastern parts of the domain are more sparsely built-up by family houses. Local specific features include green intra-blocks with gardens and trees, usually with pervious ground surfaces; Prague historic centre usually has impervious intra-blocks. The building heights alongside the streets range approximately from 20 to 30 m, with the highest building in the domain being 60 m. Both boulevards are approximately 40 m wide and contain little green vegetation, except for Jugoslávských partyzánů Street which has some broadleaf trees about 20 m high. The majority of the trees are located in the intra-blocks and parks. The landcover map of the study area, based on Urban Atlas 2012 geodatabase, is shown in Fig. S3.

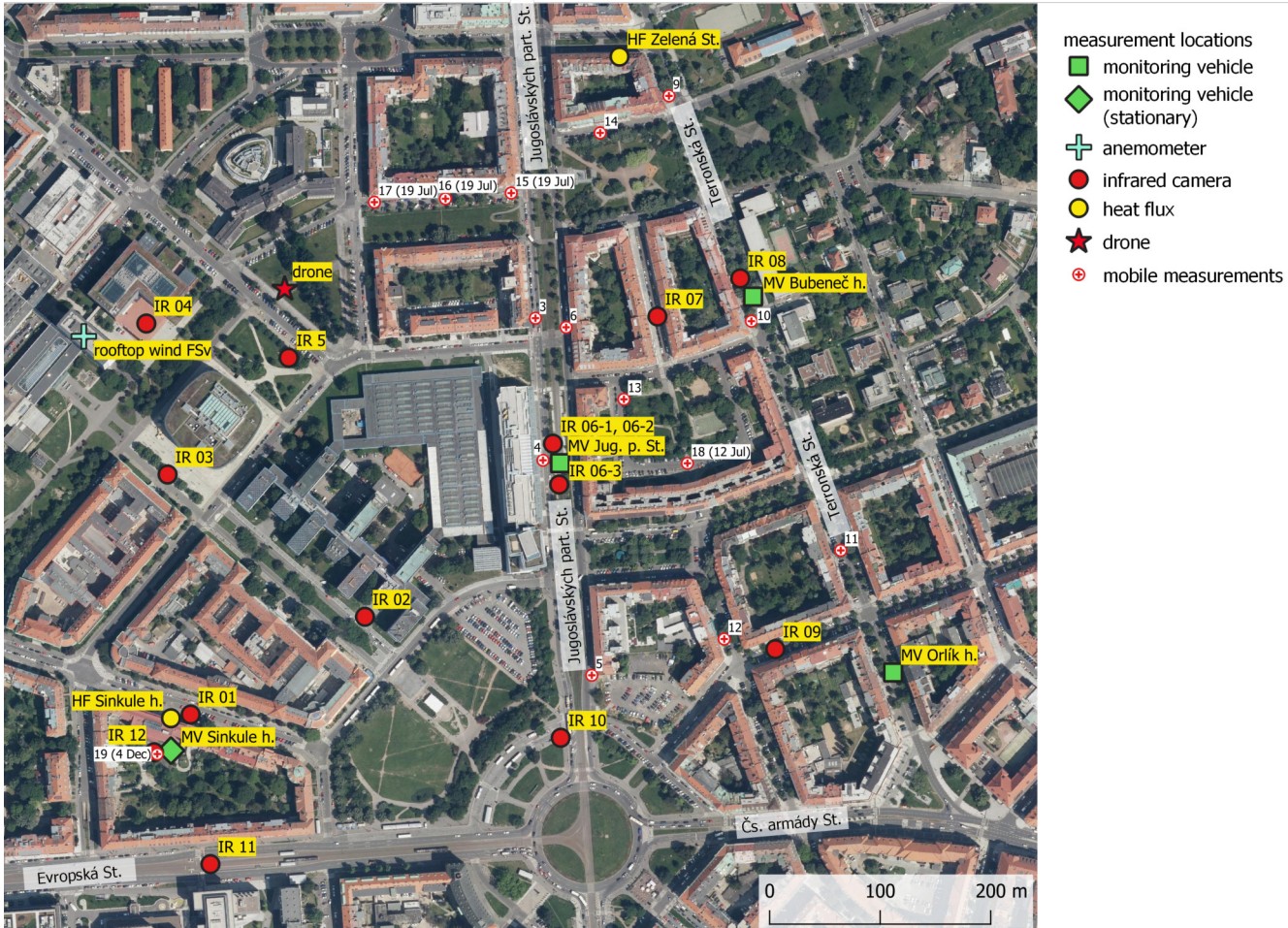

**Figure 1.** Map of measurement locations. Orthophoto was provided by WMS of the Czech Office for Surveying, Mapping and Cadastre (ČÚZK, 2020). For more information about point location (longitude, latitude etc.) see Table S1.

## 2.2 Validation episodes and synoptic situation

### 2.2.1 Summer campaign

The summer observation campaign ran for two weeks from 10 July 2018 to 23 July 2018 (see Table S2 in supplements), out of which two shorter episodes were selected for model simulations: 14–16 July (e1) and 19–23 July (e2). Synoptically, for most of the summer campaign the weather was influenced by a high pressure ridge over Central Europe between an Icelandic low and an Eastern European low-pressure system. Daily maximum temperature as measured at the Praha-Karlov (WMO ID 11519) station was below 30°C for the entire period, with the exception of 21 July when the maximum temperature reached 31.2°C. The beginning of the period was partially cloudy, mostly with altostratus clouds which formed in the morning and

early afternoon on 19 July. The period between the afternoon of 19 July and late afternoon on 21 July was mostly clear with cirrus clouds. The end of the campaign was cloudy, mostly with low-level cumulus. Important solar parameters mid-episode (19 July 2018) were: sunrise at 03:13 UTC, sunset at 19:02 UTC and solar noon at 11:08 UTC.

### 2.2.2 Winter campaign

The winter part of the observation campaign lasted from 24 November 2018 to 10 December 2018 (see Table S3 in supplements) and for the purposes of model validation, three episodes were selected: 24–26 November (e1), 27–29 November (e2), and 4–6 December (e3). Weather was influenced by a typical late autumn synoptical situation with westerly flow and low-pressure systems and a series of fronts separated by two anticyclonic situations (27–29 November and 5 December). During the campaign several occluded frontal passages were recorded in Prague: 24 and 30 November and 2, 3, 4 and 6 December, with rainfall on 30 November (4.3 mm at Praha-Ruzyně station; WMO ID 11518) and 2 and 3 December (9.8 mm and 3.6 mm at Praha-Ryzyně station). Average daily temperatures ranged from –4 °C on 29 November to 9 °C on 3 December. Average daily wind speed was around 3 m s$^{-1}$, except for 26 November when it reached 4.4 m s$^{-1}$ and 4–6 December with daily values of 4.8, 6.0 and 5.7 m s$^{-1}$. Important diurnal solar radiation parameters in Prague were (1 December 2018): sunrise at 6:39 UTC, sunset at 15:02 UTC, solar noon at 10:51 UTC.

## 2.3 Observed quantities and equipment used

### 2.3.1 Infrared camera measurements

Surface temperature measurements by an infrared (IR) camera were carried out during two days (45 hours total) of the summer and 3 days (50 hours total) of the winter campaigns (see Table S2 and Table S3). Measurements were taken at twelve locations shown in Fig. 1 approximately every 60–80 minutes. At each location, several directions were chosen and usually two snapshots capturing horizontal (ground) and vertical (wall) surfaces were taken in each direction. We use the following nomenclature further in the text: <location_number>-<direction_number>_H/V. For example 02-1_H means image of the ground taken from the second location in the first direction. In every image, a few evaluation points labelled by numbers were chosen and temperature time series extracted. The particular point at which modelled and observed values are compared is then referred to e.g. as 02-1_H3. The observation campaign in total gathered time series of surface temperature for 66 ground and 73 wall evaluation points, representing various surface types, in order to evaluate model performance under different surface parameter settings such as different surface materials and conditions.

Temperature was measured by the FLIR SC660 (FLIR, 2008) - the same camera used in Resler et al. (2017). As in this article, the camera's thermal sensor field of view is 24 by 18° and spatial resolution (given as an instantaneous field of view) is 0.65 mrad. The spectral range of the camera is 7.5 to 13.0 $\mu$m, and the declared thermal sensitivity at 30°C is 45 mK. The measurement accuracy for an object with a temperature between 5 and 120°C given an ambient air temperature between 9 and 35°C is ±1°C, or ±1 % of the reading. The camera offers a built-in emissivity-correction option, which was not used for this study. Apart from the infrared pictures, the camera allowed us to simultaneously take pictures in the visible spectrum.

Where possible, pictures were processed semi-automatically as described in Resler et al. (2017). This processing requires the presence of 4 well-defined points in each picture, which are used to correct for changes in camera positioning between measurements as the camera was rotated around locations. Pictures which did not allow for semi-automatic processing (mostly ground images) were handled manually and temperatures were extracted by the FLIR Tools v5.13.18031.2002 software (https://www.flir.com/products/flir-tools/). Examples of semi-automatic and manually processed images are shown in Fig. S4.

Surface temperature measured by the FLIR SC660 was compared with the data from heat flux measurements at Sinkule house captured by the heat flux measuring system TRSYS01 (see Sect. 2.3.2). The results are shown in Fig. S5. The IR camera generally gives higher values than the TRSYS01 system (instantaneous measurements are compared with 10-min averages): in summer ground floor temperatures are on average 1°C higher (difference range 0.0–2.8°C) and 1st floor on average 0.1°C higher (range of differences between –2.0 and +1.3°C). In winter the ground floor temperatures are on average 2.1°C higher (difference range 0.5–3.5°C) and 1st floor on average 1°C higher (range of differences between –0.6 and +2.0°C).

### 2.3.2 Wall heat fluxes measurement

Heat fluxes through the building facade and windows were measured by the high-accuracy building thermal resistance measuring system TRSYS01 equipped with two HFP01 heat flux plates and two pairs of thermocouples (TC). The operating temperature range of HFP01 and TC is –30 to +70°C. The declared sensitivity of temperature difference measurements between inner and outer sides of the wall is 0.02°C and heat flux measurement resolution 0.02 W m$^{-2}$. The calibration uncertainty of HFP01 is ±3 % (Hukseflux, 2020). Heat fluxes were measured through the north-east-facing wall of Sinkule house and through the north-facing wall and window of the building in Zelená Street (Fig. 2). The position of the sensors on both buildings is shown in Fig. S6. Silicone glue was used to attach the sensors to the outside wall on the 1st floor of Sinkule house during the winter campaign. Otherwise sensors were mounted by a two-sided carpet tape.

Sinkule house was built before World War II with walls made of construction blocks. The ground floor wall is 34 cm thick without insulation, and the facade is made of ceramic tiles. The wall of the 1st floor is 41 cm thick including 6 cm thick polystyrene insulation on the outer side. The facade surface is scratched plaster with scratches of 1–2 mm depth (see Fig. 2).

The house in Zelená Street is a typical representative of buildings in the area, with walls made of construction blocks. Wall thickness at the place of measurement was approx. 30 cm with 2.5-cm lime-cement plaster on the inner and outer sides of the wall. Heat flux measurement through the window was not used in PALM validation and therefore is not described here.

A quality check measurement was done at the beginning of the summer campaign – sensors were placed side-by-side in the 1st floor of Sinkule house between 19 July 17:40 CEST and 20 July 12:00 CEST. The absolute difference of the facade surface temperature was 0.0–1.5°C with a median value of 0.1°C. The absolute difference of measured heat fluxes was 0.0–2.1 W m$^{-2}$ with a median value of 0.6 W m$^{-2}$.

### 2.3.3 Vehicle observations

Air quality and meteorological measurements in the street canyons were obtained by two monitoring vehicles, which were shuttled periodically among the three locations marked as green squares in Fig. 1. One location was in Jugoslávských partyzánů

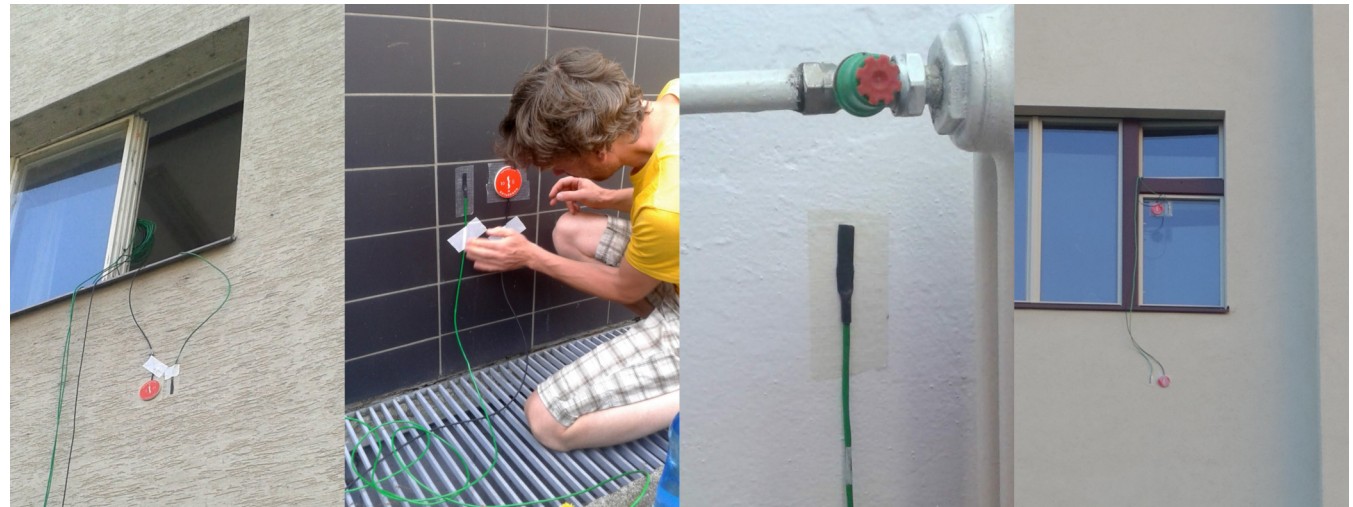

**Figure 2.** Detail of heat flux sensor and thermocouple mounting. Left - Sinkule house $1^{st}$ floor; center-left - Sinkule house ground floor; center-right - Sinkule ground floor - inner temperature sensor; right - Zelená Street. For Sinkule house and Zelená Street locations see Fig. 1.

Street (Jug. p. St.), an approx. 42-m wide boulevard with sparse trees. The two remaining locations were in the 25-m wide Terronská Street, one next to Bubeneč house and the other next to Orlík house. Near to Bubeneč house there are full-grown broadleaf trees with crowns covering the whole street. Broadleaf trees near to Orlík house are smaller and their crowns cover a maximum of two thirds of the street canyon. Buildings in all locations are approx. 25 m high. Pictures of the measurement locations are shown in Fig. S7. The observations were organised so as to provide information about air quality and meteorological conditions in the three locations and also to compare the east and west sides of the street canyons. Each monitoring vehicle remained at a particular location for at least two whole days (see Table S2 and Table S3). Based on our own traffic census from 4–6 December 2018, the total workday load in Terronská St. past Bubeneč house is 7,700 vehicles, which is approximately 44 % of the traffic intensities in Jug. p. St. The number of small trucks (60) in Terronská St. is only 20 % of that in Jug. p. St. and the number of buses (20) is only 2 % of the number in Jug. p. St. There was only one large truck per day noted in Terronská St., compared to approx. 80 in Jug. p. St. Apart from the street canyon measurements, one stationary monitoring vehicle was located in the courtyard of Sinkule house throughout the whole campaign to provide the urban background meteorological and air quality values.

The vehicles in the street canyons were equipped with analyzers of $NO_X$, $NO_2$, $NO$, $O_3$, $SO_2$, $CO$, $PM_{10}$, $PM_{2.5}$, and $PM_1$ measured at the top of the vehicle roof (approx. 4.6 m). Calibrations of all air quality analyzers were performed during transfer between locations to eliminate loss of data during parallel measurements. Meteorological variables measured included wind speed and direction and turbulent flow characteristics measured by the METEK 3D ultrasonic anemometer on a meteorological mast at a height of about 6.8 m above the ground (to fit under the tree crowns in Terronská St. next to Bubeneč house), air temperature, relative humidity, global radiation, and atmospheric pressure. Wind and turbulent flow characteristics measured by the METEK anemometer had a 10 minute resolution, with the remaining variables recorded by the instruments at 1-minute

resolution. For further analysis and PALM evaluation, 10-minute averages of measured variables were used. Both vehicles also had a video camera placed at the front windscreen. These recordings were then used for detailed time disaggregation of traffic

emissions at the measurement location and for calibration of an automatic counting system (see Sect. 3.4).

The vehicle in Sinkule house courtyard measured the same variables with the same time resolution except for the following differences: $PM_1$, $PM_{2.5}$ and turbulence characteristics were not measured; wind speed and direction were measured by the GILL 2D WindSonic anemometer at the standard height of 10 m.

### 2.3.4 Mobile measurements

On selected days of the measurement campaigns, to get more detailed information on air quality in the child domain, mobile measurements using a dedicated monitoring vehicle were made (12, 18, 19 July, 26 November, and 4 December). This vehicle travelled between the locations shown on Fig. 1, stopping and measuring at each of them for five minutes. Two loops were made on every measurement day. On 19 July only one loop among locations 3, 6, and 15–17 was made, but with measurements taken over 15–20 min. The vehicle was equipped with $NO_X$, $NO_2$, $NO$, $O_3$, $SO_2$, $CO$, $PM_{10}$, $PM_{2.5}$, and $PM_1$ analyzers. Starting from

the second measurement on 17 July, a Garni 835 weather station was used for an indicative measurement of temperature, wind and relative humidity. Some measurements were not available on particular days – details are given in Table S2 and Table S3.

### 2.3.5 Higher level observations

To get information about higher levels, the observation campaign used two other measurement platforms. The first was a stationary measurement of wind flow on the top of the highest building in the child domain (approx. 60 m high). A 2D

anemometer was installed on the flat roof of the Faculty of Civil Engineering of the Czech Technical University – (FSv; see Fig. 1). The anemometer was positioned approximately in the middle of the highest roof section, 2 m above the flat roof top. The location was the same in summer and winter campaigns. Measurement frequency was 1 second and 10-minute averages were used for further evaluation. The second was a measurement of vertical profiles in the lowest part of the atmosphere by drone. Originally, two one-day drone observation campaigns were scheduled. Due to administrative restrictions, the summer

drone observations were not realised and the winter ones had to be moved from the centre of the child domain to the location marked in Fig. 1. Additionally, the maximum flight altitude had to be limited to 80 m above ground. The drone was equipped with the GRIMM Portable Laser Aerosol spectrometer and Dust Monitor Model 1.108 and a HC2A-S probe from ROTRONIC for temperature and relative humidity measurements (ROTRONIC, 2020). Unfortunately, the probe showed a longer than expected relaxation time which meant the observation instruments were not able to stabilize quickly enough during the descent.

Recalculation of particle counts to mass concentration was also burdened with large errors. The results obtained were not reliable enough to be used for PALM validation, but temperature and relative humidity profiles are provided in supplements (Fig. S8 and Fig. S9).

### 2.3.6 Standard CHMI observations used for validation

Relevant standard meteorological and air quality measurements were used for the evaluation of WRF and CAMx simulations which provided initial and boundary conditions for PALM as described in Sect. 3.3. This evaluation is presented in Sect. 4. WRF vertical profiles were evaluated against the upper air soundings from Praha-Libuš (WMO ID 11520) station located in a southern suburb of Prague, 11 km from from the center of the PALM child domain. A radiosonde is released every day at 0, 6, and 12 UTC. For evaluation of global radiation, two meteorological stations were selected: Praha-Libuš, and the Praha-Karlov (WMO ID 11519) station situated in a densely built-up area nearer the center of Prague approximately 4 km from the PALM child domain. $PM_{10}$ and $NO_X$ concentrations from the CAMx model were compared with measurements from automated air quality monitoring stations. Only the 5 background stations closest to the PALM child domain were used. Station locations are shown in Fig. S10. More detailed information about the stations is given in Table S4 and Table S5.

Observations from the Praha-Ruzyně station (WMO ID 11518) situated at Prague airport approximately 9 km west from the center of PALM domain were used to evaluate WRF wind speed and, in conjunction with the campaign wind measurements on the FSv building roof, the modification of wind speed by the orography and buildings and how PALM captures this effect.

## 3 Model simulation setup

### 3.1 PALM model and domains configuration

The PALM model system version 6.0 revision 4508 (Maronga et al., 2015, 2020) was utilized for this validation study. It consists of the PALM model core and components which have been specifically developed for modelling urban environments. The PALM model core solves the incompressible, filtered, Boussinesq-approximated Navier-Stokes equations for wind (u, v, w) and scalar quantities (potential temperature, water vapor mixing ratio, passive scalar) on a staggered Cartesian grid. The sub-grid scale terms that arise from filtering are parametrized using a 1.5-closure by Deardorff (1980) with modifications after Moeng and Wyngaard (1988) and Saiki et al. (2000). Buildings and orography are mapped onto the Cartesian grid using the mask method (Briscolini and Santangelo, 1989), where a grid cell is either 100% fluid or 100% obstacle. The advection terms are discretized by a fifth-order scheme after Wicker and Skamarock (2002). For temporal discretization, a third-order low-storage Runge-Kutta scheme (Williamson, 1980) is applied. The Poisson equation is solved by using a multigrid scheme (Maronga et al., 2015).

The following are the urban canopy related PALM modules employed in this study. The land surface model (LSM, Gehrke et al., 2020) was utilized to solve the energy balance over pavements, water, and low-vegetated surfaces. The building surface model (BSM, in previous versions and in Resler et al., 2017 called USM) was used to solve the energy balance of building surfaces (walls and roofs). The BSM was configured to utilize an integrated support for modelling of fractional surfaces (Maronga et al., 2020). Dynamic and thermodynamic processes caused by resolved trees and shrubs were managed by the embedded plant-canopy model (PCM). Radiation interaction between resolved scale vegetation, land-surface, and building surfaces was modelled via the radiative transfer model (RTM, Krč et al., 2020). Downwelling shortwave (SW) and longwave

(LW) radiation from the upper parts of the atmosphere, which were used as boundary conditions for the RTM, were explicitly prescribed from the stand-alone Weather Research and Forecasting model (WRF; see Sect. 3.3 for details) simulation output for the respective days, rather than being modelled by e.g. the Rapid Radiation Transfer Model for Global Models (RRTMG). This way, effects of mid- and high-altitude clouds on the radiation balance were considered in the simulations. It is important to note that by not using RRTMG some physical processes were missed, such as vertical divergence of radiation fluxes leading to heating / cooling of the air column itself; these may become especially important at nighttime. However, sensitivity tests with RRTMG applied revealed that the effect on nighttime air temperature was negligible in our simulations. In addition to the meteorological quantities, the embedded online chemistry model (Khan et al., 2020) was applied to model concentrations of $NO_X$, $PM_{10}$, and $PM_{2.5}$. Chemical reactions were omitted in this case to simulate purely passive transport of the pollutants.

Both self and online nesting features of PALM were utilised. Self-nesting means that a domain with a finer resolution can be defined inside a larger domain and this subdomain (child domain) receives its boundary conditions from the coarse-resolution parent domain at every model timestep. Here, a one-way nesting without any feedback of the child simulation on the parent simulation (Hellsten et al., 2020) was applied. The coarse-resolution parent simulation itself received its initial as well as lateral and top boundary conditions from the simulations of the mesoscale model WRF transformed to a PALM dynamic driver (see Sect. 3.3). This process is hereafter referred to as mesoscale nesting (Kadasch et al., 2020). The values of the velocity components, potential temperature, and values for the mixing ratio at the lateral and top boundary were updated at every model time step, while linear interpolation in time was used to interpolate between two WRF timesteps. The WRF solution was mapped fully onto the boundaries starting at the first grid point above the surface; boundary grid points that lie below the surface were masked and were not considered further. As the mesoscale model does not resolve turbulence, turbulence was triggered at the model boundaries using an embedded synthetic turbulence generator (STG) according to Xie and Castro (2008), which imposed spatially and temporally correlated perturbations every time-step onto the velocity components at the lateral boundaries. For additional details on PALM's mesoscale nesting approach we refer to Kadasch et al., 2020.

The initial and boundary concentrations of modelled pollutants of the parent domain were taken from simulations of the CAMx model (Comprehensive Air-quality Model with Extensions; see Sect. 3.3). For more detailed information about the PALM model, embedded modules, and the PALM-4U components see Maronga et al. (2020) and the companion papers in this special issue.

The location of the parent and child modelling domains is shown in Fig. S1. The parent domain extends horizontally by $4 \times 4 \text{ km}^2$ in the $x$- and $y$-direction, respectively, with an isotropic grid spacing of 10 m. The vertical $z$-direction is covered by 162 layers for summer and 82 layers for winter simulations, respectively. The vertical grid spacing is 10 m for the lower 250 m of the domain. Above 250 m, when the height was well above the building-affected layer, the vertical grid was successively stretched up to a maximum vertical grid spacing of 20 m in order to save computational resources. The domain top is at 2,930 m for summer and 1,330 m for winter simulations, respectively. This extent safely covers the convective layer with a sufficient buffer. We note that the 10 m resolution of the parent domain is sufficient to explicitly resolve the majority of the buildings and trees (see Fig. S11 and Fig. S12 in supplements) which means that no additional parameterization of the urban canopy is

needed. The child domain extent is 1,440 × 1,440 × 242 m$^3$ in the x-, y-, and z-directions respectively, with an isotropic grid
spacing of 2 m.

Parent and child domains were initialized by vertical profiles of u, v, w, potential temperature and mixing ratio, and soil
moisture and soil temperature, transformed from WRF simulations (see Sect. 3.3). Since the initial soil and wall temperatures
from a mesoscale model are only a rough estimate due to its aggregated nature, the PALM spin-up mechanism was applied
(Maronga et al., 2020). During a 2-day spinup, the atmospheric code was switched-off and only the LSM and BSM together
with the radiation and RTM model were executed. By this method, the material temperatures were already close to their
equilibrium value and significant changes in material temperatures at the beginning of the simulation were avoided.

## 3.2 Urban canopy properties

Data availability, their harmonization, and cost/efficiency trade-offs often need to be considered (Masson et al., 2020). For
solving the energy balance equations as well as for radiation interactions, BSM, LSM, and RTM require the use of detailed
and precise input parameters describing the surface materials such as albedo, emissivity, roughness length, thermal conduc-
tivity, thermal capacity, and capacity and thermal conductivity of the skin layer. Also the plant canopy (trees and shrubs) is
important as it affects the flow dynamics, heating, and evapotranspiration as well as radiative transfer within the urban envi-
ronment. Urban and land surfaces and sub-surface materials become very heterogeneous in a real urban environment when
going to very fine spatial resolution. Any bulk parameterization for the whole domain setting would therefore be inadequate.
Instead, a detailed setting of these parameters was supplied wherever possible. To obtain the needed detailed data, a supple-
mental on-site data collection campaign was carried out and a detailed database of geospatial data was created. Land-cover
data are based on a combination of national (ZABAGED) and city of Prague (Prague OpenData) databases. ZABAGED geo-
database (ČÚZK, 2020) distinguishes 128 categories of well-targeted geographical objects and fields, for example built-up
areas, communications, hydrology, vegetation, and surface. The Prague OpenData geodatabase (Prague Geoportal, 2020) dis-
tinguishes many local, user-specified GIS layers, e.g. plans showing actual and future development, land-cover for architects, a
photogrammetry-based digital elevation model (DEM) etc. Building heights were available from the Prague 3D model, main-
tained by the Prague Institute of Planning and Development. For the first tree canopy data mapping, LiDAR scanning was used
in combination with a photogrammetric-based DEM. Derived heights were manually calibrated using data from the terrain
mapping campaign and extended with additional parameters like crown height, width and shape, and trunk height and width.
All descriptions of surfaces and materials and their properties were collected in GIS formats and then preprocessed into a
PALM NetCDF input file corresponding to the PALM Input Data Standard (PIDS; Heldens et al., 2020). This file includes
information on wall, ground, and roof materials and properties similar to those used to estimate surface and material properties
in Resler et al. (2017) and Belda et al. (2020).

Each surface is described by material category, albedo, and emissivity. BSM surfaces additionally carry thickness and win-
dow fraction. Parameters such as thermal conductivity and capacity are assigned to categories and estimated based on surface
and storage material composition. In the case of walls and roofs, which are limited to four layers in the current version of
BSM, this means the parameters of the two outer layers were assigned according to the properties of the covering material (e.g.

plaster or insulation), while remaining layers were initialized by the properties of the wall material (e.g. bricks, construction blocks, concrete, insulation). Wall and roof properties are described in Table S6. For pavements and other LSM surfaces, all

parameters except albedo and emissivity were assigned according to the PALM LSM categories.

Each tree in the child domain was detailed by its position, diameter, trunk parameters, and vertically stratified base leaf area density. The actual distribution of the leaf area density (LAD) within the treetop was then calculated according to the available light exposure of the particular gridbox inside the treetop according to the Beer-Lambert law, leading to lower LAD in the centres of large and/or dense treetops. At the moment PALM does not consider the effect of trunks on the dynamic flow field

and the thermodynamics; only LAD is considered. However, for the winter case leafless deciduous trees were considered to be 10 % of their summer LAD to account for the effect of trunks and branches on the flow field.

### 3.3 Initial and boundary conditions

Initial and boundary meteorological conditions for the parent domain of the PALM simulations were obtained from the WRF model (Skamarock et al., 2008), version 4.0.3. The WRF model was run on three nested domains, with horizontal resolutions

of 9 km, 3 km and 1 km and 49 vertical levels. The child domain has $84 \times 84$ grid points in the horizontal. The choice of configuration started from the most usual settings for the given resolution and required latitude. Then minor variations in parameterizations were tested so as to provide the best possible boundary conditions to PALM for each simulation. Consequently the NOAH LSM (Chen and Dudhia, 2001) and RRTMG radiation (Iacono et al., 2008) have been used in all simulations. Urban vs. non-urban parametrizations for PBL were tested and, as a result, the Yonsei University PBL scheme (Hong et al.,

2006) was chosen for the summer episodes while for the winter episodes the Boulac urban PBL (Bougeault and Lacarrère, 1989) gave a better agreement with observations. With this exception, no other urban parameterizations have been used in the WRF model. MODIS land use categories have not been altered. WRF was initialized from the GFS operational analyses and forecasts and output data from overlapping WRF 12 hour runs was collected. The first six hours of each run served as a spin-up. The boundary conditions for the mesoscale nesting were then generated from forecast horizons 7–12.

Air quality simulations that served as chemical initial and boundary conditions were made using the chemistry transport model (CTM) CAMx version 6.50 (ENVIRON, 2018). CAMx is an Eulerian photochemical CTM that contains multiple gas phase chemistry options (CB5, CB6, SAPRC07TC). Here, the CB5 scheme (Yarwood et al., 2005) was invoked. Particle matter was treated using a static two-mode approach. Dry deposition was calculated following Zhang et al. (2003) and for wet deposition, the Seinfeld and Pandis (1998) method was used. To calculate the composition and phase state of the ammonia-

sulfate-nitrate-chloride-sodium-water inorganic aerosol system in equilibrium with gas phase precursors, the ISORROPIA thermodynamic equilibrium model was used (Nenes et al., 1998). Finally, secondary organic aerosol (SOA) chemistry was solved using the semi-volatile equilibrium scheme SOAP (Strader et al., 1999).

CAMx was coupled offline to WRF, meaning that CAMx ran on WRF meteorological outputs. WRF outputs were translated to CAMx input fields using the WRFCAMx preprocessor provided along with the CAMx source code (see http://www.camx.

com/download/support-software.aspx). For those CAMx input variables that were not available directly in WRF output, diagnostic methods were applied. One of the most important inputs for CAMx, which drives the vertical transport of pollutants, is

the coefficient of vertical turbulent diffusion (Kv). Kv is a significant parameter that determines the city scale air pollution and it is substantially perturbed by the urban canopy effects (Huszar et al., 2018a, b, 2020a, b). Here, the "CMAQ" scheme (Byun, 1999) was applied for Kv calculations.

WRF and CAMx outputs were then postprocessed into the PALM dynamic and chemistry driver. The data were transformed between coordinate systems and a horizontal and vertical interpolation was applied. As the coarse-resolution model terrain would not match the PALM model terrain exactly, the vertical interpolation method included terrain matching and the atmospheric column above the terrain was gradually stretched following the WRF hybrid vertical levels as they were converted to the fixed vertical coordinates of the PALM model. The interpolated airflow was adjusted to enforce mass conservation. Detailed

technical description of the 3D data conversion procedure is given in the supplements in Sect. S6. The Python code used for processing the WRF and CAMx data into the PALM dynamic driver file has been included in the official PALM distribution and published in the PALM SVN repository since revision 4766 in the directory trunk/UTIL/WRF_interface.

    Emission data for Prague used in the CAMx model were as described in the following chapter. Other emission inputs are described in detail in Ďoubalová et al. (2020).

## 3.4    Emission data

Air pollution sources for our particular case are dominated by the local road traffic. Annual emissions totals were based on the traffic census 2016 conducted by the Technical Administration of Roads of the City of Prague – Department of Transportation Engineering (TSK-ÚDI). The emissions themselves were prepared by ATEM (Studio of ecological models; http://www.atem.cz) using the MEFA 13 model. Jugoslávských partyzánů and Terronská Streets, where air quality was mea-

sured during the campaigns, were both covered by this census. Emissions from streets not included in the census were available on a grid with 500m spatial resolution. These emissions were distributed between the streets not covered by the census according to their parameters. Particulate matter (PM) emissions included resuspension of dust from the road surface (Fig. 3). Time disaggregation was calculated using a Prague transportation yearbook (TSK-ÚDI, 2018), public bus timetables, and our own short-time census (19–21 July and 4–6 December; days in which traffic intensities were derived from camera records). This

time disaggregation was the same for the primary emissions (exhaust, brake wear etc.) as well as for resuspended dust. Higher dust resuspension caused by sprinkle material during winter time was not considered.

    Traffic data were supplemented by emissions from stationary sources from the Czech national inventory REZZO. Point sources correspond to the year 2017, the latest year available at the time of model input preparation. Residential heating was based on 2017 inventory and rescaled to 2018 by multiplying by the ratio of degree days DD(2018)/DD(2017); DD is the

sum of the differences between the reference indoor temperature and the average daily outdoor temperature on heating days. Residential heating emissions were available on elemental dwelling units - urban areas with average area 0.5 $km^2$ - and were spatially distributed to building addresses, where local heating sources are registered, in proportion to the number of flats. Time disaggregation of point source emissions was based on monthly, day-of-week, and hour-of-day factors (Builtjes et al., 2003; available also in Denier van der Gon et al., 2011). Residential heating emissions were allocated to days according to

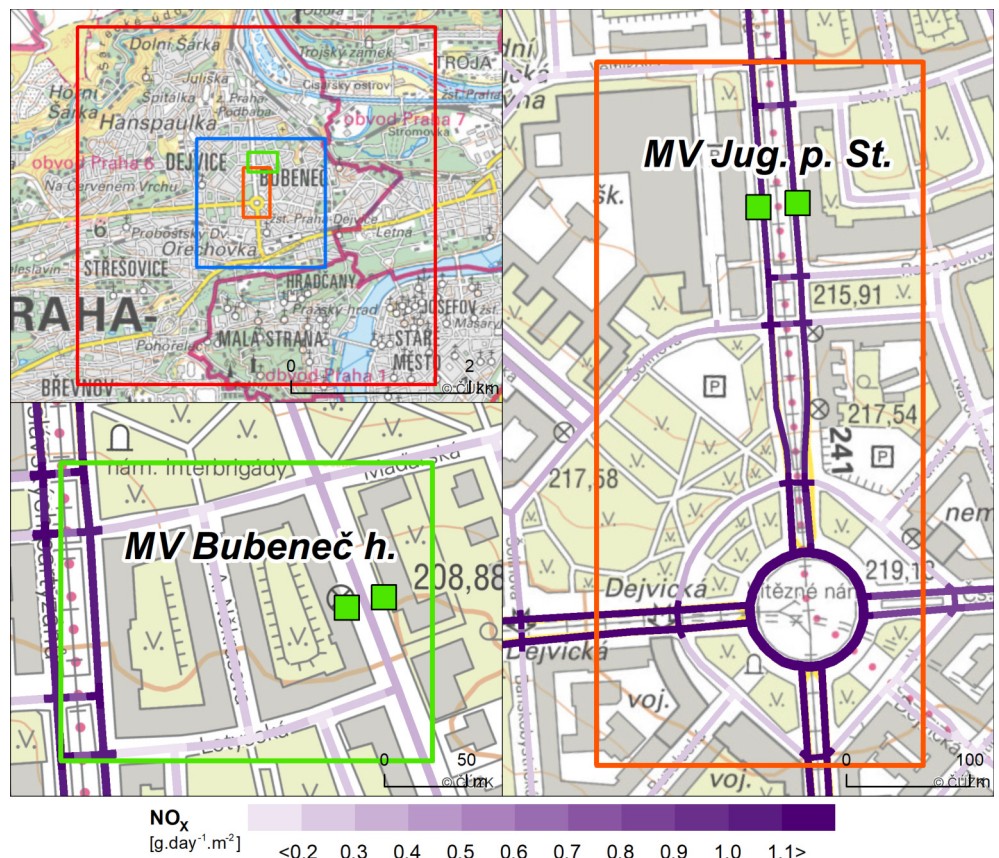

**Figure 3.** Nitrogen oxides ($NO_X$) emitted by cars along their trajectories in selected locations in Prague-Dejvice. Emissions were summarized in $\mathrm{g\,day^{-1}\,m^{-2}}$ and disaggregated to 1-hour time steps. The red and blue squares in the top left map indicate the extent of the parent and child PALM domain, respectively. The orange and green rectangles show the location of the expanded views at right and lower left. The expanded views show as green squares the air-quality measurement locations (MV) in Terronská St., Bubeneč house (lower left) and Jugoslávských partyzánů St. (right). The base map of the Czech Republic at 1:10,000 for the city of Prague was provided by Czech Office for Surveying, Mapping and Cadastre (ČÚZK, 2020).

the standardized load profile of natural gas supply for the households, which use it for heating only (Novák et al., 2019; OTE, 2020). Daily variation of residential heating emissions was taken from Builtjes et al. (2003).

All these input emission data were processed into PALM input NetCDF files corresponding to the PALM Input Data Standard (PIDS).

### 3.5 Observation operator

To compare modelled and observed values, an observation operator which links model variables to observed quantities is needed. For vehicle measurements, the situation was straightforward; horizontally, we used atmospheric quantities and chem-

ical compounds at the grid cell closest to the real placement of the sensors while vertically, we performed linear interpolation to the real height of the sensor. This approach was sufficient given the fine 2m resolution within the child domain. For surface observations at grid-aligned surfaces (wall sections without significant influence of step-like structures), the modelled values
at the nearest grid face according to the actual placement of the sensor or evaluation point were also taken. However, at non grid-aligned walls, i.e. walls which are oriented in one of the south-west, south-east, north-west, and north-east directions, walls are approximated by step-like structures and choosing the nearest grid face is no longer unique, as illustrated in Fig. 4. In these cases, the orientation of the real wall cannot be sufficiently represented by one grid face but is approximated by grid faces with perpendicular orientation. For this reason, we virtually sampled surface quantities at the two perpendicular surfaces
and calculated the modelling counterpart of the observation as the average of these values. In the graphs of the surface temperature, the sampled values are plotted by thin dashed lines in addition to their average representing the modelled value which is shown by thick solid lines. Implications of this for the model evaluation as well as for the comparability of the model to the observations are discussed in Sect. 5.1.7, along with the grid discretization.

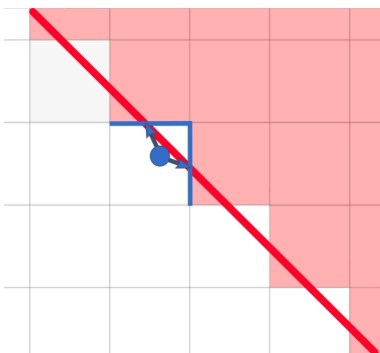

**Figure 4.** Sketch to illustrate the mapping of a wall surface observation point to a gridded step-wise approximation of the wall. The red line represents the real wall surface, light grey lines delineate the grid cells, the light red area shows the footprint of the gridded building, blue circle the surface evaluation point, and the blue arrows represent the assignment of this point to the grid faces (blue lines) used for calculation of the corresponding modelled values.

## 4   Evaluation of model simulation setup

To ensure the correct model couple setup and correspondence to general meteorological conditions, basic characteristics are evaluated in this section. This includes the evaluation of the driving synoptic-scale simulations of WRF and CAMx models, the vertical representation of the boundary layer in PALM, and the spatial development of the turbulent flow characteristics from the boundaries of the PALM parent and child domains. Special focus is put on the summer e2 and winter e3 episodes, in which IR camera observations took place. A description of the statistical methods used is given in the appendix A.

## 4.1 Meteorology

### 4.1.1 Evaluation of the driving synoptic-scale simulation

Since the boundary conditions for the PALM simulations come from a model simulation as well, we need to check for potential misrepresentation of the real atmospheric conditions. First we assess the overall performance of the WRF model simulation on the synoptic scale by comparing the results with the known state of the atmosphere, represented here by the ERA-Interim reanalysis and atmospheric soundings obtained by the CHMI radiosondes (downloaded from the University of Wyoming database; http://weather.uwyo.edu/upperair/sounding.html). Fig. S13 and Fig. S14 show maps of geopotential height at 500 hPa and 850 hPa comparing the results of the WRF simulation (9km domain) with the ERA-Interim reanalysis. Generally, the WRF simulations, being driven by the Global Forecast System (GFS), correspond well to the ERA-Interim reanalysis in terms of the 500 hPa geopotential height field, with some shifts of the pressure field eastward on 19 July and northward on 21 July. Geopotential height at 850 hPa is also very well represented with some added detail, mainly during the day in the summer due to a better resolved topography in the higher-resolution regional model simulation.

Additionally, we compared the WRF results with atmospheric soundings for the station closest to our domain of interest, Praha-Libuš, which is about 11 km south-southeast of the modelled area. Figures 5 and 6 show observed and modelled profiles of the potential temperature and wind speed at the sounding location for 20–21 July (episode summer e2) and 4–5 December (episode winter e3), respectively. Graphs for other episodes are provided in supplements (Fig. S15, S16, and S17). The radiosonde measurements are taken three times per day at 00, 06 and 12 UTC. The modelled values are inferred from the 1 km resolution WRF model. In order to estimate spatial variability and consequently the utility of the sounding for validation of the WRF profiles within the PALM domain, WRF profiles for the centre of the PALM domain are also shown. Modelled profiles from the PALM parent domain simulation are also included in these graphs; these are discussed in Sect. 4.1.2 below.

WRF profiles of potential temperature generally correspond well with the observations with some notable exceptions near the surface, where WRF tends to underestimate nighttime stability and shows less marked near-surface instability during daytime in the summer case. However, here we emphasize that the near-surface profiles might also be affected by the fact that the relevant WRF model surface is not necessarily representative of local detail. The WRF wind-speed profiles also mainly reflect the conditions as observed, with a well modelled nighttime low-level jet (e.g. 21 July at 00:00 UTC, 5 December at 06:00 UTC). However, compared to potential temperature, modelled wind speed exhibits larger discrepancies to observations at various times, for example 20 July at 00:00 and 21 July at 12:00, and also tends to be higher, especially near the surface in the winter scenario. As discussed in the preceding paragraph, the radiosonde location is not within the PALM model domain, However, WRF profiles at the radiosonde location and the PALM domain center show only marginal differences. Hence, we are confident that the modelled boundary layer profiles from WRF, which are used as boundary conditions for PALM, are a sufficiently good representation of reality for this study.

Another factor needing consideration is that in the summer cases the boundary-layer depth during the daytime is within the range of the 1 km horizontal grid resolution in the WRF simulations. Ching et al. (2014); Zhou et al. (2014) showed that in such situations resolved scale convection can develop, also altering the boundary-layer representation and leading to too large a

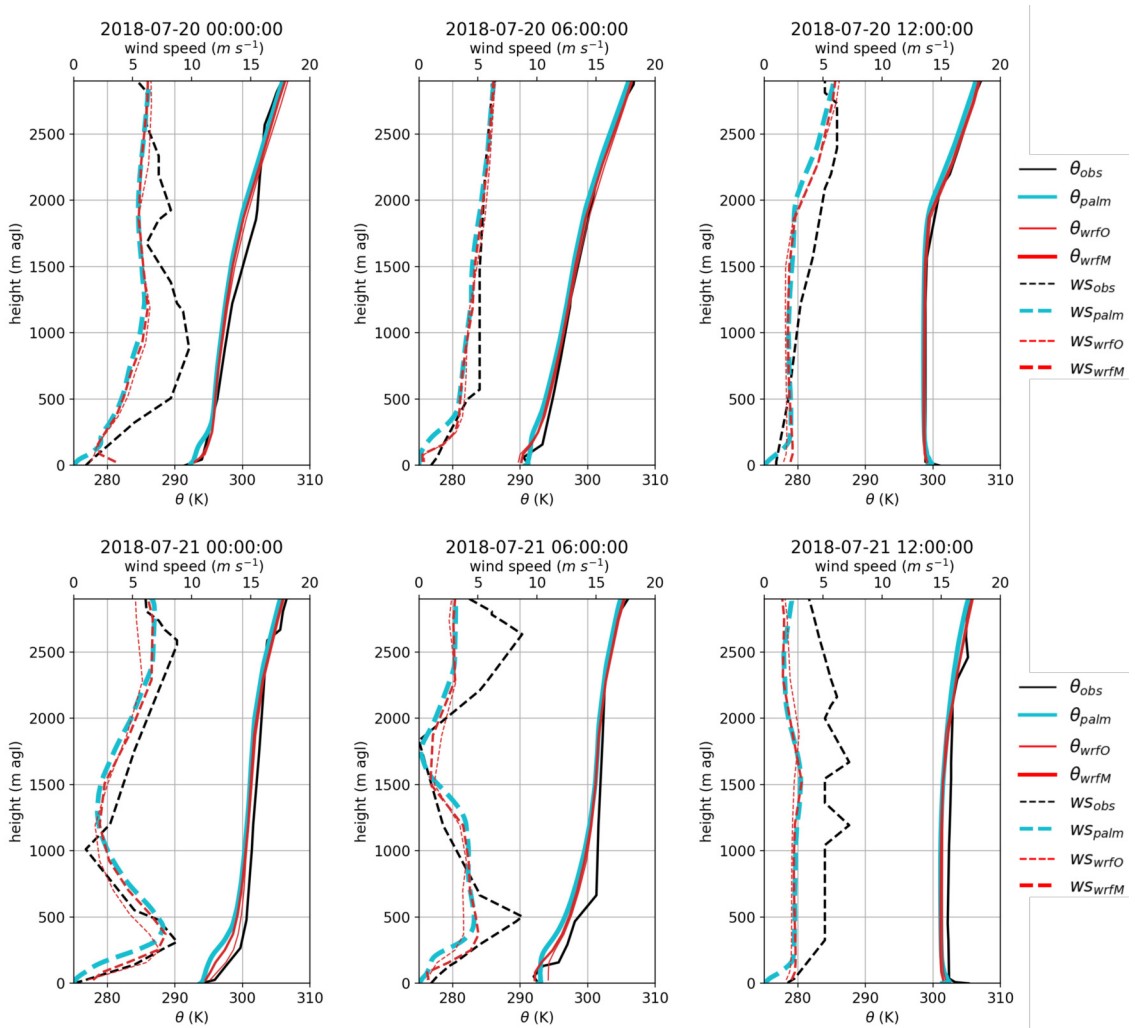

**Figure 5.** Vertical profiles of potential temperature and wind speed from the radiosonde observations at Praha-Libuš station for 20–21 July, with corresponding WRF (1 km horizontal resolution) and PALM (average from parent 10 m resolution domain) profiles. The potential temperature is represented by the solid lines, the wind speed is denoted by the dashed lines. The black line is the sounding observation, the cyan line the PALM model, the red line the WRF model. The thin red line is the WRF model at the sounding location while the thick red line is the WRF model in the centre of the PALM domain.

vertical energy transport. For an LES nested into a mesoscale WRF simulation, Mazzaro et al. (2017) showed that such under-470 resolved convection may propagate into the LES domain, biasing the location of the updrafts and downdrafts. In order not to bias our simulation results by under-resolved convection in WRF propagating into the LES, we checked the WRF-simulation output for the occurrence of under-resolved convection but did not find any (not shown).

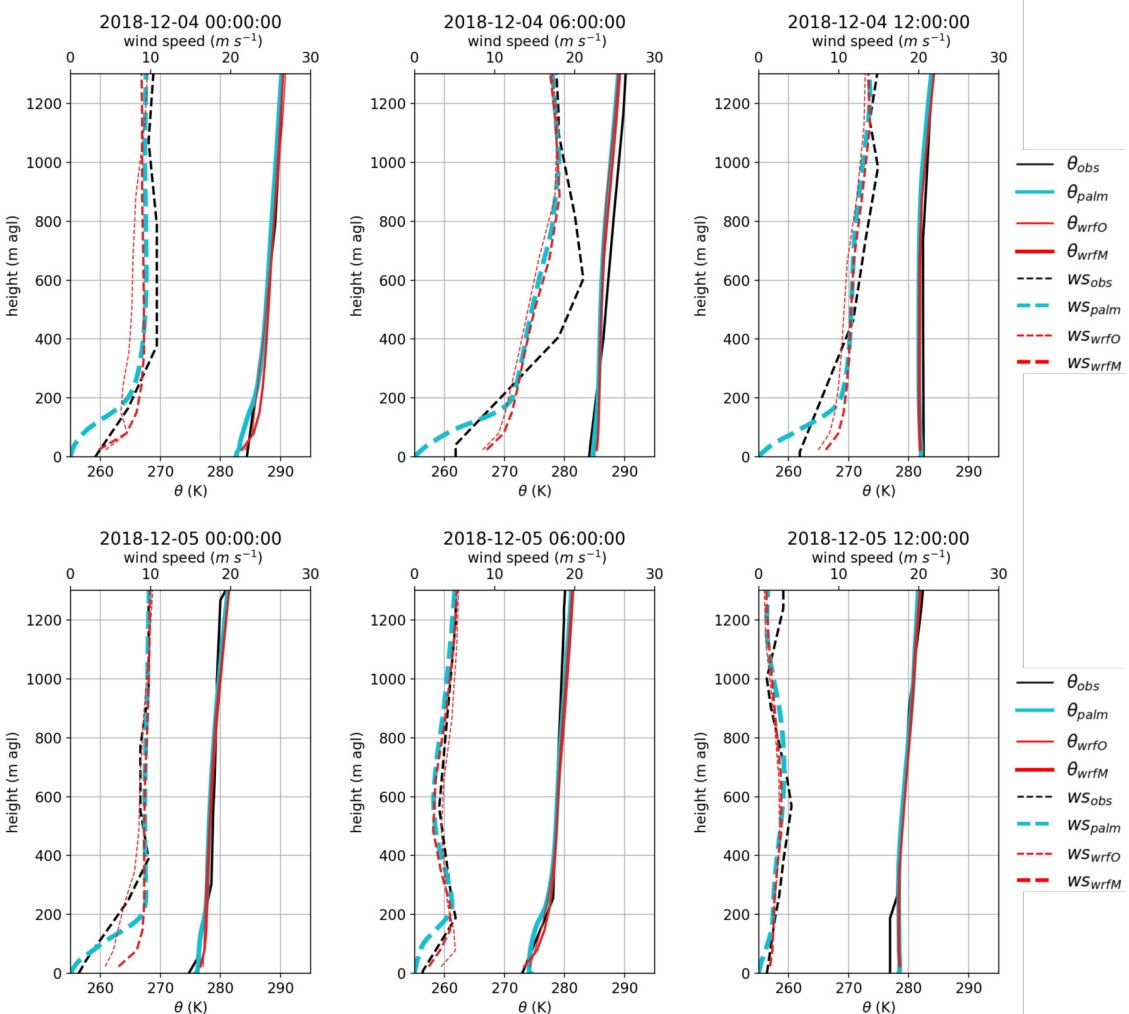

**Figure 6.** Vertical profile of potential temperature and wind speed from the radiosonde observations at Praha-Libuš station, with corresponding WRF (1km horizontal resolution) and PALM (average from parent 10 m resolution domain) profiles for 4–5 December. The potential temperature is represented by the solid lines, the wind speed is denoted by the dashed lines. The black line is the sounding observation, the cyan line the PALM model, the red line the WRF model. The thin red line is the WRF model at the sounding location while the thick red line is the WRF model in the centre of the PALM domain.

In the PALM simulations we prescribed the incoming long and shortwave radiation obtained from the WRF simulations. To check for potential errors in incoming radiation, we compare downwelling SW radiation as simulated by WRF in the

grid box covering the center of the PALM child domain with observations at two CHMI stations in Prague with continuous downward short wave radiation measurements: Praha-Karlov approx. 4 km southeast from the modelled area, and Praha-Libuš 11 km south-southeast (Fig. 7). WRF simulations show good agreement with observations in the summer campaign, with some

overestimation of the SW radiation on 14 and 23 July at noon which we attribute to the underestimation of cloud cover in the WRF simulation. During the winter campaign, the downwelling SW radiation in WRF agrees with the observation on 26,

28, and 29 November, and on 5 December, while WRF significantly overestimates the SW radiation on other days due to underestimated cloud cover.

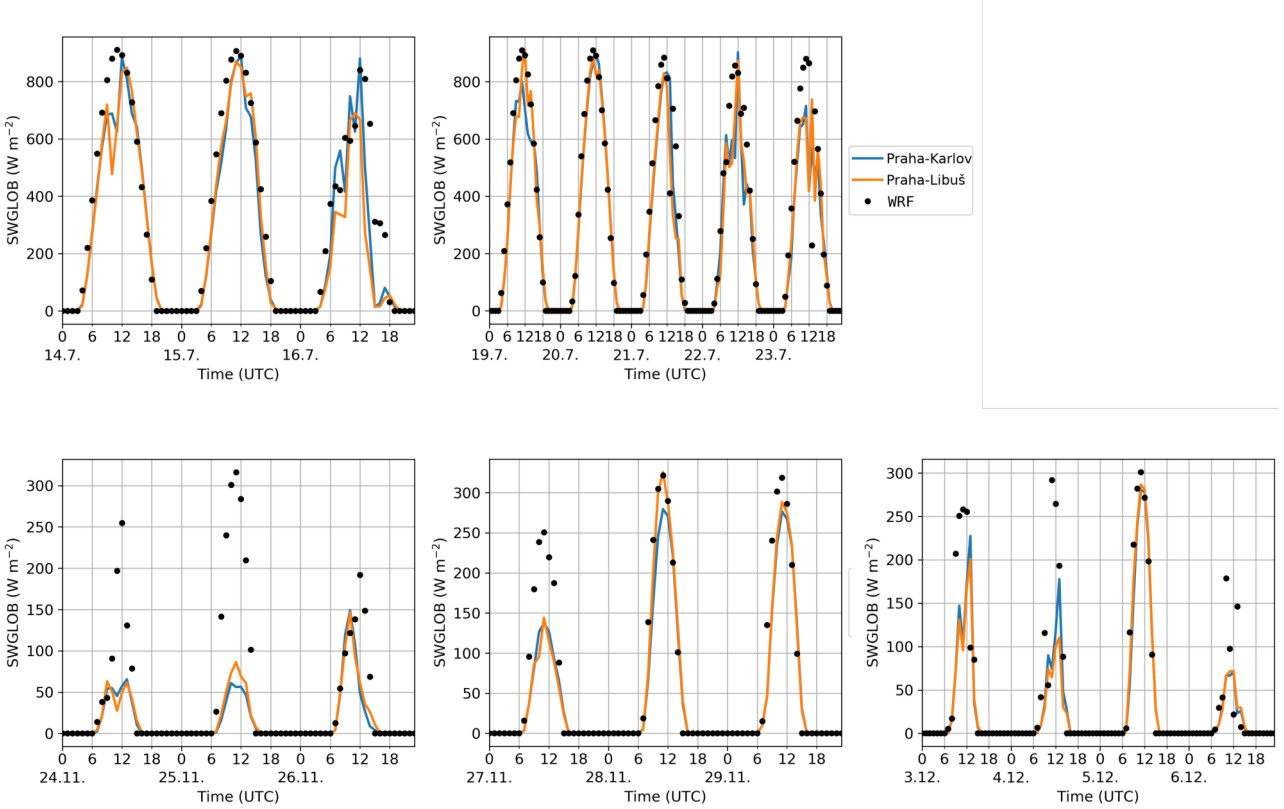

**Figure 7.** WRF modelled and observed downwelling SW radiation for modelling episodes summer e1, e2 (top row) and winter e1, e2, e3 (bottom row): CHMI station Praha-Karlov (blue line); CHMI station Praha-Libuš (orange line); WRF simulation (black dots).

### 4.1.2   Boundary-layer representation in PALM

In order to check whether the observed boundary-layer structure is represented realistically by the LES simulation, we compare domain-average model results from the parent domain against radio-soundings from the Praha-Libuš station located roughly

11 km south-southeast of our area of interest. Praha-Libuš is in an area with slightly different topography and urban topology located at the southern edge of the city, which means that comparison with the model simulation cannot be exact and, especially within the lower parts of the boundary layer, modelled and observed profiles cannot be expected to match. To estimate the spatial variability of the atmosphere between these two locations and thus to assess whether the soundings can be reliably used

for evaluation of the PALM profiles, the WRF modelled profiles in both locations, the sounding location and the PALM area, are provided.

Figure 5 shows vertical profiles of potential temperature and wind speed from PALM together with the soundings for the 20–21 July (episode summer e2). Taking into account the limitations of this comparison, for temperature the model simulations show good agreement with observations, capturing the overall shape of the profile with a slight tendency to underestimate actual values. However, in the lower layers the model tends to underestimate the diurnal variations, showing lower stability during the night and lower instability during the day. The wind speed generally follows the driving WRF profile except near the surface, where the wind speed tends to exhibit lower values due to increased surface friction from the explicit representation of microscale terrain features, buildings, and tall vegetation. During the first night (Fig. 5) the modelled and observed temperature profiles agree well. The modelled wind speed in the residual layer is generally lower than the radiosonde. On the following day, the modelled and observed potential temperature profiles agree very well, both indicating a vertically well mixed boundary layer. During the second night, the modelled profile indicates a cooler boundary layer which is less stable near the surface. On 21 July at 00:00, the wind speed profile agrees well with the measurements. However, at 06:00 the low-level jet is still present in the observations but missing in the simulation. On the following day, again the modelled and the observed temperature profiles agree, although the modelled boundary layer tends to be cooler by about 1 K. The wind weakens during the day and is lower than the observations throughout the entire depth of the model domain.

Figure 6 shows the modelled and observed profiles of potential temperature and wind speed for 4–5 December (episode winter e3). During the first night the temperature profile suggests a more pronounced stable boundary layer. On the following day the modelled temperature profile agrees fairly well with the observed profile. On the second night and during the second day the temperature profiles agree reasonably well, even though the modelled profile indicates a slightly warmer near-surface layer of about 1 K. Considering the entire period, wind speed mostly matches the WRF-modelled profiles above 200 m but with some notable discrepancies compared to observations. Near the surface, PALM shows lower wind speeds compared both to the observations and WRF. At this point, however, we would like to emphasise again that a direct comparison between the PALM-modelled profiles and the observation should be taken with care, especially within the near-surface layer where the profiles can be significantly affected by the different local surroundings.

### 4.1.3 Spatial development of the urban boundary layer

As described in Sect. 3.1, the parent domain receives boundary conditions from WRF where turbulent structures are not explicitly resolved. To trigger the spatial development of turbulence in the LES, synthetic turbulence is imposed at the lateral boundaries (Kadasch et al., 2020). However, even though this accelerates the development of turbulence in the LES, it still requires sufficiently large fetch distances for the turbulence to be spatially fully developed. Lee et al. (2018) have pointed out that an insufficiently developed turbulent flow can bias results in urban boundary-layer simulations. Hence, in order to assess how the turbulent flow develops within the model domain, Fig. 8 shows horizontal profiles of the turbulent kinetic energy (TKE) in the parent domain as distance from the inflow boundary increases. The TKE was computed as $TKE = 0.5 \cdot \sum \overline{u_i' u_i'}$, with $\overline{u_i' u_i'} = \overline{u_i u_i} - \overline{u_i}\,\overline{u_i}$; the overbar denotes a 30-min temporal average. For each grid point we determined its distance to the

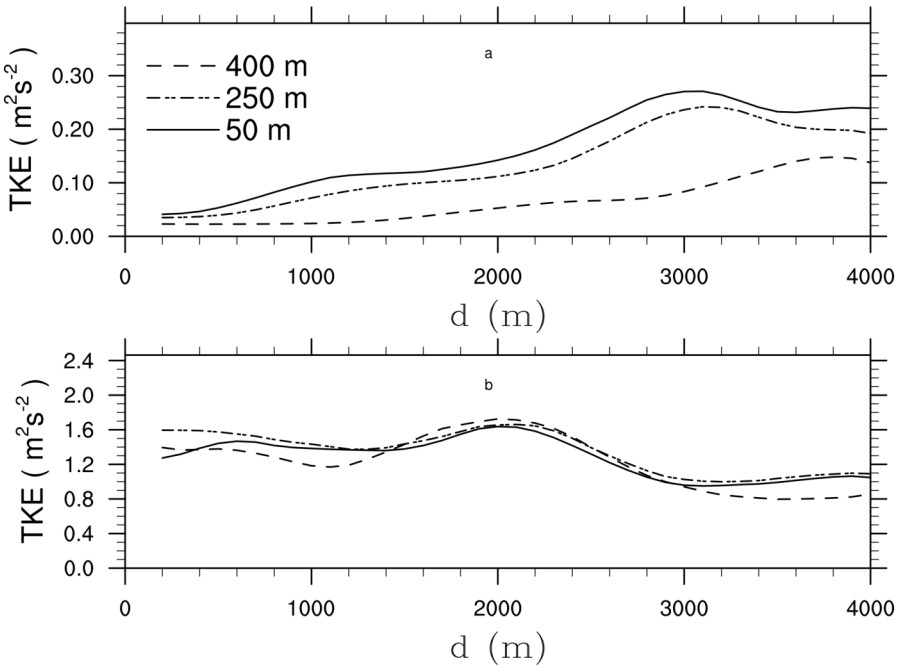

**Figure 8.** Horizontal profiles of 30-min time-averaged resolved-scale turbulent kinetic energy (TKE) in the parent domain plotted against distance from the inflow boundary (d) for a) the winter case at 14 UTC on 5 December and b) the summer case at 13 UTC on 20 July. The TKE is shown for heights at 50 m, 250 m and 400 m above the terrain surface.

inflow boundary for the given wind direction. In doing this, we calculated backward trajectories from the mean wind direction and determined the distance between the sampling location and the intersection point of the backward trajectory with the

closest inflow boundary. Further, variances were averaged over similar distances to the inflow boundary; we then sorted similar distances into equally-sized bins of 100 m to obtain a sufficiently large sample size for each discrete distance. Furthermore, we note that the TKE is evaluated at relative heights above the surface. In the winter case, which is characterized by neutrally-stratified conditions at the given time point (see Fig. 6), the TKE increases with increasing distances from the inflow boundary at all illustrated heights and peaks at about $d = 3000\,\mathrm{m}$ in the surface layer, while the peak position at larger heights is shifted

towards larger distances. In the summer case, which is characterized by convective conditions at the given time point, the TKE is approximately constant up to 2 km from the inflow boundary and then slightly decreases with further increasing distances. However, the heterogeneous orography and nature of the buildings means local effects that will also play a role so we would not expect to obtain a constant equilibrium TKE value. Taking into account that the child domain inflow boundary is placed at about $2\,\mathrm{km}$ from the parent inflow boundary in both cases, turbulence has already been developed at the child domain

boundary, so we are confident that the error made by the too short adjustment fetch length is minor, though we emphasize that

especially for the winter case larger horizontal extents of the parent domain are also desirable in order to better represent mixing processes in the upper parts of the boundary layer. Moreover, the turbulent flow depends on the upstream surface conditions (terrain, buildings, land-use, etc.) which in turn depend on the wind direction. With insufficiently large model domains such effects might not be well represented. However, as our validation study mainly focuses on the building layer where turbulence 540 is produced by building-induced shear, we believe the error induced by not completely representative upstream conditions is small and does not significantly affect our validation results.

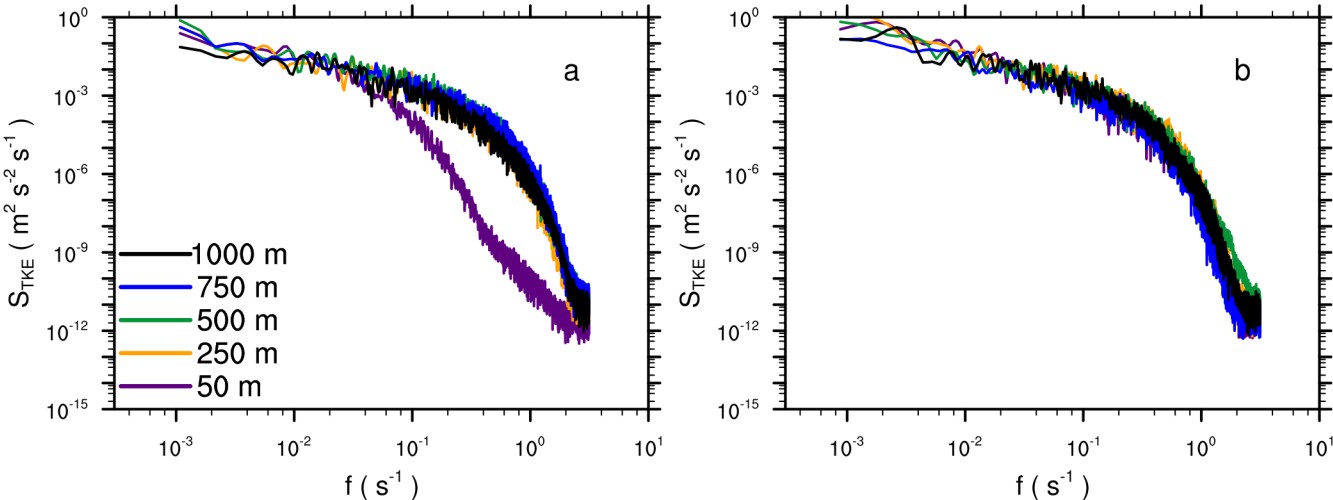

**Figure 9.** Frequency spectra of the TKE within the child domain at z = 50 m above the surface evaluated at locations with different distances downstream of the inflow boundary for a) the winter case at 14 UTC on 5 December, and b) the summer case at 13 UTC on 20 July.

Beside the transition of the turbulent flow in the parent domain, the flow also undergoes a transition after entering the child domain with its finer grid resolution as discussed in detail in Hellsten et al. (2020). In order to evaluate whether turbulence has been sufficiently adapted within the child domain at locations where simulation results are compared against observations, Fig. 545 9 shows frequency spectra of the TKE at different distances to the inflow boundary. We sampled time series of the velocity components at different positions over one hour and calculated the spectra for each sampling location; afterwards we averaged over all spectra with similar distance to the inflow boundary. In the winter case the spectra close to the inflow boundary show a significant drop-off of energy at smaller frequencies compared to spectra at distances $>= 250\,\mathrm{m}$, indicating that especially the smaller scales are still not sufficiently resolved on the numerical grid, while at larger distances no dependence on the 550 sampling location can be observed. In the summer case the flow transition from the coarse into the fine grid is even faster; even spectra close to the inflow boundary indicate similar turbulence properties compared to the locations farther downstream. This is also in agreement with the findings presented in Hellsten et al. (2020) that under convective conditions the transition is small compared to neutrally-stratified or stable conditions as TKE is mainly produced locally by buoyancy rather than by shear.

**Table 1.** Evaluation of CAMx 1-h concentrations against urban background stations for summer and winter episodes.

| | $NO_X$ | | $PM_{10}$ | |
|---|---|---|---|---|
| | **Summer** | **Winter** | **Summer** | **Winter** |
| **N** | 684 | 816 | 907 | 1078 |
| **mean obs** ($\mu$g.m$^{-3}$) | 22.6 | 59.5 | 22.1 | 30.4 |
| **mean mod** ($\mu$g.m$^{-3}$) | 10.1 | 24.4 | 13.4 | 33.3 |
| **FB** | -0.76 | -0.84 | -0.49 | 0.09 |
| **NMSE** | 1.51 | 2.15 | 0.65 | 0.53 |
| **FAC2** | 0.38 | 0.31 | 0.50 | 0.69 |
| **R** | 0.54 | 0.28 | 0.34 | 0.13 |

**N** = ensemble size; **mean obs** = observed mean value; **mean mod** = modelled mean value; **FB** = fractional bias; **NMSE** = normalized mean square error; **R** = Pearson correlation coefficient.

## 4.2 Air quality

For the CAMx model evaluation, urban background air quality monitoring stations closest to the PALM parent domain were used (see Sect. 2.3.6). Validation was performed for hourly average concentrations of $NO_X$ and $PM_{10}$. Evaluation was done for all PALM simulation episodes which were then grouped as summer and winter. Metrics according to Britter and Schatzmann (2007) and Chang and Hanna (2004) for both campaigns are summarized in Table 1. For graphs of diurnal variation plotted by openair package (Carslaw and Ropkins, 2012) see Fig. S18.

For $NO_X$, the metrics show a significant underprediction of the measured concentrations (FB ca. –0.8) for both summer and winter episodes. Nevertheless, the diurnal variation is captured quite well, although in winter modelled peaks in the evening are larger than in the morning while the reverse is seen in the observed data.

Summer $PM_{10}$ concentrations are less underestimated with FB ca. –0.5, and morning and evening peaks are sharper and appear about 1 h earlier than in observations. Winter $PM_{10}$ are even slightly overestimated but the CAMx model is not able 565 to represent their real diurnal variation. Modelled diurnal variation is very similar to that for $NO_X$, which indicates that it is dominated by diurnal variation of traffic, while in reality different sources play a important role as well.

## 5 Results

### 5.1 Surface temperature

In the following section we will discuss the model performance with respect to the surface temperature. First, we will show 570 general surface temperature results and show an example of direct comparison against observed values. Then we will draw a broader picture of model performance for different types of surfaces, supported by relevant statistical measures. Subsequently,

particular cases at individual locations will be presented and the related shortcomings of the model and the observations, as well as the implications of the shortcomings of the fine-scale input data, are discussed.

### 5.1.1 Overall performance

Figure 10 shows an example of a 3D view of instantaneous surface temperature in the child domain at 13:00 UTC 20 July. The heterogeneous distribution of surface temperature reflects the distribution of pavements and green areas, with higher temperatures over paved areas and at building walls and roofs. Below the trees, where most of the shortwave direct radiation is absorbed within tree crowns, surface temperatures of about 290 K are modelled (e.g. on the right side of the figure or within courtyards), while higher surface temperatures up to 330 K are modelled at intensively irradiated vertical building

walls. Moreover, the effect of different wall and roof material parameters on surface temperature can be identified, with roofs showing lower surface temperatures where green fractions are present, while some other walls and roofs show values up to 320 K. In order to evaluate the modelled surface temperature more quantitatively, we compare the modelled surface temperature against observed values in the following parts of this section.

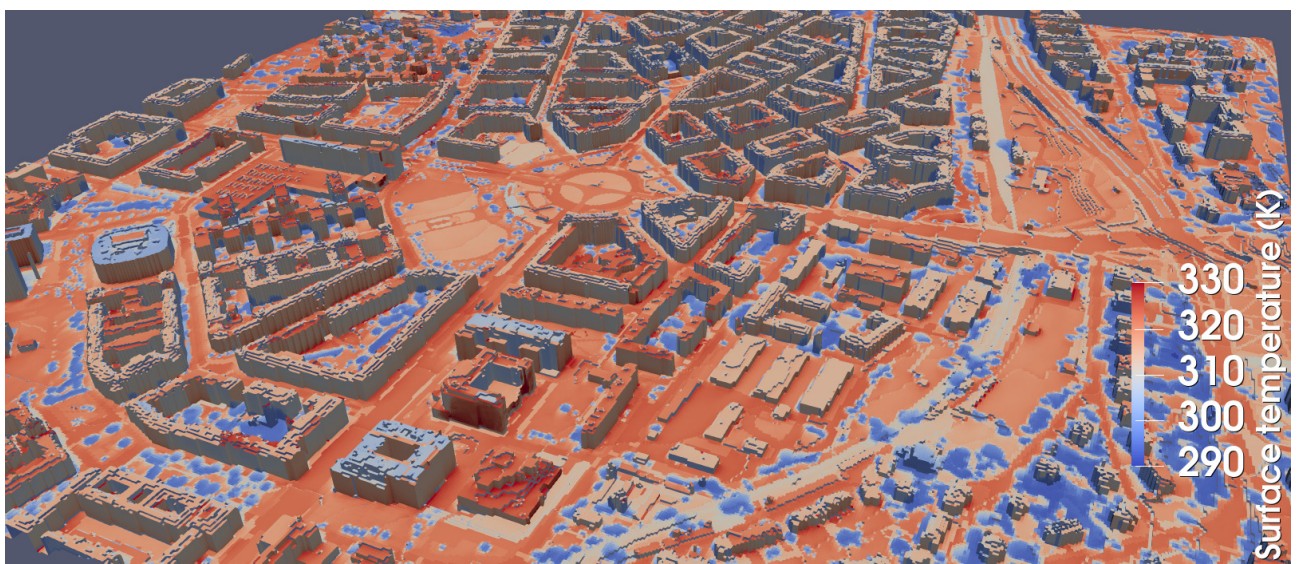

**Figure 10.** Example 3D view of the child modelling domain with 2m resolution from the south-west direction on 20 July at 13:00 UTC (14:00 CET). The colour scale represents the modelled surface temperature.

Figure 11 shows an example for the observed and modelled diurnal cycle of surface temperature profiles at one particular

evaluation location, 11-1, along with a street view of the location area and the RGB and IR views of the location with the evaluation points labelled. 11-1 is located on Evropská třída street, a west-east oriented boulevard between 40 and 50 m in width (building to building), with evaluation points placed on the concrete tramway belt, pavement, and on the nearly south oriented wall of two traditional five-floor brick buildings, the left of which has an additional thermal insulation layer. For the

summer scenario, the modelled surface temperature agrees fairly well at the horizontal and vertical locations with respect to
the diurnal amplitude and temporal evolution. However, at the horizontal surfaces the modelled nighttime surface temperatures
are underestimated by about 3–4 K. When the sun comes up the next day the modelled surface temperature again matches the
observed surface temperature, so the nighttime bias in surface temperature does not propagate into the next day simulation. In
the winter case the modelled surface temperatures also agree with the observations, except for the nights where the modelled
surface temperatures are about 1–2 K higher than the observed ones at both horizontal and vertical surfaces. Further, two sharp
peaks in the modelled daytime surface temperatures during the morning hours as well as during early afternoon hours are
striking, and not present in the observations. Similar peaks can be also observed at some other locations mainly during the
winter episode. For a detailed discussion concerning these peaks we refer to Sect. 5.1.3 where this effect and its causes are
analysed.

A complete set of modelled and observed diurnal cycles of surface temperature for all evaluation points in all observation
locations (see Fig. 1 in Sect. 2.1) for the summer e2 episode (19–21 July 2018) and for the winter e3 episode (4–6 December
2018) is given in the supplements in Sect. S3. As supporting information, the graphs of the modelled values of the surface
sensible heat flux, ground heat flux, net radiation, and incoming and outgoing shortwave and longwave radiation are also
available in the supplements in Sect. S4.

The observations cover a wide range of surface types. Since we cannot show daily cycles for all observation points we
condensed the results to show the general performance of the ground and wall modelling capability of PALM. To distinguish
model behaviour for different types of surfaces, the evaluation points were put into the following categories: pavements (paved
areas without traffic), streets (paved areas with traffic), grass, wall of traditional building, wall of contemporary office building,
wall of building with glass or glass-like surface, and plant canopy affected surface. The complete assignment of the evaluation
points to the particular categories is given in table Table S7. Figure 12 shows scatter plots of the modelled and observed
surface temperature for particular surface types during the summer e2 episode. The best agreement can be observed for street
and pavement surfaces, and traditional building walls. At lower temperatures (which corresponds to nighttime values) the
scatter is generally lower compared to higher surfaces temperatures, where, especially at the buildings, a wide scatter can be
observed. To support this qualitative impression from the scatter plots, Table 2 provides statistical error measures. Modelled
surface temperatures at pavements and streets are slightly too cool, especially at nighttime, as indicated by the negative bias.
Further, the RMSE indicates higher uncertainty at daytime and lower uncertainty at nighttime, especially at building walls. The
main reason for this behaviour is probably the typically lower thermal conductivity in comparison with ground surfaces, which
causes more rapid reactions of the surface temperature to the changes in radiative forcing. This effect, in connection with binary
changes in direct radiation during the course of the day due to shading effects, along with possible geometrical imperfections in
the discretized terrain and building model, can cause temporally and spatially limited strong discrepancies between modelled
and observed point values. This issue is analysed in more detail using the example of location 11-1_V in Sect. 5.1.5. Mismatch
of shading can also be caused by the imprecise description of the shapes of the tree crowns (see Sect. 5.1.6) . Modelled surface
temperatures at grass-like surfaces also show good agreement with the observations, with mostly low scatter both during the
day and at night, but with slightly overestimated nighttime values. A wider scatter, even at lower temperatures, can be observed

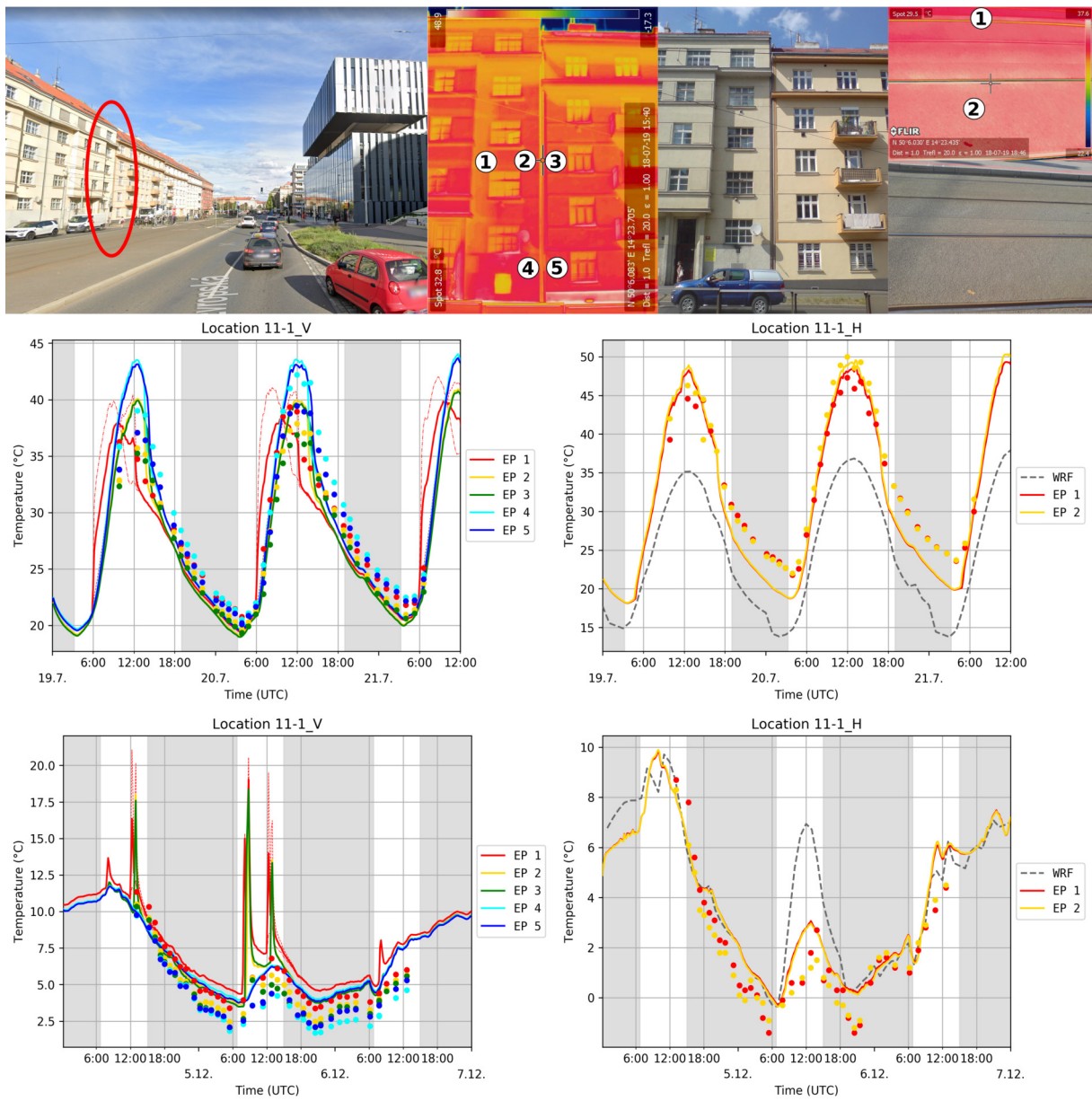

**Figure 11.** Observation location 11-1: the view of the observation location and IR and RGB photos with placement of the evaluation points (upper row) and graphs of observed (dots) and modelled (lines) surface temperature for wall (left) and ground (right) for particular evaluation points (EP) for summer e2 (middle) and winter e3 (bottom) episodes. The modelled values come from the child PALM domain, the dotted and dashed lines represent the modelled temperature for left and right grid faces (see Sect. 5.1.1). The grey dashed line shows the corresponding WRF skin layer temperature for horizontal surfaces. The grey areas denote the night time. Top left image © 2020 Google.

**Table 2.** Statistics of observed and modelled surface temperatures (K) for the simulation episodes summer e2 and winter e3.

| Surface type | | All times | | | Daytime | | | Nighttime | | |
|---|---|---|---|---|---|---|---|---|---|---|
| | | MB | MAB | RMSE | MB | MAB | RMSE | MB | MAB | RMSE |
| Pavements | S | -0.7 | 2.1 | 2.7 | -0.2 | 2.3 | 3.0 | -1.6 | 1.7 | 2.0 |
| | W | 1.5 | 1.7 | 1.9 | 1.9 | 2.0 | 2.4 | 1.4 | 1.5 | 1.7 |
| Streets | S | -1.6 | 2.5 | 3.2 | -1.4 | 2.7 | 3.6 | -2.1 | 2.1 | 2.3 |
| | W | 0.9 | 1.0 | 1.4 | 1.3 | 1.4 | 2.0 | 0.7 | 0.8 | 1.0 |
| Grass | S | 0.6 | 2.7 | 4.1 | 0.3 | 3.2 | 4.9 | 1.1 | 1.7 | 2.0 |
| | W | 1.2 | 1.5 | 2.1 | 1.5 | 1.9 | 2.7 | 1.0 | 1.3 | 1.8 |
| Walls (trad. building) | S | -0.5 | 2.0 | 3.3 | -0.3 | 2.5 | 3.9 | -0.9 | 1.1 | 1.4 |
| | W | 1.7 | 1.9 | 2.6 | 2.2 | 2.3 | 3.5 | 1.5 | 1.7 | 2.0 |
| Walls (contemp. building) | S | -0.1 | 5.5 | 7.4 | -0.4 | 6.4 | 8.8 | 0.2 | 4.2 | 4.5 |
| | W | 4.9 | 5.1 | 6.8 | 5.8 | 6.3 | 9.6 | 4.5 | 4.5 | 5.1 |
| Walls (glass-like) | S | 1.9 | 3.6 | 5.3 | 1.8 | 4.2 | 6.2 | 2.1 | 2.6 | 3.2 |
| | W | 7.1 | 7.1 | 7.9 | 6.8 | 6.8 | 7.8 | 7.2 | 7.2 | 8.0 |
| Plant canopy affected | S | -0.8 | 2.5 | 3.6 | -0.7 | 2.8 | 4.1 | -1.0 | 1.6 | 1.8 |
| | W | 1.0 | 1.5 | 1.9 | 1.3 | 1.7 | 2.1 | 0.9 | 1.4 | 1.7 |

**S** = summer e2 episode; **W** = winter e3 episode; **MB** = mean bias; **MAB** = mean absolute bias; **RMSE** = root mean square error.

for both glass-like surfaces and contemporary buildings walls, with the largest RMSE in the daytime. The reason for this higher
spread is probably a more complex wall structure and the higher uncertainty in its identification (see Sect. 5.1.3). In the case of
glass-like surfaces these causes are accompanied by the fact that the IR camera photos of such locations contain a substantial
amount of reflections from other surfaces (opposite buildings, sky), and so does not provide an adequate measure of the surface
temperature. These effects are discussed in detail in Sect. 5.1.4.

     Similarly, Fig. 13 shows scatter plots for episode winter e3. Again, at streets, pavements, grass-like, and traditional wall
surfaces the scatter is relatively low though it does not show large difference between daytime and nighttime (see also RMSE
in Table 2), in contrast to the summer case. In general, it is striking that in the winter case modelled surface temperatures are
slightly overestimated, as indicated by the positive bias values. This is especially true for glass-like materials which show far
too high modelled surface temperatures as well as a large scatter. However, the problems of surface temperature measurements
of glass-like surfaces by IR cameras due to direct reflection from other surfaces, which is mentioned above and discussed
in detail in Sect. 5.1.4, applies here. Grass surfaces modelled temperatures are also overestimated. This overestimation can
be seen in many individual locations (see supplements Sect. S3). The reason for this overestimation of surface temperatures,
which is more pronounced in wintertime (compare Fig. 12) than in summertime, however, remains unknown at this point.
There is further discussion of modelling grass surfaces in summertime and the necessary pre-requisites below in Sect. 5.1.2.

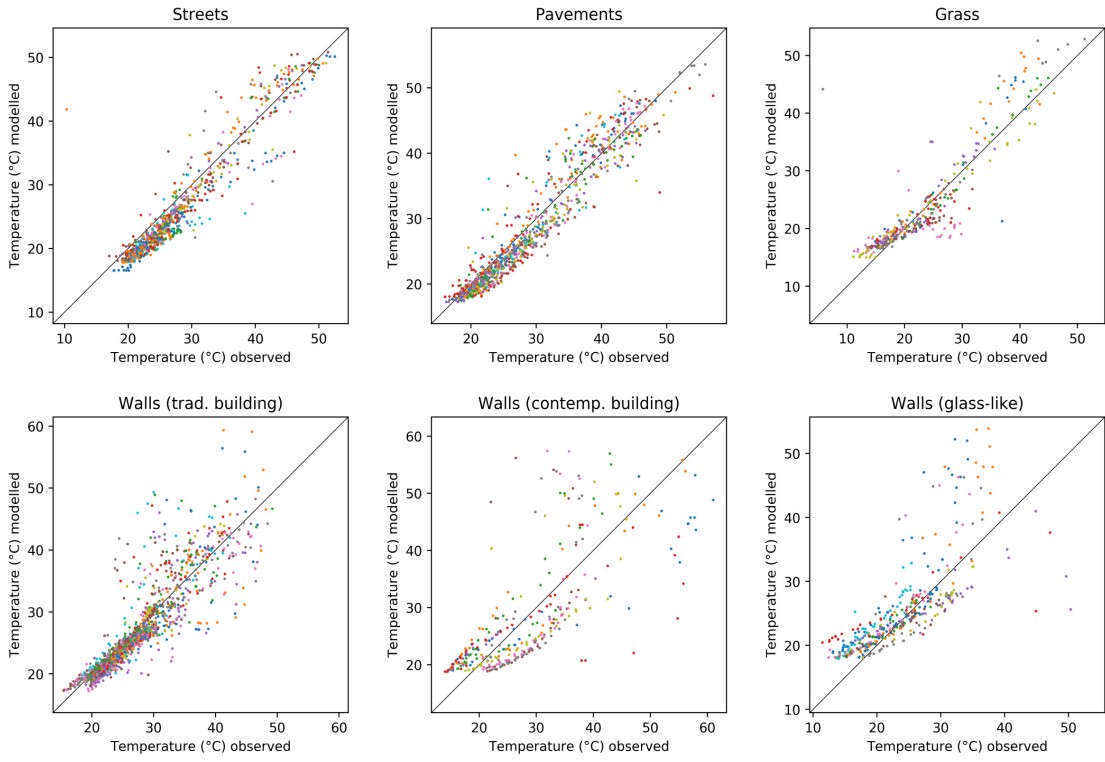

**Figure 12.** Scatter plots of the modelled and observed surface temperature for particular types of the surfaces in Table S7 during the summer e2 episode. Individual evaluation points are plotted by dots of different colours.

### 5.1.2   Grass surfaces

The energy balance of a grass-covered area may strongly depend on soil-water content, assumed plant cover, leaf-area index (LAI), along with other factors (Gehrke et al., 2020), and these are mostly unknown in this study. Let us examine three grass covered points, evaluation point 3 (EP 3) at location 05-1_H, EP 2 at location 06-3_H, and EP 1 at location 10-3_H during the second day of the summer e2 episode, 20 July (see Fig. 14 and Sect. S3 of the supplements for detailed information on these particular locations). These points are not significantly influenced by any adjacent tree or wall and thus they are not affected

by possible imperfection of the radiative transfer in the model. These points represent examples of three different grass-type surfaces. The first point is placed in a recently built park with an integrated irrigation system, the second one is located in a green tram line with a shallow soil layer and without any watering, and the third point is located in a quite large lawn in an open square area with a deep soil layer without watering, and so resembling natural grass conditions. To account for local differences in soil conditions for summer simulations, the grass areas within the model domain were split into three categories:

natural-like grass, watered grass, and an urban grass type, and the original WRF soil moisture was roughly adjusted by factors of 1.0, 2.0, and 0.5 respectively. Since we have no information about soil moisture at that level of detail, the chosen adjustment

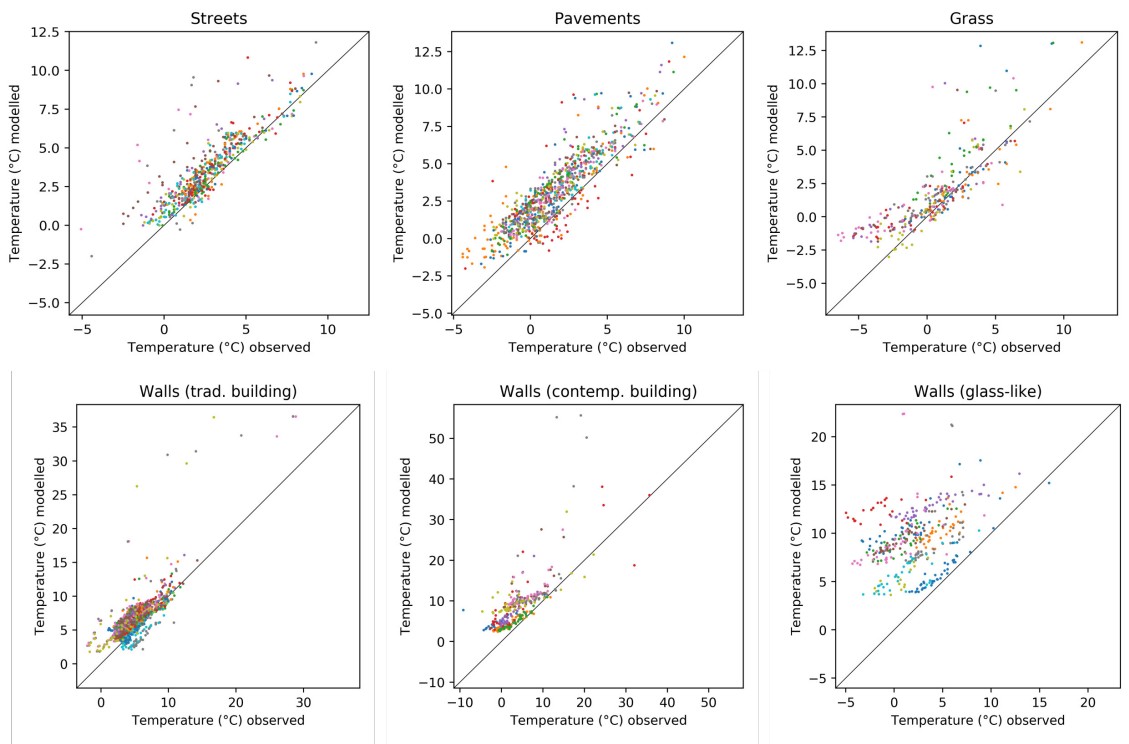

**Figure 13.** Scatter plots of the modelled and observed surface temperature for particular types of the surfaces in Table S7 during the winter e3 episode. Individual evaluation points are plotted by dots of different colours.

factors are a best guess based on a survey of the locations and personal experience. The soil moisture for winter simulations was not adjusted. The diurnal cycle of the modelled and observed surface temperature for the different grass surfaces agrees fairly well with maximum temperatures of 35 °C, 52 °C, and 45 °C, respectively. Figure 14 also shows diurnal cycles of surface temperature at these points from a test simulation where the soil moisture of grass surfaces was uniformly prescribed from the WRF simulation. With non-adjusted soil moisture, the daytime surface temperature at urban grass (location 06-3, EP 2) and watered grass (location 05-1, EP 3) is under- and overestimated compared to observations, respectively, though it agrees fairly well for the adjusted soil-moisture case. This indicates that using correct soil moisture values is a necessary prerequisite to adequately model grass-like surfaces within an urban environment. For additional details concerning the sensitivity of surface temperatures modelled by PALM to the initial soil moisture in urban environments we also refer to Belda et al. (2020). Apart from soil moisture, sensitivity of grass surface temperatures to other parameters such as LAI, plant cover, root-distribution, etc., might also be important. For details in this regard we refer to Gehrke et al. (2020) who studied the sensitivity of the energy-balance components to different soil as well as land-surface parameters.

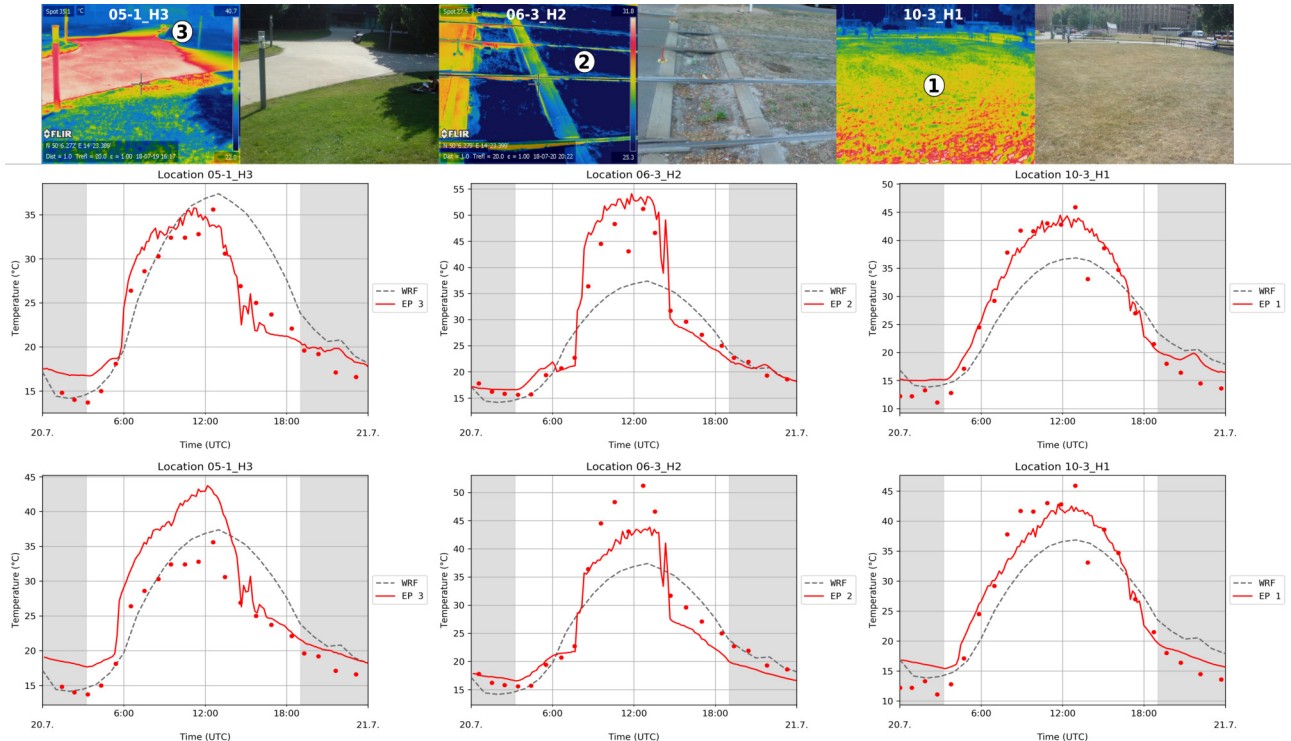

**Figure 14.** IR and RGB photos of locations 05-1_H, 06-3_H, and 10-3_H with placement of selected evaluation points (EP) (upper row) which represent three different grass-type surfaces found in the modelled urban area. The observed and modelled surface temperature at these locations for 20 July 2018 (second day of episode summer e2, middle row), and the same results from a test one-day simulation with all grass surfaces initialised with soil moisture uniformly prescribed from WRF output (bottom row). The grey dashed line shows the corresponding WRF skin layer temperature. The grey areas denote the night time. All results are from the child 2m resolution domain.

### 5.1.3 Complex structure of the walls

In the case of vertical surfaces ("walls"), the model behaves well for most cases of walls of traditional buildings, while walls of contemporary office buildings are modelled less accurately (see Fig. 12 and Fig. 13). We are convinced that the reason for this is the more complex structure of these walls which can not be fully described by the four layers allowed by the current version of the PALM input standard. Moreover, gathering precise information about this type of structure proved to be quite difficult. Let us show an example using evaluation points 2 and 3 at location 02-3_V (see Fig. 15 and Sect. S3 of the supplements for

full information about the location).

    While point 1 is captured by the model quite well except for slight overestimation during the night and morning hours, point 2 evinces an overestimation of around 15°C during the afternoon hours. A closer direct inspection of this wall revealed that it consists of a thin outer layer followed by a 10 cm layer of air before the rest of the wall structure, while in the model all this

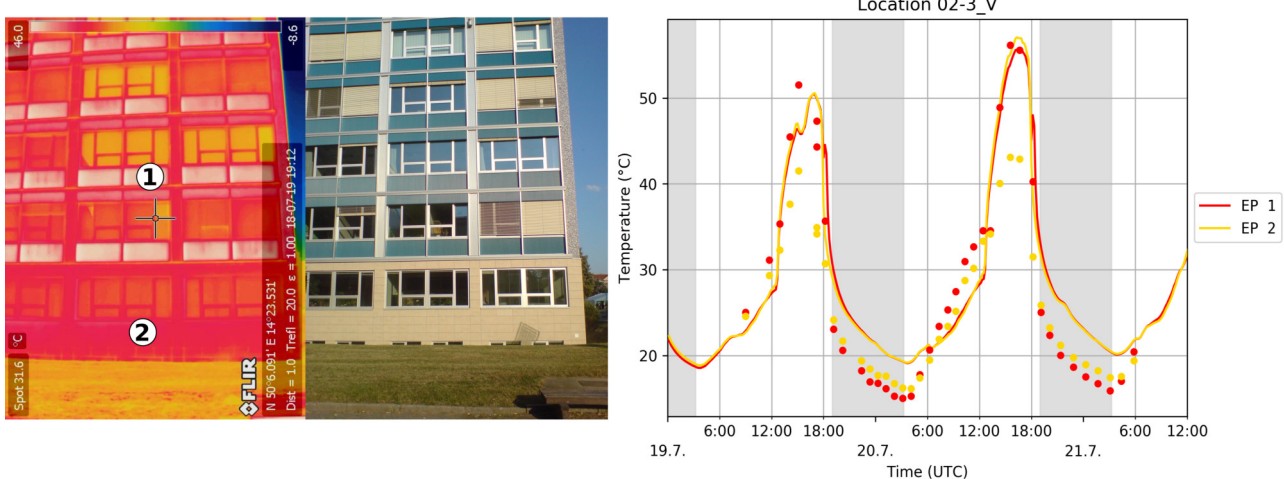

**Figure 15.** Observation location 2-3_V: IR and RGB photos of the building with placement of the evaluation points (left) and graph of observed (dots) and modelled (lines) surface temperature for particular evaluation points (EP) for summer e2 episode (right). The grey areas denote the night time. Top left image © 2020 Google.

is considered as a continuous wall. The observed outermost layer was thus cooled from both sides, an effect which was not
captured by the wall model.

### 5.1.4 Glass surfaces

Some buildings have walls covered with glass or similar types of reflecting surface. These walls present a challenge for both observation and modelling. The main problem is the fact that the surfaces of these buildings are more or less specular, which means that a substantial part of the LW radiation entering the IR camera is a reflection of whatever is behind the camera. For
example, location 11-2_V (see Fig. 16 and Sect. S3 of the supplements for full information about the location) is a north-facing building, the lower part of which has a glass surface. The area of the building around evaluation point 2 reflects the sky into the camera, while the area around evaluation point 3, located just below, reflects the building opposite into the camera (the building opposite is around location 11-1_V). Consequently, the derived values of the surface temperature represents primarily the surface temperature of the reflected object (wall, ground, treetop, sky), not of the observed object itself. This can be well
demonstrated by the different observed values at points EP 2 and EP 3. The modelling of this type of building thus cannot be validated by means of IR camera temperature measurements.

The modelling of the surroundings of these points can be partly influenced by the fact that the current version of RTM considers all surfaces as Lambertian (see Krč et al. 2020). This means that they reflect radiation in all directions in the model while in reality, part of the radiation undergoes specular reflection according the law of reflection. This fact does not directly
affect the reflecting surface itself but it can influence the distribution of reflected SW and LW radiation among nearby surfaces. As the amount of incoming direct radiation is significantly larger than the incoming reflected radiation (direct radiation can

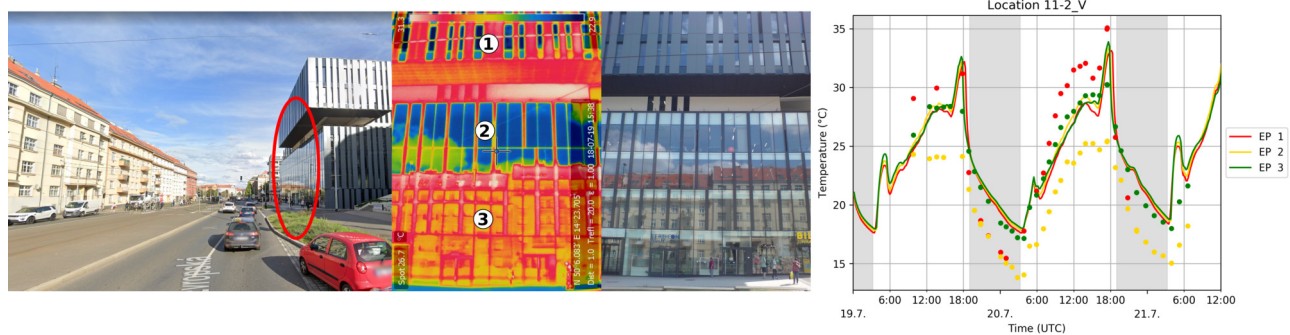

**Figure 16.** Observation location 11-2_V: the view of the observation location (left), IR and RGB photos of the building with placement of the evaluation points (centre) and graph of observed (dots) and modelled (lines) surface temperature for particular evaluation points (EP) for summer e2 episode (right). The grey areas denote the night time. Top left image © 2020 Google.

reach up to 900 W m$^{-2}$ while the reflected radiation is limited to 200 W m$^{-2}$ for most common cases), this effect has usually little practical impact and is masked by the effect of the direct radiation. An example of this effect can be seen in location 06-1_H by comparing evaluation point 2 with EP 3 (see Fig. 17 and alternatively supplements Sect S3 for full information about the location). These points are placed on similar asphalt concrete surfaces but with different distance to the nearby glass facade. While the surface temperature at the more-distant point 3 is modelled well, point 2 is overestimated by about 7 °C on 20 July 2018 at hours 11-13 UTC. The observation at point 2 at these times shows an atypical increase of about 7°C which is not observed at other points placed on the same surface type. We can attribute this lift to the effect of the specular reflection from the glass facade. As this effect is not considered by the model, the model gives similar results for both points EP 2 and EP 3. Results for EP 1 (limestone pavement) are less affected by the missing specular radiation in the model due to its much higher albedo.

### 5.1.5 Rapid changes of surface temperature

Some of the graphs of the surface temperature contain strong "peaks" in the diurnal cycle of the modelled wall temperature (see e.g. Fig. 11). This effect can be seen mainly during the winter episode e.g. at locations 6-4_V, 7-1_V, 7-2_V, 8-2_V, 9-2_V6, 9-2_V7, 11-1_V). Similar peaks can be observed in the corresponding radiative, surface, and ground heat fluxes (see Sect. 4 in supplements). Some of these peaks can also be found in measurements (clearly visible e.g. for 6-4_V), though most observations contain no corresponding peaks. Let us analyse in more detail location 11-1_V (see Fig. 11) where this effect is very strong for evaluation points 1, 2, and 3 on 5 December.

Figure 18 shows the observed IR and RGB camera photos at corresponding observation times together with modelled counterparts at the closest saved model timestep. For easier orientation, Fig. S19 in the supplements shows an overview of the modelled surface temperatures in the given area at the same time-steps. Fig. 19 provides the complete timeline of 10-minute

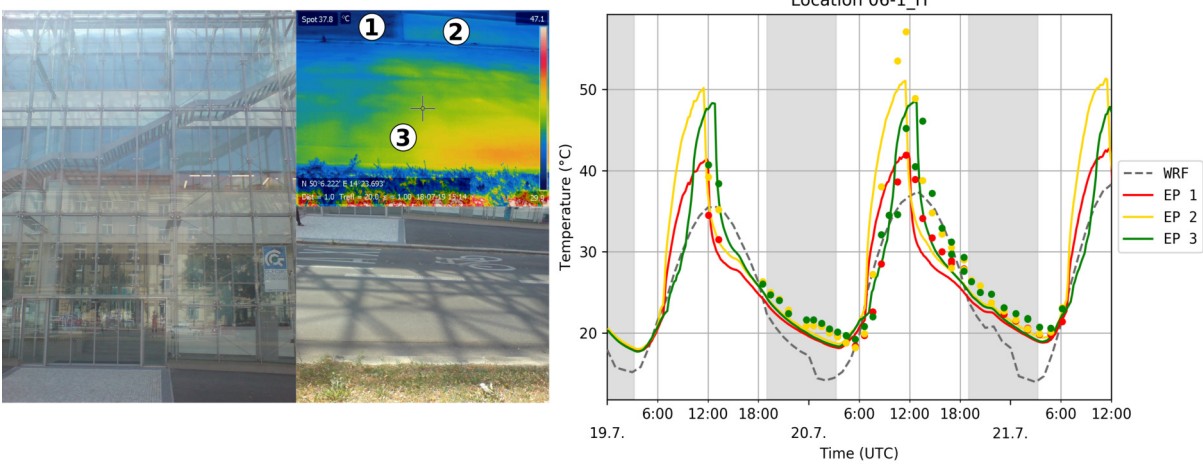

**Figure 17.** Location 06-1: IR and RGB photos of the observation location with placement of the evaluation points (top), observed (dots) and modelled (lines) surface temperature for summer e2 episode (bottom). The grey dashed line shows the corresponding WRF skin layer temperature. The grey areas denote the night time. Top left image © 2020 Google.

model outputs of the wall surface temperature from 5:28 to 12:48 UTC. The time-steps shown in the previous figure Fig. 18 are highlighted with a red frame, the red dots denote the position of the evaluation points 1, 2, and 3.

The first peak takes place between the first and second observation times (7:51 and 9:26 UTC) and thus it does not appear in the IR observations. The situation of the second peak is more complicated. This peak partly overlaps with the fourth observation at 12:48 UTC, which is only reflected in the observations by a very small increase of the surface temperature at point 1. The reason for this can be seen in the comparison of the shading from direct radiation in the RGB photo and the corresponding figure for the modelled SW radiation (see Fig. 18). The shade created by the building on the opposite side of the street is approximately 3 meters lower in the model than in reality at this time. These differences can be attributed to the geometrical imperfections of the digital building elevation model (BEM) used, as well as to the errors introduced by its discretization and by the PALM process of the placing of the buildings on the terrain. One of the sources of the imprecision in BEM can also be peripheral objects on the roof area (e.g. banisters, air-conditioning systems) which create shading but are not considered in BEM (see street view of shading buildings in Fig. S20 in supplements).

Figure 20 shows a detailed graph of location 11-1_V for times from 7:00 to 14:00 UTC and provides additional information about the diurnal cycle of the surface temperature at this location. The evaluation points 1, 2, and 3 correspond to points from the graph in Fig. 11 while new evaluation points 4, 5, and 6 were added on the top layers of the wall. The graph shows that the diurnal variability of the surface temperature in this location has similar magnitude in the model as in the observations. This supports our conclusion that the model (namely the radiative transfer and surface energy balance) works reasonably well and the differences in the values at particular evaluation points and times can be attributed mainly to the geometrical imperfections of the model which produce differences of shading of the direct radiation. These changes of surface temperature also cause

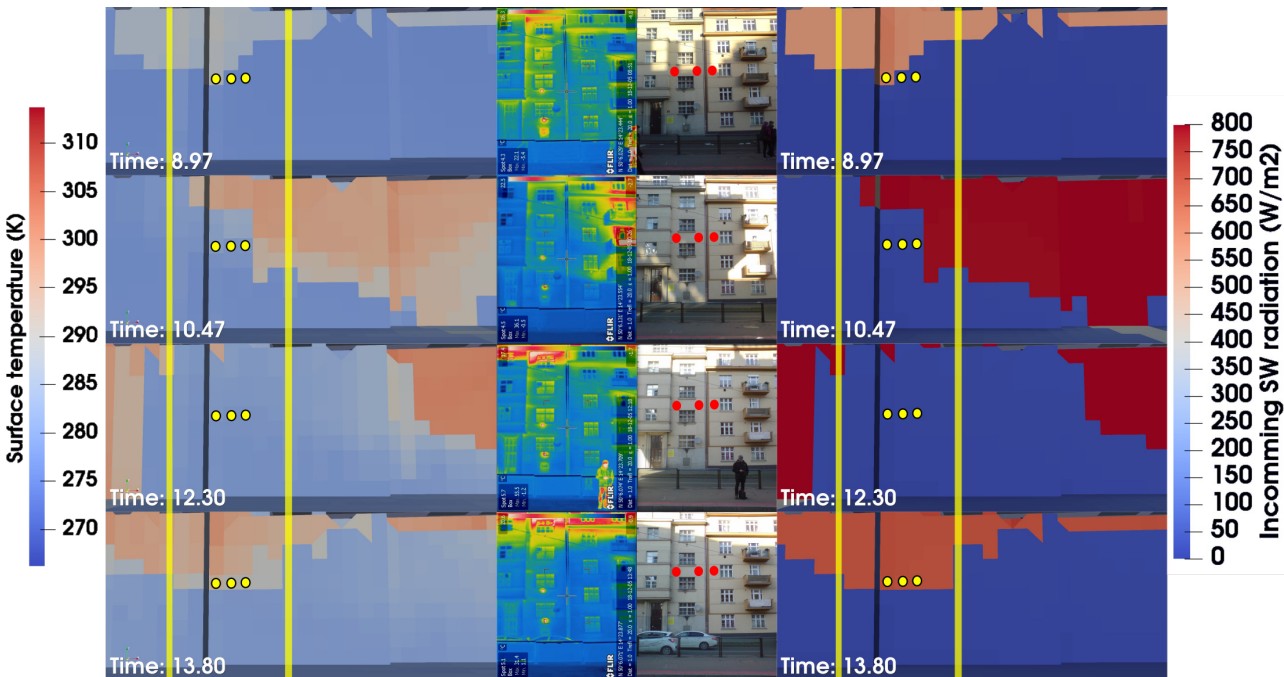

**Figure 18.** Observed camera photos (IR centre left and RGB centre right) on 5 December 2018 at observation times 7:51, 9:26, 11:18, and 12:48 UTC and the modelled counterparts for the closest saved model time-step: surface temperature (left) and incoming SW radiation (right). The yellow dots denote positions of evaluation points 1,2, and 3 (Fig. 11) and the yellow lines show the extent of the area shown on IR and RGB photos. For technical reasons, the step times for the model views express minutes as decimal fractions of the hours.

rapid changes of the temperature gradient in the wall which explains the peaks in the surface and ground heat flux visible in the corresponding graphs in Sect. 5 of supplements. The positive and negative peaks in the ground heat flux correspond to start and end times of irradiation of the given point by direct radiation. This analysis also suggests how complicated a problem is represented by spatially and temporally detailed modelling of radiation energy processes and the surface energy balance in the complex heterogeneous urban environment.

### 5.1.6 Plant canopy effects

Trees and shrubs are modelled in PALM as the resolved plant canopy (PC) which is described by a 3D structure of leaf area density (LAD). Beside affecting the turbulent flow by adding LAD-dependent drag, resolved plant canopy also affects radiative transfer by partially intercepting SW and LW radiation as well as emitting LW radiation (see Krč et al. 2020). Further, the absorbed incoming radiation is transformed into latent and sensible heating terms which are considered within prognostic equations of potential temperature and humidity. Many evaluation points are affected to different degrees by PC. A list of evaluation points where a significant impact of PC can be seen is given in Table S7 in the row "Plant canopy affected surface". In this section, we focus only on the summer scenarios since deciduous trees (which constitute the majority of the trees in the

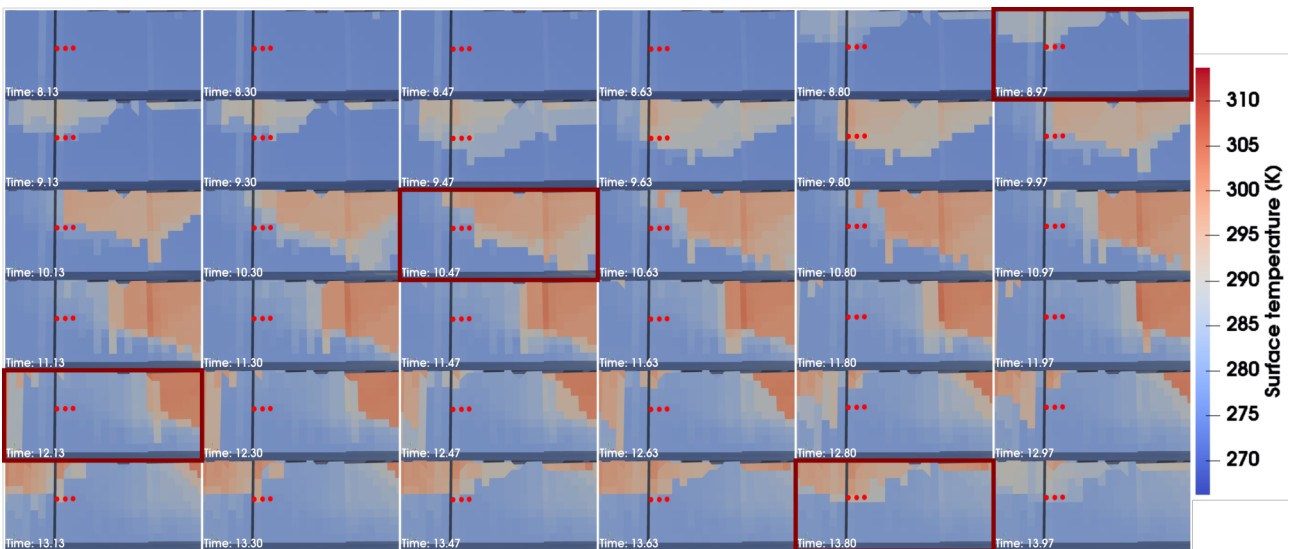

**Figure 19.** Timeline of 10-minute model outputs of wall surface temperature on 5 December 2018 from 5:28 to 12:48 UTC. The time-steps from the previous figure Fig. 18 are highlighted with a red frame, the positions of the evaluation points 1,2, and 3 are marked by red dots. For technical reasons, the step times for the model views express minutes as decimal fractions of the hours.

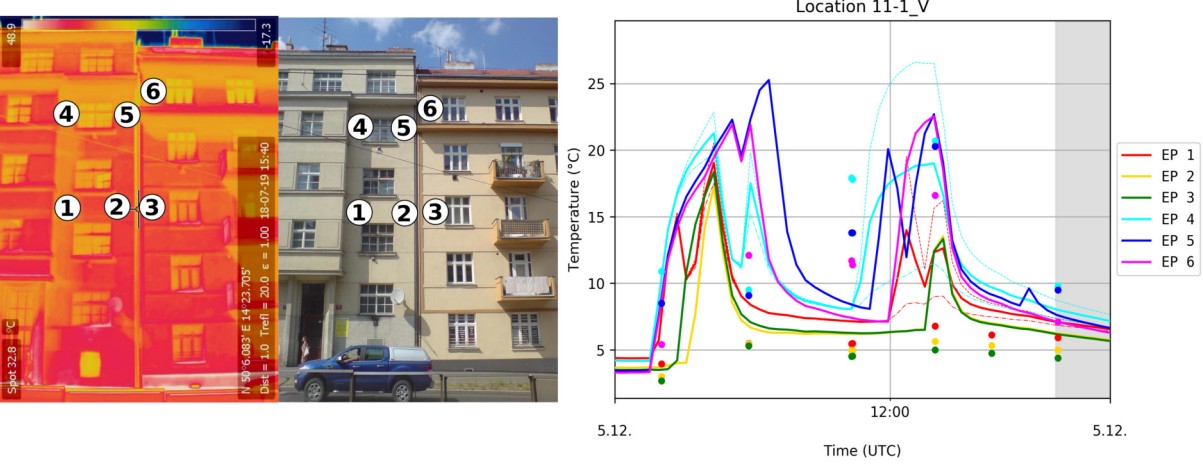

**Figure 20.** Comparison of IR observations and model at location 11-1_V on 5 December 2018 from 7:00 UTC to 16:00 UTC. The left photos show IR and RGB images of the location with marked places of the evaluation points and the right image shows the graph of the modelled (line) and observed (dots) values of the surface temperature for these evaluation points. The grey area denotes the night time.

domain) carry no leaves during the winter. Impact of branches during the winter episodes is roughly modelled as 10 % of the
summer LAD.

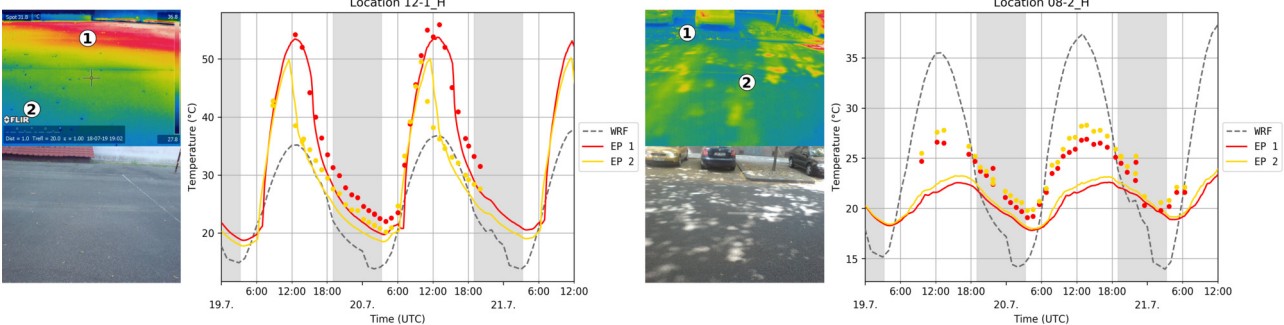

**Figure 21.** IR and RGB photos with locations of evaluation points and the graph of the observed (dots) and modelled (lines) surface temperature for these evaluation points during summer e2 episode (19-21 July 2018). The left half of the figure shows location 12-1_H (the asphalt playground in the courtyard of Sinkule house) and the right part location 08-2_H (asphalt concrete surface in Terronska street). The grey dashed line shows the corresponding WRF skin layer temperature. The grey areas denote the night time.

Figure 21 shows two examples of locations affected by trees (locations 12-1_H and 08-2_H; for full information about these locations see Sect S3 in supplements). Location 12-1_H is on the left, with two evaluation points placed on the same surface (asphalt concrete). The direct radiation at point 2 is influenced by tree shading but the tree shade does not reach evaluation point 1 at this time of year. The shading of the treetop decreases the surface temperature after noon, which is well captured
by the model. A similar situation at location 08-2_H is shown on the right. The evaluation points are similarly placed on an asphalt concrete surface in a street canyon surrounded by two alleys of trees with linked treetops forming an umbrella-like covering. The street surface temperature at location 08-2_H is underestimated by the model by up to 5 °C. Because a similar type of surface is modelled well at 12-1_H and other locations, the most probably explanation for this discrepancy is the tree shading. The reason could be a general overestimation of LAD in the input data and/or a discrepancy in its spatial distribution.
The large tree crowns tend to arrange themselves into clusters with free space between them (see e.g. Mottus, 2006). Figure 21, with spots of direct shortwave radiation passing through the canopy,and location views in Sect S3 of the supplements suggests that this is the case at location 08-2_H. However, the method used for calculation of the LAD distribution within the tree crown does not consider such clusters, leading to possible underestimation of total transmissivity of the whole tree crown. Moreover, PALM uses a constant extinction coefficient for calculation of the optical density from the LAD value, which can lead to
overestimation of optical density if clusters are significant at the subgrid scale. However, this can be mitigated by decreasing the LAD value. These examples confirm the importance of the precise estimate of the structure of the tree LAD in the inputs for the PALM simulations, though gathering of such information presents a complicated task.

### 5.1.7 Discretization issues

PALM discretizes the domain in a Cartesian grid where all values in every grid box are represented by one value. This leads to standard discretization errors. Moreover, the current version of PALM uses the so-called mask method to represent obstacles (terrain, buildings), where a grid box is either 100% fluid or 100% obstacle and consequently any surface is represented by orthogonal grid faces (see. Fig. 4). Besides implications with respect to the near-surface flow dynamics, which can be locally affected, this discretization increases effective roughness and enlarges surface area. The step-like surface representation also modifies the direction of the normal vector and the mutual visibility of the particular grid surfaces, which in turn also affects the surface net radiation and thus the surface energy balance. The observations of the surface temperature allow us to demonstrate a few selected implications for radiative transfer and surface energy balance.

The first observed consequence of the discretization is the fact that the subgrid-size surface features cannot be represented, while in reality, these objects can significantly influence the shading of parts of the surface. This effect can be observed in many of the studied locations (see Sect. S3 in supplements) and it needs to be carefully taken into account when making point comparison of the related surface values.

The effects caused by the step-like surface representation include artificial shading and the alteration of the surface normal vector. Both these effects can be observed and studied in the case of slope terrain as well as in the case of non grid-aligned walls. As an example, let us show the wall around the observation location 07-1 (see Fig. 22; for complete location information see Sect S3 in the supplements). This wall is oriented to the east with a slight inclination to the north. The upper row of the figure shows the observed photo on 20 July 2018 at 9:37 UTC and the 3D view of the modelled incoming SW radiation on this wall at the corresponding modelling time step. The bottom row shows the same situation approximately one hour later at 10:38 UTC. In the first case, all the wall is irradiated by direct solar radiation while the model results indicate artificial shading of some grid faces caused by the step-like representation of the wall. The second case one hour later shows the situation when the wall is shadowed in reality but some of the corresponding model grid faces are irradiated by direct solar radiation due to their slight turn to the east in comparison with the real wall.

Two further consequences of the orthogonally gridded model surfaces are an altered distribution of the reflected radiation and artificial self-reflections owing to the step-like terrain and wall representation. The first effect is difficult to demonstrate in the observed data due to less direct attribution of the reflected radiation to the individual source surfaces and due to the partial masking of reflected radiation by the stronger direct radiation. The second effect can be demonstrated e.g. on the wall around location 07-2_V on 20 July at 11:37 CET (see Fig. 23). The wall is not irradiated in reality by direct solar radiation at this moment as can be seen from the RGB photo. The south facing grids of the model wall ("steps") are illuminated by the direct radiation and the radiation reflected from them then irradiates adjacent grid faces turned to the west (oriented close to the original wall direction), an effect which has no counterpart in reality.

These potential sources of problems especially need to be considered, due to their local nature, when making point-to-point comparisons of modelled and observed quantities. However, when averaging over larger areas, one may expect that these artificial effects to partially mutually compensate due to the unchanged amount of incoming global radiation, which represents

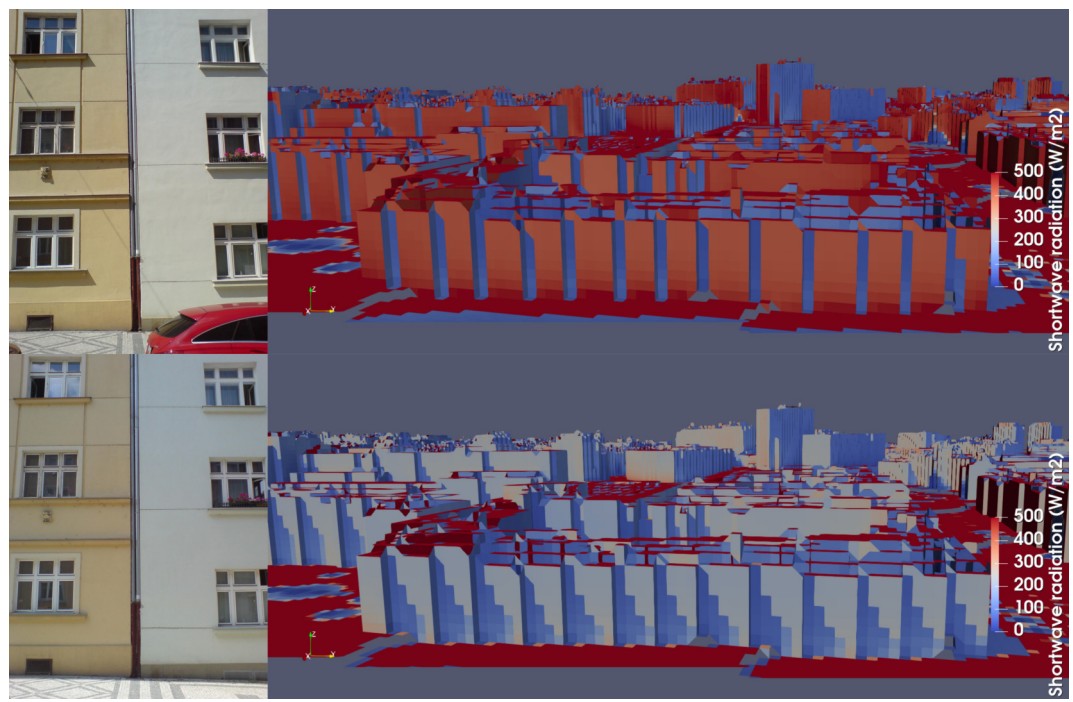

**Figure 22.** East facing wall in N. A. Někrasova Street around location 07-1_V (see Fig. 1 and detail location information in supplements section S3.). The top row shows the observed photo on 20 July 2018 at 9:37 UTC and the 3D view of the modelled incoming SW radiation on this wall at the corresponding time step. The bottom row shows the same situation at 10:38 UTC.

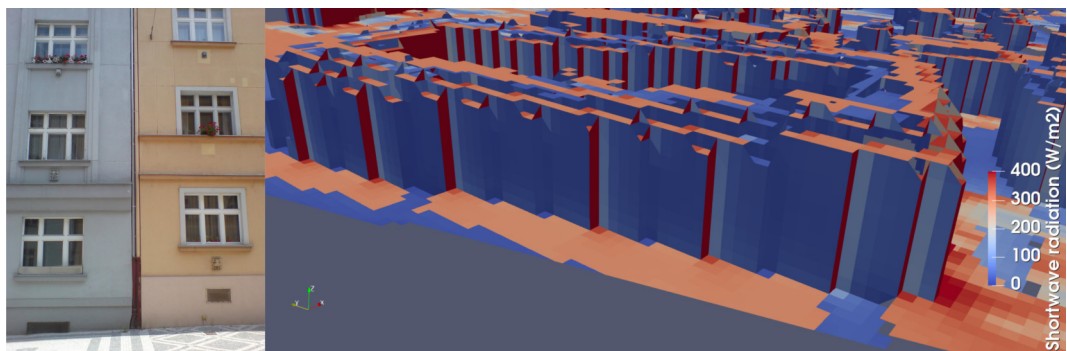

**Figure 23.** West facing wall in N. A. Někrasova Street around location 07-2_V (see Fig. 1 and detail location information in supplements section S3.). The figure shows the observed photo on 20 July 2018 at 10:37 UTC (left) and the 3D view of the modelled incoming SW radiation on this wall at the corresponding modelling time step (right).

the strongest radiative forcing. However, the differences in reflections can still lead to significant changes in the global energy balance of the surface.

To estimate the impact of the discretization on the averaged simulation results of the wall, we ran two idealized simulations of a street canyon. The simulation domain had 2 m grid resolution and it contained one west-east oriented 30 m wide street canyon of height 20 m. The simulated day was 19 July 2018 (the first day of the episode summer e2). The radiation was simulated by the coupled RRTMG model and the meteorological conditions were set constant (west wind 1 m s$^{-1}$, potential temperature at surface 295 K). The simulation started at 3:00 am with a preceding 24 hours spin-up run and covered the 16 sunny hours of the day. The first simulation employed the standard grid with no rotation while the second one had the grid rotated by 45°, utilizing PALM's ability to set grid rotation. This means that the walls of the street canyon were precisely aligned with the grid in the first case while they were represented by steps-like structures in the second case due to the 45° angle they form with the grid. The averaged results of the surface temperature, shortwave irradiation and net radiation over the south facing wall are presented in the supplements in Fig. S22 and S23. The results shows that the differences can reach about 3 °C for surface temperature, over 100 W m$^{-2}$ for shortwave irradiance and about 80 W m$^{-2}$ in the case of net radiation. These effects cannot be simply neglected and further more focused research is needed. Some potential ways to amend the model are discussed in Sect. 6.2.

### 5.2  Wall heat flux

The observations of the wall heat flux (HF) in two locations (see Sect. 2.3.2) allow direct comparison with the wall heat flux simulated by the model. Moreover, the observations of the surface temperature from the sensor allow both validation of the PALM model and the observations obtained by the IR camera (see Sect. 2.3.2).

During the summer campaign, HF observations took place in Sinkule house from 19 July to 3 August and at the Zelená location from 3 to 7 August. This period overlaps only partly with the modelling episode summer e2. The graphs of heat flux and surface temperature are shown in Fig. 24. The sharp rise in observed HF and temperature before 06:00 UTC is caused by the direct irradiation of the sensors by the Sun and the data between around 6 and 8 UTC cannot therefore be taken as valid measurements. (Similar peaks are visible in the PALM outputs before sunset.) The sharp drop of HF on 20 July after 06:00 UTC was caused by the sensor becoming unglued, which was fixed at about 08:00 UTC. The modelled and observed wall heat flux on the ground floor shows a similar diurnal cycle with similar amplitude, though the model slightly overestimates the observed values by about 5 to 10 W m$^{-2}$, while the corresponding modelled surface temperature agrees fairly well with the observations. The modelled wall heat flux on the first floor shows a pronounced diurnal cycle, while the observed wall heat flux shows only a weak diurnal cycle with a significantly smaller amplitude. The modelled surface temperature, however, shows a smaller amplitude with higher nighttime but lower daytime temperatures compared to the observations, which is in agreement with the respective wall heat fluxes where the model increasingly partitions the available energy into the wall heat flux.

The winter HF observations at Sinkule house cover the episode e3 from 4–6 December and the observations at the Zelená location fit with the e2 episode for 27–28 November (see Fig. 25). Even though the modelled surface temperature at Sinkule house for the ground floor observation is overestimated by around 2 °C with respect to the observed one during day time, the modelled and observed wall heat fluxes agree fairly well during the period shown, especially for the first and second day. In contrast, on the first floor the modelled wall heat flux (absolute value) and surface temperature are strongly overestimated,

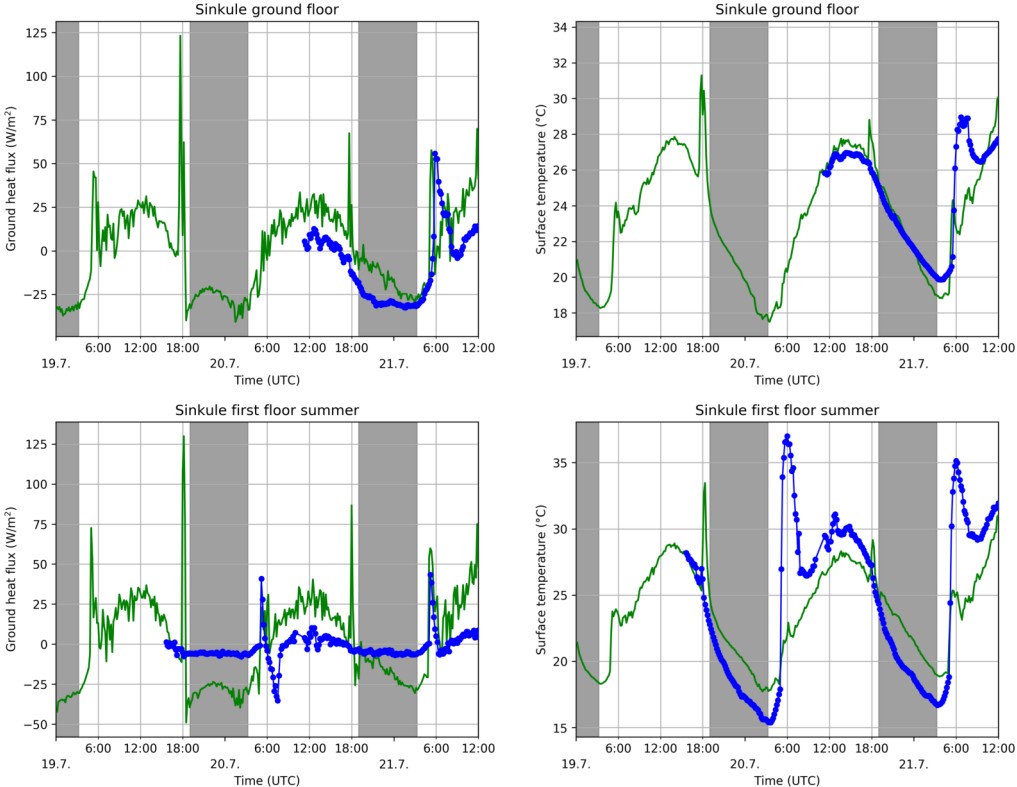

**Figure 24.** Modelled (green) and observed (blue) wall heat flux (left) and surface temperature (right) for days 19–21 July 2018 for location Sinkule house at the ground floor wall (top) and at the first floor wall (bottom). The grey areas denote the night time

especially during the nights. The minimum of the modelled wall heat flux goes down to –50 W m$^{-2}$ during the night from 5 December to 6 December while observations suggest values between –10 and –15 W m$^{-2}$. The situation at the Zelená location is similar; the observed HF fluctuates around –40 W m$^{-2}$ during the nights while the modelled counterpart goes down to –80 W m$^{-2}$. This behaviour suggests that the thermal wall resistance in the case of higher floors of the Sinkule and Zelená buildings are underestimated. Sinkule house is an older building which had been insulated in the past except for the ground floor. The real thermal resistance of this additional insulation layer, which is set in the input data to approximately 6 cm of polystyrene, is probably underestimated and the real insulation is more efficient. The details of the material of wall of Zelená building were not available and some type of construction block was assumed but its thermal conductivity in the model is probably overestimated.

### 5.3 Street canyon meteorological quantities

Data collected by the mobile meteorological stations and vehicles allow us to compare modelled atmospheric quantities against observations within several street canyons. This section presents graphs and statistics of modelled and observed temperature

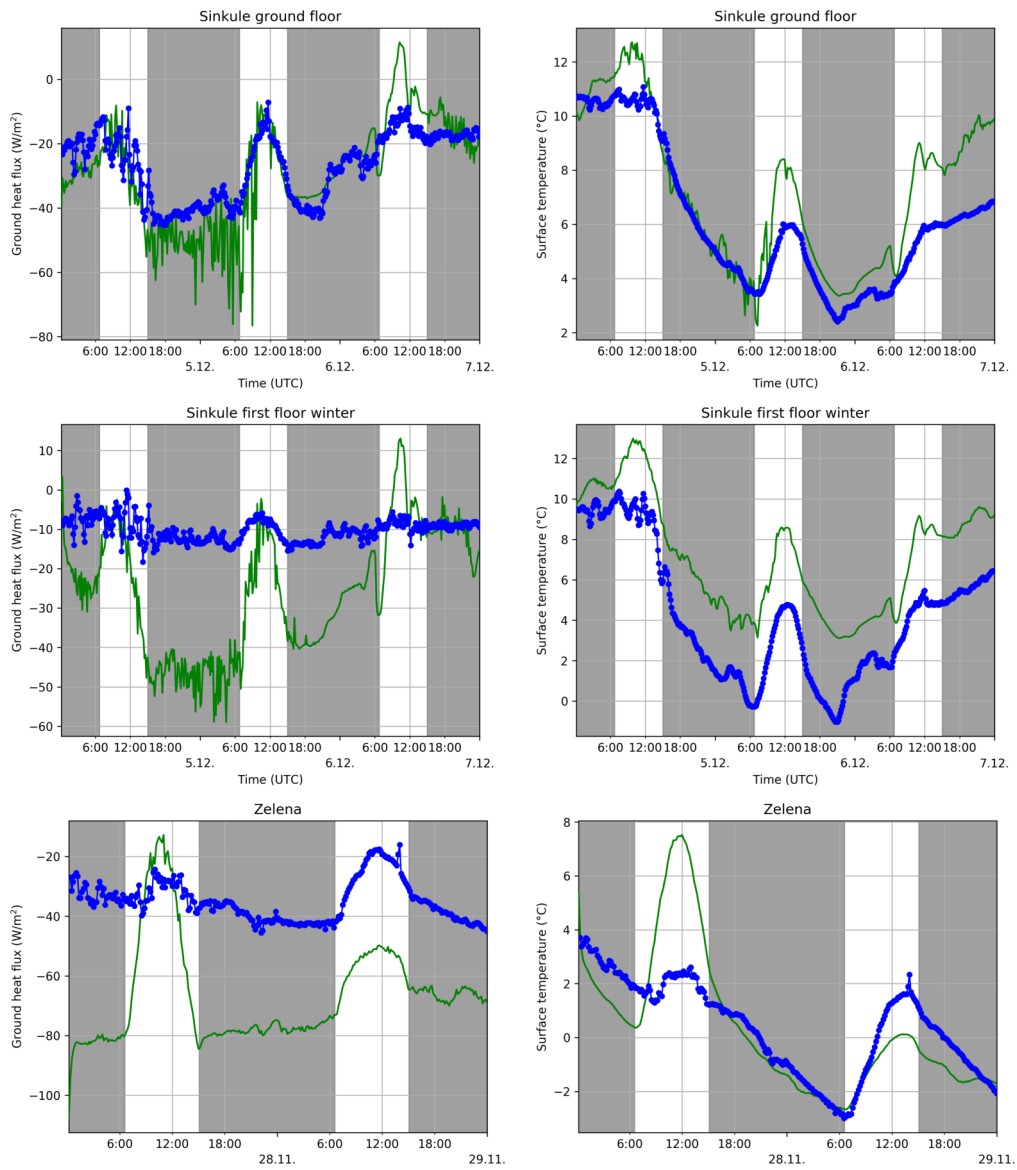

**Figure 25.** Modelled (green) and observed (blue) wall heat flux (left) and surface temperature (right) for 5–6 December for location Sinkule house at the ground floor (top) and at the first floor (middle) and at the Zelená location on 27–28 November (bottom). The grey areas denote the night time.

and wind speed. Graphs are presented for summer e1, summer e2, and winter e3 episodes here; the complete results for all episodes are available in supplements Sect. S5 which also contains corresponding graphs of vertical sensible heat flux and

relative humidity. The comparison graphs contain values from the WRF simulation to allow assessment of the benefits of the micro-scale model.

### 5.3.1 Air temperature

Figure 26 shows timeseries of modelled and observed air temperature within different street-canyons for the summer e1, summer e2, and winter e3 episodes. In the summer scenarios, the daily cycle of air temperature is generally captured by PALM. The modelled maximum air temperature generally agrees well with the observed maximum but is somewhat underestimated, especially at the Sinkule location. The modelled nighttime minimum values tend to be too warm compared to the observation, which is in accordance with the less stable modelled conditions as indicated by Fig. 5. The spatial variability of the shown

model air temperature, as indicated by the red-shaded area, is rather low, suggesting that the comparison of modelled and observed air temperature do not suffer from any location biases. In addition, Fig. 26 also shows one-hourly averaged 2 m air temperature as modelled by WRF and inferred from the WRF-grid point closest to the observations. These values allow us to estimate whether deviations of PALM-modelled values arise from the driving synoptic simulation or from a different source. As WRF was set up without urban parameterization and no buildings were directly considered in the WRF simulation,

a comparison between PALM and WRF with respect to street canyon temperature would thus not be expedient. Similar to the temperature simulated in PALM, the WRF-modelled 2 m air temperature also shows lower daytime maximum temperatures compared to the observations, while at nighttime even lower minimum temperatures are modelled which is in contrast to PALM. This, in turn, suggests that the too warm nighttime temperatures within the street canyon do not arise from the driving mesoscale simulation but for a different reason.

In the winter case, the modelled air temperature reflects the evolution of the observed air temperature, though the air temperature during the first day and the minimum temperature during the first night is overestimated in all street canyons while WRF-modelled temperatures agree well with the observations. Starting from the second night until the end of the simulation, it is striking that the modelled air temperature is significantly overestimated by about 2 to 5 K. This can be attributed to the driving mesoscale WRF simulation which indicates a similar overestimation of air temperature when WRF was not able to cap-

ture nighttime cooling. This nicely shows that the performance of the building-resolving LES strongly depends on the driving mesoscale simulation. If the results on the mesoscale are biased this error will also propagate into the LES.

Statistical metrics for the model performance over all locations and scenarios considered are given in Table 3. For the summer scenarios the street-canyon air temperature is slightly underestimated during daytime, while it is overestimated during nighttime due to insufficient cooling near the surface. For the winter scenarios PALM overestimates the day- and night-time temperatures

by about 1.5 K, which can be partly explained by the driving synoptic conditions. The scatter between observations and model results is about 2 K without any significant difference between day- and night-time or as summer- and winter-time. It is striking that the correlation between modelled and observed air temperature is higher during the daytime where the daily cycle is usually well captured, whereas the correlation is less at nighttime where the nighttime air temperature is often overestimated.

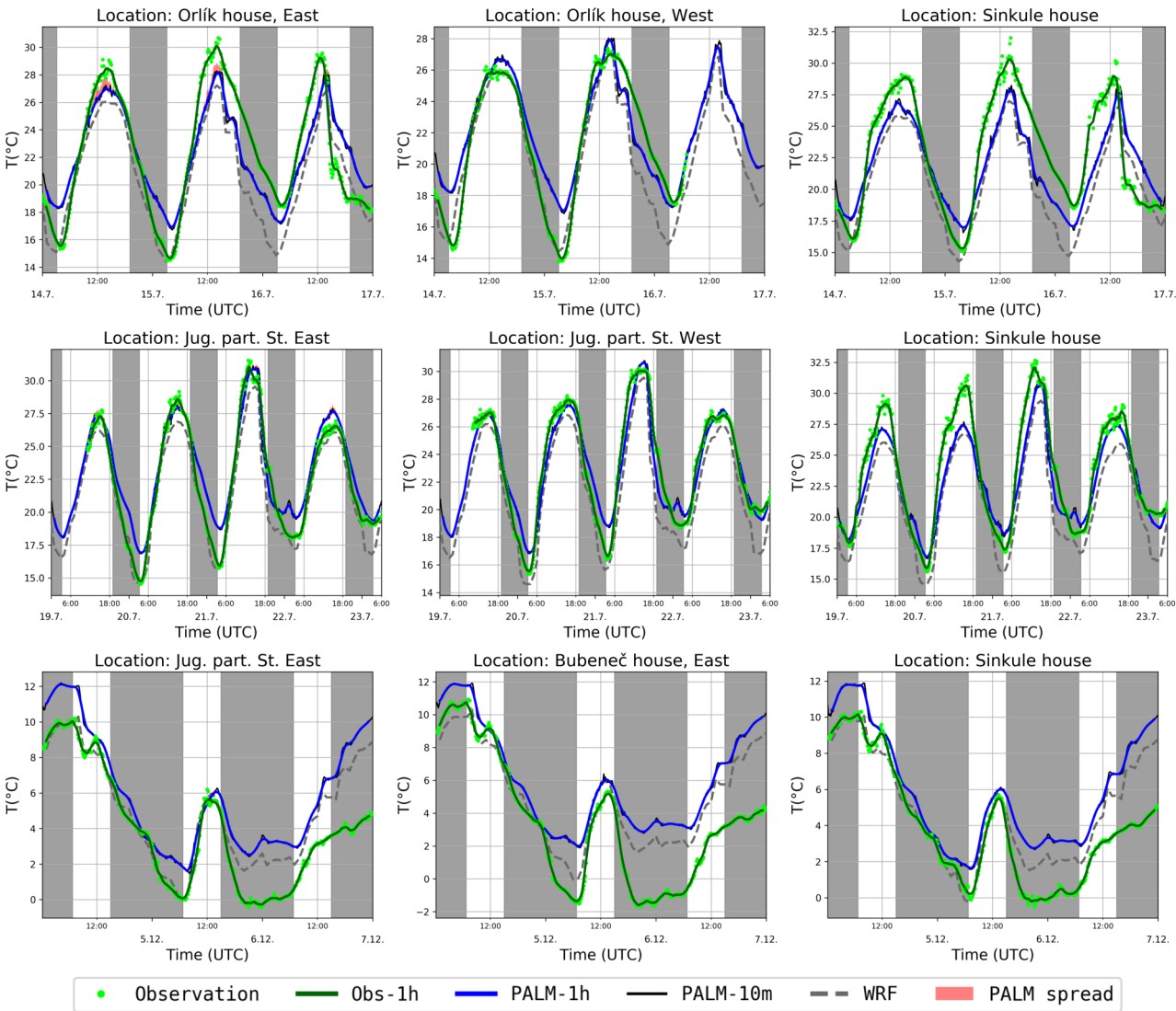

**Figure 26.** Street-canyon temperature at 3.9 m (Sinkule house) and 4.6 m (other) for summer e1 (top row), summer e2 (middle row) and winter e3 (bottom row) episodes. Observations are shown as 10-minute averages (green dots) and moving 1-hour averages (green line). PALM simulations are shown as moving 1-hour averages (blue line), 10-minute averages (solid black line) and the interval between the smallest and the largest 10-minute average among the neighbouring grid points (red band). The grey dashed line denotes the 1-hour averages of temperature at 2 m at the closest WRF grid point. The grey-shaded areas indicate the night time. Please note the black curve indicating the 10-minute average is mostly hidden by the blue curve indicating the one-hour average. The spatial variations in temperature are usually very small, especially in the winter case, meaning that the red curve is hidden most of the time.

**Table 3.** Statistical metrics of modelled one-hour averaged air temperature within different street canyons. The statistics are evaluated over all locations and episodes considered and are partitioned into summer and winter, as well as day- and night-time. The statistical metrics for the modelled 2 m air temperature in WRF are also given for completeness.

| | Summer episodes | | | | Winter episodes | | | |
|---|---|---|---|---|---|---|---|---|
| | Day | | Night | | Day | | Night | |
| | PALM | WRF | PALM | WRF | PALM | WRF | PALM | WRF |
| **N** | 233 | 233 | 122 | 122 | 210 | 210 | 370 | 363 |
| **mean obs** (°C) | 24.1 | 24.1 | 19.3 | 19.3 | 3.5 | 3.5 | 2.4 | 2.4 |
| **mean mod** (°C) | 23.5 | 22.4 | 20.0 | 17.7 | 5.1 | 4.2 | 4.0 | 2.7 |
| **MB** (°C) | -0.6 | -1.7 | 0.7 | -1.6 | 1.6 | 0.7 | 1.6 | 0.3 |
| **RMSE** (°C) | 2.0 | 2.4 | 1.8 | 2.3 | 2.1 | 1.7 | 2.5 | 2.2 |
| **R** | 0.91 | 0.93 | 0.73 | 0.78 | 0.91 | 0.89 | 0.85 | 0.81 |

N = ensemble size; **mean obs** = observed mean value; **mean mod** = modelled mean value; **MB** = mean bias; **RMSE** = root mean square error; **R** = Pearson correlation coefficient.

### 5.3.2 Wind speed

The simulated and observed wind speed in the respective street canyons for episodes summer e1, summer e2, and winter e3 is summarized in Table 4 and plotted in Fig. 27; the complete graphs for all episodes are shown in the supplements in Sect. S5. The graphs also show values simulated by the WRF model to illustrate the added value of the high-resolution LES simulations. Summary metrics for both models and all episodes (Table 4) show similar model performance in summer and winter with only slightly better statistics in summer. Both campaigns exhibit a significant over-estimation. However, all measures show that 885 PALM is partially able to correct biases imposed by its driving boundary conditions.

The wind speed in the summer campaign generally shows good agreement except at the Orlík location, where significantly larger wind speeds are simulated by the model. We hypothesize that this is attributable to the nearby tree crowns in the street, which have a radius of 2 m in the model, but a radius of about 5 m in reality (see corresponding photo in Fig. S7 in the supplements). The uncertainty of the results in this location is also increased by large spatial gradients of the wind speed near 890 the buildings which makes precise fitting of the modelled and observed values sensitive to any spatial inaccuracy.

The daily cycle of the modelled wind speed in the winter scenario is roughly captured at the Sinkule location, except for the nighttime where the PALM-modelled wind speed is generally overestimated as also indicated by Fig. 6. This overestimation of the modelled wind speed, which is also accompanied by increased temporal variability, is also visible at the other stations; this might be linked to the insufficient representation of the stable boundary layer. Also the daytime values are mostly overesti-895 mated but this overestimation is much lower than that during nights. The overestimation in general could also be linked to the inaccuracies in the boundary conditions from WRF which overestimates near-surface wind speed, which is expected when not using an urban paramaterization, see e.g. Halenka et al., 2019, while also at higher levels the wind speed is partly overestimated (see Fig. 6).

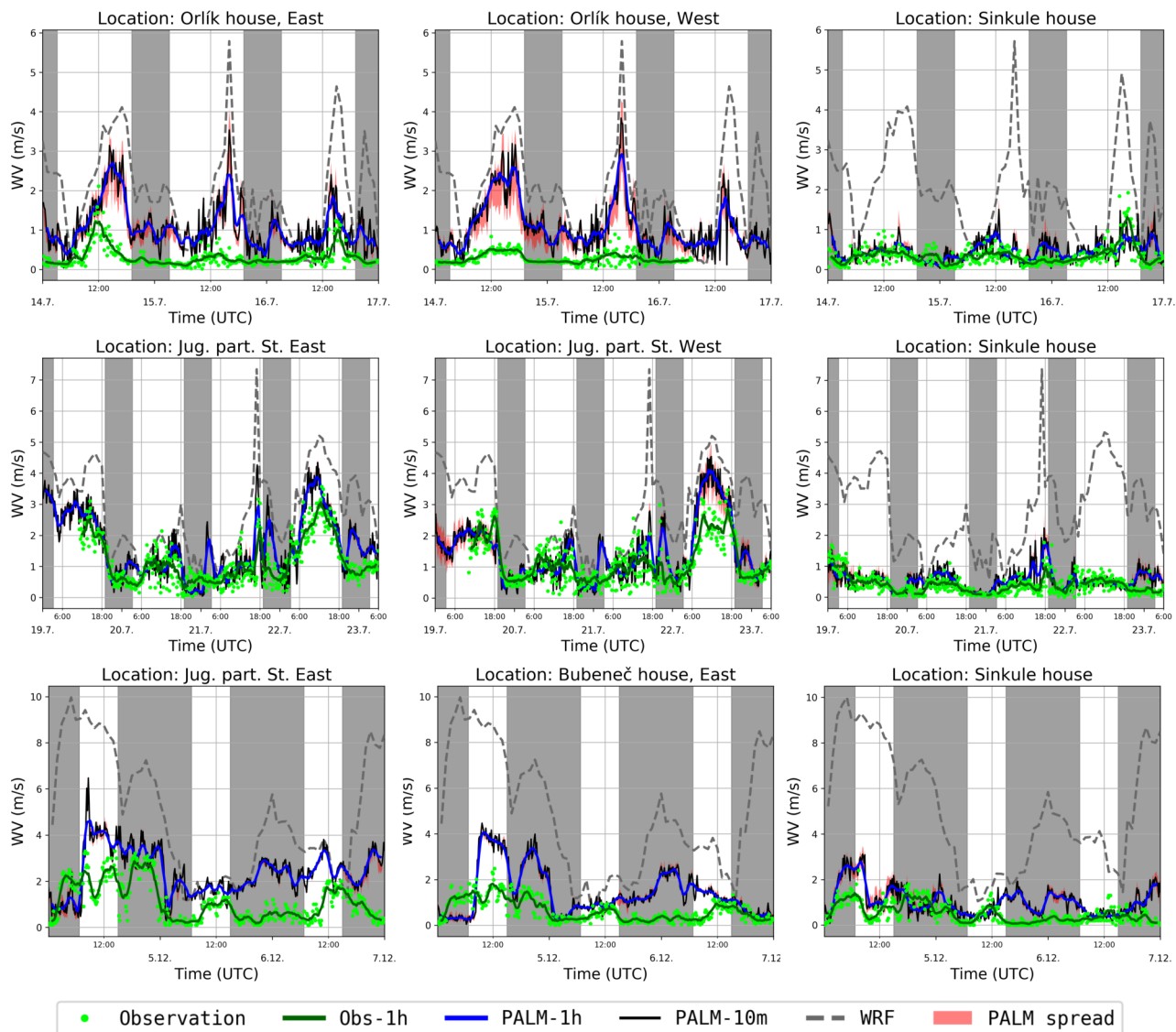

**Figure 27.** Street-canyon wind speed at 10 m (Sinkule house) and 6.8 m (other) for summer e1 (top row), summer e2 (middle row), and winter e3 (bottom row) episodes. Observations are shown as 10-minute averages (green dots) and moving 1-hour averages (green line). PALM simulations are shown as moving 1-hour averages (blue line), 10-minute averages (solid black line) and the interval between the smallest and the largest 10-minute average among neighbouring grid points (red band). The grey dashed line denotes 1-hour averages at 10 m at the closest WRF grid point. The grey-shaded areas indicate the night time.

**Table 4.** Statistical metrics of modelled one-hourly averaged wind velocities within different street canyons. The statistics are evaluated over all locations and episodes considered. Summer and winter episodes are distinguished. The statistical metrics for the modelled 10 m wind speed in WRF are also given for completeness.

| | Summer episodes | | Winter episodes | | All episodes | |
|---|---|---|---|---|---|---|
| | PALM | WRF | PALM | WRF | PALM | WRF |
| N | 354 | 354 | 580 | 573 | 934 | 927 |
| mean obs $(\mathrm{m\,s^{-1}})$ | 0.5 | 0.5 | 0.6 | 0.6 | 0.5 | 0.5 |
| mean mod $(\mathrm{m\,s^{-1}})$ | 0.9 | 2.0 | 1.1 | 3.5 | 1.0 | 2.9 |
| FB | 0.5 | 1.2 | 0.6 | 1.4 | 0.6 | 1.4 |
| NMSE | 1.0 | 3.4 | 1.3 | 6.6 | 1.2 | 5.9 |
| R | 0.50 | 0.38 | 0.55 | 0.45 | 0.53 | 0.42 |

N = ensemble size; **mean obs** = observed mean value; **mean mod** = modelled mean value; **FB** = fractional bias; **NMSE** = normalized mean square error; **R** = Pearson correlation coefficient.

### 5.3.3 Wind speed on the roof

To assess model behaviour in the urban canopy outside the street canyon, a comparison of the wind speed measured at the roof of the highest building in the child LES domain (FSv - Faculty of Civil Engineering CTU) with PALM is presented. In order to illustrate the added value of the high-resolution LES simulations, outputs of the WRF are provided as well together with measurements from the nearest synoptic station Praha-Ruzyně for reference (reliable wind direction measurements were only available from the synoptic station). The graphs for summer episode e2 and winter e3 are in Figure 28. The time series for
episodes summer e1, winter e1, and e2 are presented in the supplements in Fig. S21. Summary metrics for all episodes are in Table 5. The wind speed is generally overestimated, with smaller errors in the summer simulations, a difference already present in the driving WRF simulation. In a comparison of the two models, PALM shows better agreement with observations with the exception of the correlation coefficient, which is similar in summer and even higher for WRF results in winter. For most of the episode, the PALM simulated wind speed is closer to the FSv observations than the WRF results as well as the background
Praha-Ruzyně observations. During the winter e3 episode, the differences are considerable. In particular, there is a large peak in the evening of 6 December, which confirms the disagreement of the wind profiles in Fig. 6.

### 5.4 Street canyon air quality

This section presents a comparison of modelled and observed concentrations of $NO_X$ and $PM_{10}$. The simulated and measured concentrations of $NO_X$ in the summer e1, e2 and the winter e3 episodes are shown in Figure 29. The complete graphs for
$NO_X$, $PM_{10}$, and $PM_{2.5}$ for all episodes can be found in the supplements in Sect. S5. Summary statistics for $NO_X$ 1 hour average concentrations for aggregated summer and winter episodes are presented in Tables 6 and 7. Statistics were calculated separately for street canyon locations influenced directly by the traffic and for the courtyard of Sinkule house, which, with

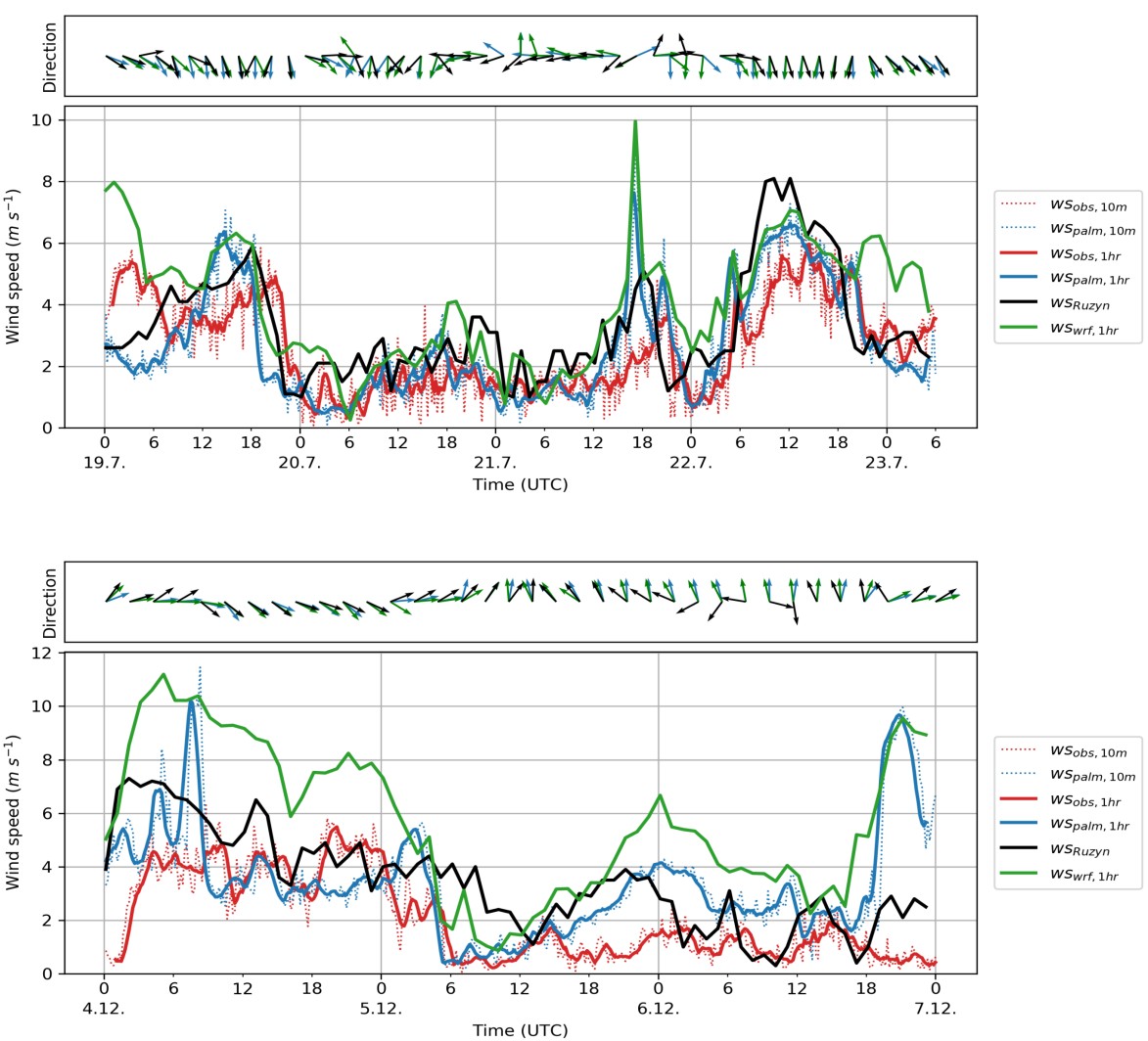

**Figure 28.** Time series of wind speed and wind direction at the roof of the tallest building of the Faculty of Civil Engineering of the Czech Technical University for summer episode e2 (left) and winter episode e3 (right). The graphs show wind speed and the boxes of arrows wind direction. The red colour represents the observations, the blue colour the PALM modelled values, green colour values from the WRF model, and the black line the values from the nearest synoptic station at Praha-Ruzyně. Thin dotted lines represent 10-minute averages and the thick solid lines 1-hour moving averages of wind speed. The arrows represent 2-hour averages of wind direction. PALM model results are taken from the child domain with 2 m horizontal resolution.

respect to traffic, represents an urban background. Similar summary statistics for $PM_{10}$ are presented in the supplements (Table S8 and S9).

**Table 5.** Comparison of 1-h average wind speed measured on the rooftop of FSv with WRF and PALM results for the same location.

| | Summer episodes | | Winter episodes | | All episodes | |
|---|---|---|---|---|---|---|
| | **PALM** | **WRF** | **PALM** | **WRF** | **PALM** | **WRF** |
| **N** | 176 | 172 | 219 | 213 | 395 | 385 |
| **mean obs** $(\mathrm{m\,s^{-1}})$ | 2.3 | 2.3 | 1.7 | 1.7 | 2.0 | 2.0 |
| **mean mod** $(\mathrm{m\,s^{-1}})$ | 2.5 | 3.5 | 2.6 | 4.1 | 2.5 | 3.8 |
| **FB** | 0.07 | 0.41 | 0.43 | 0.85 | 0.26 | 0.65 |
| **NMSE** | 0.34 | 0.47 | 0.75 | 1.47 | 0.54 | 0.97 |
| **R** | 0.61 | 0.60 | 0.43 | 0.59 | 0.49 | 0.52 |

N = ensemble size; **mean obs** = observed mean value; **mean mod** = modelled mean value; **FB** = fractional bias; **NMSE** = normalized mean square error; **R** = Pearson correlation coefficient.

PALM coupled with a driving mesoscale model has a potential to represent both the magnitude and the temporal evolution of street level $NO_X$ concentrations and thus eliminate the underprediction of the mesoscale model. This is especially true for different types of street canyons, but it is also important to mention that the differences between urban background and street canyon locations are captured well. Variability of PALM 1-hour average $NO_X$ concentrations expressed as a standard deviation is about 50 % larger than that of observed data in summer episodes for both street canyon and background locations. In winter episodes the situation is opposite. When we check the large PALM overpredictions (e.g. 15 July after sunset, 21 July in the morning, or 25 November after sunrise), these all happen, almost exclusively, when the driving CAMx model gives values within the range of, or even largely overestimates, the observations. Similarly, situations when PALM underpredicts $NO_X$ concentrations happen when the increase in observed values is not reflected by the driving model as is the case for the 2nd half of the winter e2 episode. As can be seen from Fig. S17, a strong surface temperature inversion on 28 November at 00:00 UTC and especially 06:00 UTC is not captured by the WRF which in turn impacts PALM meteorology (which at least partially reflects the observed inversion) and boundary concentrations.

It is also evident that the simulated $NO_X$ concentrations are closer to the measurements in the summer episodes, especially in the street canyon locations. However, a high resolution modelling of concentrations in winter is more challenging due to local heating and the associated uncertainties of the emissions. The strong simulated peak in the morning of 25 November, which is also present in the CAMx results, does not appear to be present in the measurements at all. A detailed examination of the concentration fields revealed a strong impact of local heating sources and also effects from the boundary conditions.

For $PM_{10}$ PALM overpredicts observations during winter episodes and also the variability of its outputs is ca. 50 % larger than in observed data, the complete opposite of the case for $NO_X$.

The PALM metrics for $NO_X$ and $PM_{10}$ computed from all available 1-hour concentration averages at all points where measurements were available (not shown) fulfil the criteria for dispersion models as suggested by Chang and Hanna (2004). Although these criteria were developed for simpler models, they are applied to a more complex problem here and are good indicators of fitness for purpose. More specifically, the absolute value of fractional bias is less then 0.3, the fraction of pre-

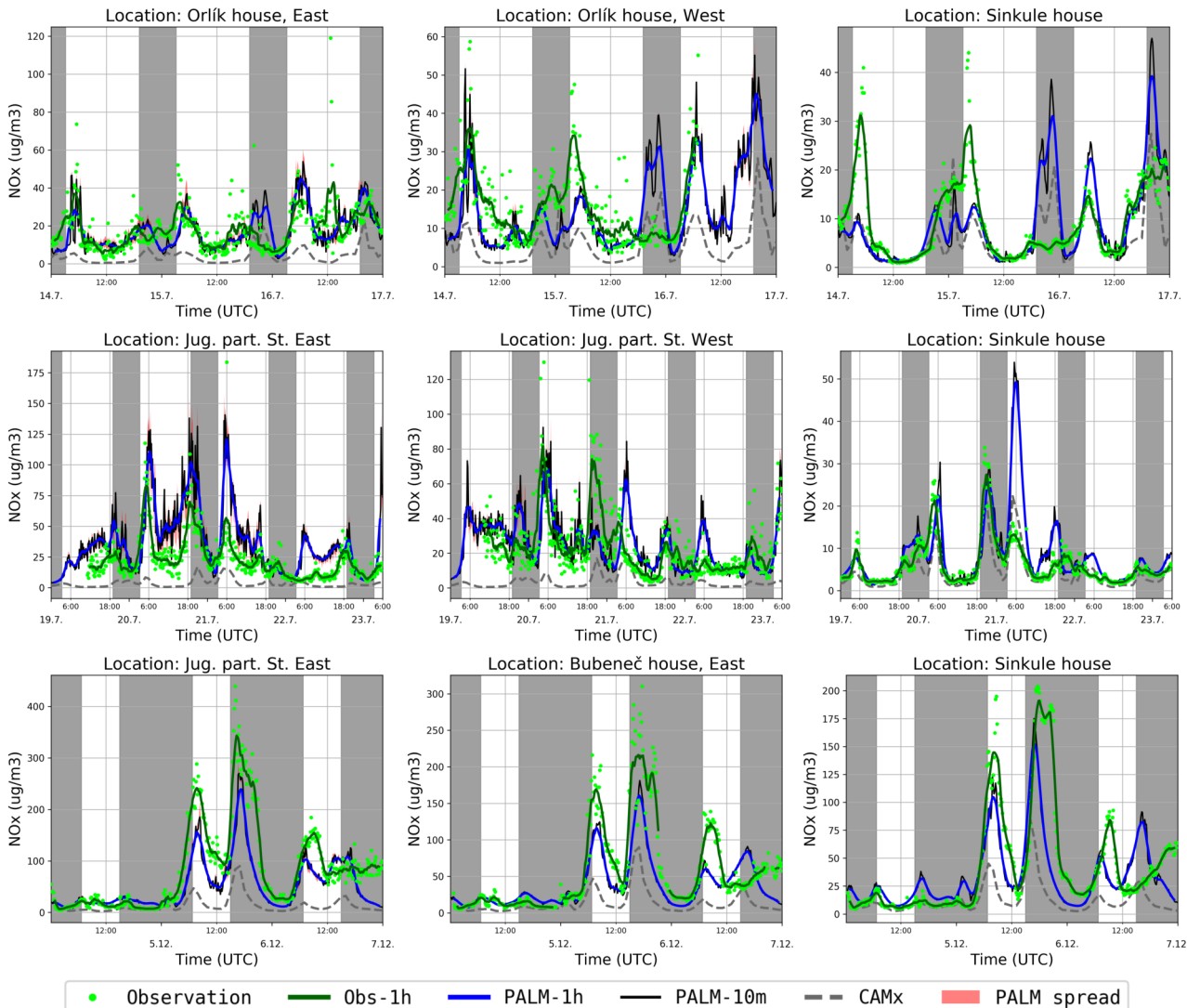

**Figure 29.** Street-canyon $NO_X$ concentrations in 3.9 m (Sinkule house) and 4.6 m (other locations) for summer e1 (top row), summer e2 (middle row) and winter e3 (bottom row) episodes. Observations are shown as 10-minute averages (green dots) and moving 1-hour averages (green line). PALM simulations are shown as moving 1-hour averages (blue line), 10-minute averages (black line) and the interval between the smallest and the largest 10-minute average among the neighbouring grid points (red band). The grey dashed line denotes CAMx 1-hour concentration for the lowest level (lowest 50 m above ground) at the closest CAMx grid point. The grey-shaded areas indicate the night time.

dictions within a factor of two of the observations is more than 50 %, and the random scatter expressed as geometric variance (VG; not shown in tables) is within a factor of two of the mean (i.e. VG < 1.6). These criteria are also fulfilled for data split into summer / winter episodes and street canyon / background locations with the following exceptions: VG is 1.8 for winter background $NO_X$ and no criteria are fulfilled for summer background $PM_{10}$.

**Table 6.** Comparison of 1-h average NO$_X$ concentrations measured in the street canyons with CAMx and PALM results for the same location.

| | Summer episodes | | Winter episodes | | All episodes | |
|---|---|---|---|---|---|---|
| | PALM | CAMx | PALM | CAMx | PALM | CAMx |
| **N** | 224 | 224 | 363 | 360 | 587 | 584 |
| **mean obs** ($\mu$g m$^{-3}$) | 22.6 | 22.6 | 54.5 | 54.7 | 42.3 | 42.4 |
| **mean mod** ($\mu$g m$^{-3}$) | 26.2 | 4.6 | 42.1 | 13.9 | 36.0 | 10.4 |
| **standard deviation obs** ($\mu$g m$^{-3}$) | 14.9 | 14.9 | 56.1 | 56.3 | 47.7 | 47.8 |
| **standard deviation mod** ($\mu$g m$^{-3}$) | 21.8 | 4.5 | 33.4 | 13.8 | 30.5 | 12.1 |
| **FB** | 0.1 | -1.3 | -0.3 | -1.2 | -0.2 | -1.2 |
| **NMSE** | 0.5 | 5.0 | 0.8 | 5.5 | 0.8 | 6.4 |
| **FAC2** | 0.70 | 0.09 | 0.67 | 0.20 | 0.68 | 0.16 |
| **R** | 0.62 | 0.29 | 0.70 | 0.52 | 0.70 | 0.57 |

N = ensemble size; **obs** = observed concentration; **mod** = modelled value; **FB** = fractional bias; **NMSE** = normalized mean square error; **FAC2** = fraction of predictions within a factor of two of the observations; **R** = Pearson correlation coefficient.

**Table 7.** Comparison of 1-h average NO$_X$ concentrations measured in the Sinkule yard with CAMx and PALM results for the same location.

| | Summer episodes | | Winter episodes | | All episodes | |
|---|---|---|---|---|---|---|
| | PALM | CAMx | PALM | CAMx | PALM | CAMx |
| **N** | 130 | 130 | 200 | 197 | 330 | 327 |
| **mean obs** ($\mu$g m$^{-3}$) | 8.6 | 8.6 | 33.9 | 34.2 | 23.9 | 24.0 |
| **mean mod** ($\mu$g m$^{-3}$) | 9.6 | 5.7 | 35.5 | 13.6 | 25.3 | 10.5 |
| **standard deviation obs** ($\mu$g m$^{-3}$) | 7.1 | 7.1 | 39.1 | 39.3 | 33.1 | 33.2 |
| **standard deviation mod** ($\mu$g m$^{-3}$) | 9.5 | 5.5 | 29.5 | 12.7 | 26.9 | 11.1 |
| **FB** | 0.1 | -0.4 | 0.0 | -0.9 | 0.1 | -0.8 |
| **NMSE** | 0.9 | 0.8 | 1.0 | 3.7 | 1.2 | 4.2 |
| **FAC2** | 0.78 | 0.60 | 0.66 | 0.49 | 0.71 | 0.53 |
| **R** | 0.50 | 0.62 | 0.54 | 0.39 | 0.61 | 0.47 |

N = ensemble size; **obs** = observed concentration; **mod** = modelled value; **FB** = fractional bias; **NMSE** = normalized mean square error; **FAC2** = fraction of predictions within a factor of two of the observations; **R** = Pearson correlation coefficient.

In addition to the stationary measurements, mobile observations of the air quality indicators were performed (see Sect. 2.3.4 for details). Fig. 30 shows graphs comparing observed values of $NO_X$ with modelled values in grid boxes corresponding to the position of the mobile instruments. For comparison of $PM_{10}$ see Sect. S5 in the supplements. The observed $NO_X$ values show quite high variability within the short timeframe of the measurements in many locations (variability between 20–160 µg m$^{-3}$). On the other hand, the oscillations are very small during some other measurements (e.g. loc. 6–17 on 19 July and partly loc. 13 on 4 December). This high variability of some measured values suggests impact of a very close local emission source (e.g. buses at bus stations or local heating) but this cannot be verified with the data available. Moreover, these oscillations are not present in the $PM_{10}$ observations, which supports the hypothesis of local $NO_X$ sources in contrast to dynamical causes.

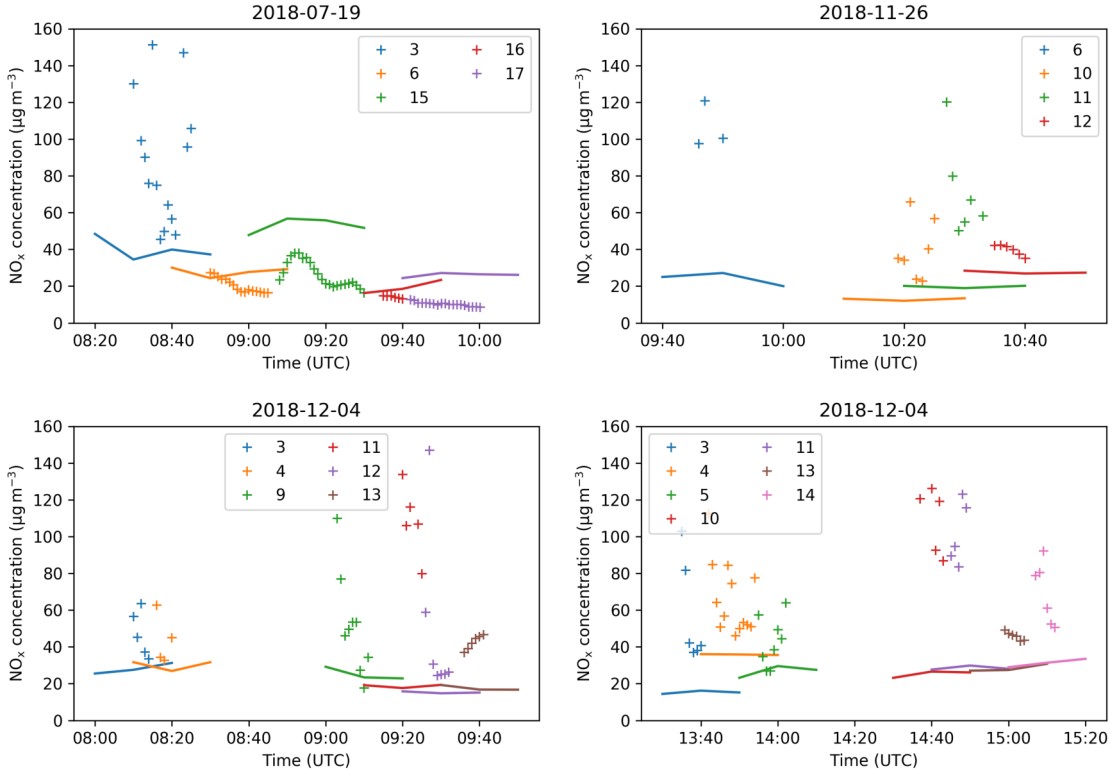

**Figure 30.** Mobile $NO_X$ measurements (+ marker) and modelled concentrations (solid) for 19 July morning (top left), 26 November morning (top right), 4 December morning (bottom left) and 4 December afternoon (bottom right). Numbers refer to mobile measurement locations according to Fig. 1

.

In the winter episode, $NO_X$ observations show much higher variability than in the summer episode. During the morning series, modelled values correspond quite well with observations for the most part with the exception of point 11 where the

model is 2–5 times lower. The afternoon series shows good agreement at points 2, 3, 4, 13 and 14. At points 10 and 11 the model results are again consistently lower than observations.

## 6 Summary and conclusions

### 6.1 Summary of the results


In this study, PALM LES simulations driven by mesoscale WRF and CAMx simulations were performed for a real urban environment in Prague-Dejvice, Czech Republic. Modelled meteorological, air quality, and surface quantities were compared against in-situ measurements taken during a specially designed observation campaign.

The PALM model properly adjusts to the temporally evolving WRF and CAMx conditions and simulates the temporal

evolution and daily amplitude of street-canyon quantities in most cases, with some noticeable exceptions such as insufficient nighttime cooling in some conditions. However, correct results depend on proper driving conditions as well as on the correct setting of the urban canopy properties in the model and the spatial and temporal distribution of emissions. The importance of the urban canopy properties was demonstrated in many particular cases. It was also shown that the driving WRF model does not perfectly reproduce the observations, resulting in discrepancies in the wind speed and potential temperature propagating

into the PALM solution via the boundary conditions given by WRF.

Concentrations of $NO_X$ were modelled well in some situations and PALM properly adds a local air pollution increment to the urban background values provided by the CAMx simulation, while for some places and times (mainly around sunset or sunrise) the model overestimates the concentrations of $NO_X$. This is probably related to atmospheric stability and uncertainties in modelling stably stratified turbulent flow. The opposite situation (i.e. the underestimation of $NO_X$) occurs less often. These

discrepancies could be partially attributed to uncertainties of the emissions and imperfection in boundary conditions provided by WRF and CAMx, though another probable cause is PALM misrepresenting the turbulent flow under some meteorological conditions. This issue needs further investigation. $PM_{10}$ concentrations were modelled less accurately than $NO_X$, which can be mainly attributed to the driving model and overestimated emissions of resuspended dust.

The modelled surface temperature agrees reasonably well with the observed one at most of the surface evaluation points.

However, it is striking that the agreement is usually better for the summer episodes when strong radiative forcing exists than for the winter episodes when the model results are more prone to uncertain specification of material properties as well as inaccuracies in atmospheric conditions from the driving mesoscale model. The surface temperature at pavement surfaces and at wall surfaces belonging to traditional buildings based on bricks or building blocks is usually modelled well, while the surface temperature at modern buildings with multi-layer prefabricated walls is captured less accurately. At low-vegetated

ground surfaces the modelled surface temperature agrees also well with the observation, even though we note that the model results strongly depend on a proper description of initial soil moisture, and probably other soil parameters. Beside an accurate prescription of surface-material parameters, an accurate representation of the LAD is also essential for accurate modelling of the local atmosphere-surface exchange. Even though this study contains some indicative sensitivity investigations for the

studied domain and episodes, we note that a systematic sensitivity study on the model input parameters is out of the scope of
this paper and the reader is here referred to Belda et al. (2020).

## 6.2 Lessons learned and outlook for future improvements

This study also points towards particular aspects in the model, its configuration, the input data preparation, and the observation strategy that deserve particular focus in the future.

The current version of the PALM input standard (PIDS) and implementation of BSM allows discretization of the walls into four layers, independent of the thickness and the structure of the real wall, meaning that the grid resolution of the wall layers may differ among different wall surfaces. Further, wall material properties for complex walls with multiple layers are sometimes not well captured by only four wall layers, leading to under- or over-estimation of the thickness of the insulating layer, among other discrepancies. A variable number of wall layers would allow more realistic representation of wall material properties. Moreover, pre-specified typical structures of complex wall compositions in BSM would simplify proper initialization of these walls.

The current method of discretization of terrain and buildings in PALM is bound to the Cartesian model grid, which means that the entire volume of each grid cell contains either atmosphere or obstacle. If the modelled domain contains uneven terrain, sloped roofs, or walls that are not parallel to the grid axes, the discretization creates artificial steps which affect radiative fluxes as well as the airflow. Such step-like surfaces on facades create both artificially shaded and artificially sunlit surfaces which also affects the energy balance of the facade. Even though these effects are strongest locally, they can also bias the aggregated values for larger surface areas. A major change of discretization is planned for future versions of the PALM model – surfaces will be represented using the Immersed Boundary Method (see Peskin, 1972). This method allows representation of surfaces with arbitrary orientation, thus negating creation of artificial steps.

In the current version of the radiative transfer model (RTM), all surfaces are considered as Lambertian reflectors, meaning that directional reflection at windows or polished materials cannot be considered, even though such reflection can be found at almost every facade. This in turn adds uncertainty to the surface net radiation and thus to the energy balance at the surrounding surfaces. Implementation of specular reflection is planned to better simulate the radiative transfer at glass and polished surfaces.

The analysis of air and surface temperatures revealed insufficient nocturnal air cooling in certain meteorological conditions where the stratification is not captured properly by the model. In this study, the incoming radiation is explicitly prescribed, while radiative cooling of the air volume itself is not considered. Hence, in order to check how sensitive the model results are to this, test simulations where we applied the RRTMG radiation scheme and where radiative cooling of the air volume is considered were run; however, we observed a similar insufficient cooling in this case. This insufficient nocturnal cooling requires further future investigation.

Another implication arises from the mesoscale nesting approach. The analysis of the wind speeds at higher levels, and of temperatures, revealed that PALM partly reflects the conditions simulated by the mesoscale model (WRF), especially during wintertime. The error made on the mesoscale is thus propagated into the LES, biasing its simulation results. To minimize this mesoscale forcing bias on the LES results, the driving mesoscale conditions might be further combined with additional

nudging terms inferred from observations, continuously nudging the imposed boundary conditions for the LES towards the observations.

The study suggests strong sensitivity of the results to accuracy of input data, such as the wall-material properties and the structure of tree crowns. The sensitivity of PALM to material parameters is more systematically investigated in Belda et al. (2020). Bulk parameters prescribed for certain building categories might strongly deviate from the actual conditions at the building. Hence, usage of bulk input parameters might significantly modify the simulation results locally. Other detailed observations are needed to improve properties of the categories of wall, roofs, and pavement materials. The study also stresses

the need for correct setting of the initial soil moisture for low vegetation surfaces.

The experimental campaign also serves as a source of useful experience for future studies of similar type. Modern buildings with high amounts of glass and other reflective exterior surfaces proved to be challenging for surface temperature measurements using an IR camera. The reflections often obscure the emitted thermal radiation from the surface and thus the IR camera does not provide a reliable way to observe surface temperature for such surfaces.

Data from mobile measurement vehicles proved to be difficult to interpret and difficult to draw statistically relevant conclusions from due to the influence of the strong local temporally and spatially evolving emissions, which are difficult to simulate in the emission model. In future, either a significantly higher number of measurements would be required or the effort should be concentrated elsewhere. One direction for consideration is a combination of traditionally comprehensive vehicle-observation stations with a wider network of more limited sensors.

Though drones at first sight offer another promising direction, drone measurements in a city are unfortunately limited by various restrictions imposed by the air traffic control and land owners. The entire city of Prague is located in controlled airspace starting at ground level and including our area of interest. Other requirements for useful drone observations are matching height and speed changes to instrumentation characteristics, such as relaxation time. Preparatory test flights in consultation with the drone operator may be necessary. Regular balloon soundings from the Praha-Libuš station proved to be indispensable. In

future, increasing the frequency of measurements during a measurement campaign would be very useful and the possibility of dedicated soundings in the area of interest should be considered. However, this is also limited by restrictions similar to those on drone observations.

In summary, the ability of PALM to represent reality to a reasonable degree depends not just on the representation of physical processes in the model itself, but on input-data quality and the accuracy of the mesoscale forcing. For future studies it is thus a

valid question where the focus should lie; should it be on further improving the model to better reflect physical processes in the urban boundary layer, or on obtaining as accurate and detailed input data as possible. In the authors' opinion, however, these options are not mutually exclusive but have to be balanced against each other. Focusing mainly on the input data will sooner or later result in a situation where the model performance is constrained by an insufficient representation of the physics, and a model with perfect physical processes will still need very good and detailed input data to produce practically relevant output.

The task of attributing relative importance of these sources of uncertainty has been extensively tested in the field of numerical weather prediction and climate modelling in a number of coordinated projects producing large ensembles of simulations, e.g. the currently ongoing CMIP6 (Eyring et al., 2016) and CORDEX (Giorgi et al., 2009; Gutowski et al., 2016). In our case, a

similar approach of employing different models and model setups, and testing their respective sensitivity to input data would allow assessment of the sources of uncertainty. However, due to the enormous computational resources required for these kinds of simulations, such an endeavour is not feasible for one modelling team and it would benefit from the kind of framework of coordinated experiments that are a norm in the climate modelling community.

*Code and data availability.* The PALM modeling system is freely available from http://palm-model.org (last access: 29 May 2020) and distributed under the GNU General Public Licence v3 (http://www.gnu.org/copyleft/gpl.html, last access: 29 May 2020). The model source code version 6.0 in revision r4508 used in this article is also available via https://doi.org/10.25835/0073713 (Resler et al. , 2020a). The configurations and inputs of the model for all simulated episodes are available via http://hdl.handle.net/11104/0315416 (Resler et al. , 2020b).

## Appendix A: Statistical measures used in manuscript

Apart from means and standard deviations of observed and modelled values, the following normalised statistics are used to summarise model performance. Please note that we adopted the convention that bias is positive when the model overestimates observations.

factor of two (FAC2): fraction of predictions within a factor of two of the observations

fractional bias:

$$FB_X = 2 * \frac{\overline{X_{\text{model}} - X_{\text{obs}}}}{X_{\text{model}} + X_{\text{obs}}}$$

normalised mean square error:

$$NMSE_X = \frac{\overline{(X_{\text{model}} - X_{\text{obs}})^2}}{X_{\text{model}} * X_{\text{obs}}}$$

For temperature given in degrees Celsius the following non-normalised statistics were used:

mean bias:

$$MB = \overline{T_{\text{model}} - T_{\text{obs}}}$$

mean absolute bias:

$$MAB = \overline{|T_{\text{model}} - T_{\text{obs}}|}$$

root mean square error:

$$RMSE = \sqrt{\overline{(T_{\text{model}} - T_{\text{obs}})^2}}$$

*Author contributions.* Coordination of the study, leading of UrbiPragensi KK4 concept: JR, coordination of the observation campaign: OV, design of the observation campaign: JKe, OV, JR, MB, realization of the observation campaign: OV, JR, JG, KE, PK, MB, VF, PH, JKa, JD, TH, KH, JKe, observation postprocessing: OV, KH, SN, JR, JG, PK, MR, MB, VF, urban input data collection and processing: JG, JR, PK, OV, NB, WRF and CAMx simulations and their processing: KE, PH, MB, JK, JR, PK, OV, NB, JD, PALM model development and testing: JR, MS, PK, VF, PALM simulations configuration and run: JR, MS, PK, result postprocessing and visualisation: JR, JG, MR, PK, MB, VF, OV, meteorology and air quality expertise: MS, MB, VF, OV, PH, JKe, text contribution and revisions: all co-authors.

*Competing interests.* The authors declare no competing interests.

*Acknowledgements.* Financial support was provided by the *Operational Program Prague – Growth Pole of the Czech Republic* project "Urbanization of weather forecast, air-quality prediction and climate scenarios for Prague" (CZ.07.1.02/0.0/0.0/16_040/0000383) which is co-financed by the EU. The co-author MS was supported by the Federal German Ministry of Education and Research (BMBF) under grant 01LP1601 within the framework of *Research for Sustainable Development* (FONA)[1].

The terrain mapping campaign of building properties was co-financed by the *Strategy AV21* project "Energy interactions of buildings and the outdoor urban environment" which is financed by the Czech Academy of Sciences. We would like to thank prof. Jiří Cajthaml and students of the Faculty of Civil Engineering of the Czech Technical University in Prague for their help with the terrain mapping campaign.

We would also like to thank the Global Change Research Institute (CzechGlobe) for lending the IR camera, Ms. Ivana Hájíčková for enabling the HF measurements in Zelená St., and the Czech Technical University (Ms. Lenka Bedrníková and Mr. Josef Šteffel) for enabling us to perform the heat flux measurement at Sinkule house and the wind measurements on the rooftop of the Faculty of Civil Engineering of CTU.

We would like to thank our colleagues who organised and carried out the measurements during the observation campaign and are not authors of this article. Monitoring vehicles - Zdeněk Běťák, Petr Goll, Jan Kufel, Luboš Vrána; mobile and rooftop wind measurement - Petra Bauerová, Jan Hadinger, Zdeněk Proškovec, Hana Škáchová; IR measurements - Jana Řadová, Michal Žák; technician - Jiří Gajdoš.

We would like to thank our colleague Martin Glew for language revision of the manuscript text.

The PALM simulations, pre- and postprocessing were performed on the HPC infrastructure of the Institute of Computer Science of the Czech Academy of Sciences (ICS) supported by the long-term strategic development financing of the ICS (RVO:67985807). Part of the simulations were performed on the supercomputer of the Czech supercomputing centre IT4I which was supported by The Ministry of Education, Youth and Sports from the *Large Infrastructures for Research, Experimental Development and Innovations* project "IT4Innovations National Supercomputing Center — LM2015070" and on the supercomputers of the North-German Supercomputing Alliance (HLRN). The WRF and CAMx simulations where done on the HPC infrastructure of the Department of Atmospheric Physics of the Faculty of Mathematics and Physics of the Charles University in Prague which was supported by the *Operational Program Prague – Growth Pole of the Czech Republic* project "Urbanization of weather forecast, air-quality prediction and climate scenarios for Prague" (CZ.07.1.02/0.0/0.0/16_040/0000383) which is co-financed by the EU.

---

[1]https://www.fona.de

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
