# Peer review of "Validation of the PALM model system 6.0 in a real urban environment; case study on Prague-Dejvice, Czech Republic"

_Geoscientific Model Development, 2020_

## Referee Comment (RC1) · Anonymous Referee #1 · 30 Oct 2020

General considerations

In this contribution, the authors use the recently developed PALM-4U urban LES model at very fine grid spacing (2 m) to model details of atmospheric properties, wall/surface temperatures etc., and compare the results to a detailed (dedicated) data set from the city of Prague. This is, first of all, a tremendous achievement – with respect to the modeling effort, as well as the dedicated observational campaign. It constitutes a major challenge for a numerical model to 'reproduce' various point measurements in an environment as complicated as an urban canopy layer.

Overall, the observations and also the modeling approach are well and comprehensively described (there are a number of editorial issues, though. See major comment 1). With respect to the description of the obtained results, however, the paper looses its conciseness. By showing too much material (major comment 2), the 35 (!) figures are spent on just showing and describing the outcome of the simulation, rather than trying to identifying the true potential of PALM-4U – and also its limitations. According to my judgement, one may indeed conclude from the present results that, provided the correct wall and surface, etc. parameters are available, one can reproduce even the details of the wall temperatures, etc. If it comes to the flow characteristics (within the canopy!), temperature is pretty well reproduced (rms's likely below 1 K even for the bad cases). The magnitude of wind speed is largely improved over the driving (WRF) diagnosed wind speed, but the details (timing of local maxima etc.) are essentially not captured. As for air pollution (which of course is the greatest challenge), we get no more than what we can expect from a 'good dispersion model' (minor comment to l. 817). If the major problems were addressed (major comments 3-5) rather than just describing the output, one could further assess the potential of these fascinating simulations. For example, one of the relevant questions would be to decide what has more potential: to further improve the model (parameterizations) or to assess all the necessary surface parameters at the best possible detail.

Major comments

1) Editorial issues. The material is not very well described. Most of the figure captions are not complete (I have added a number of specifically missing explanations in the detailed comments – but certainly not all). The authors should carefully go through all the figure captions and assess whether all the lines, symbols etc. are explained. The most important missing information for most of the figures is, which simulation (parent / child domain) is shown. Furthermore, essentially all the figures having a color code, this is not large enough to allow to distinguish the numbers (e.g., Fig, 17, 26, 28, ..). Finally, a lot of material is provided in the supplementary material (which is good in principle) but should be better identified (see detailed comment to l. 144)

2) In fact, the authors are showing much too much material. Even if they have already put quite a substantial part of the examples to the supplemental material, it remains much too much what is being presented. The problem with this is that it produces lengthy descriptions of each example (where and when which variable shows under/overestimation etc.). Even an interested reader will get lost with all the unnecessary information (e.g., on the date, site, etc.), which of course has to be provided when showing all the individual examples. But as an 'interested researcher' I am not interested in learning whether PALM underestimated the daily cycle on July 15 (and not on July 16) in the afternoon, at site 11 while the underestimation was not so great at site 07 on the same day, but with slight differences on the next). I am interested in learning i) the typical (average) behavior and ii) the exceptions. Thus, rather than showing so many different sites (e.g., for wall surfaces it is Fig 13-15 and associated text, and Fig 18-21 plus associated text), one should seek a form to best characterize the mean performance and show one example and then provide statistics on the other sites (the reader could then see what is presented in Fig. xx and judged 'average performance', and learn that this figure is associated with a mean bias of yy and an rms of zz (whatever the statistics are to best characterize the behavior). This would leave more room (and attention) for the exceptions – e.g., for the wall temperatures there is this example with 'old buildings' having been insulated in the meantime (since the data base has been produced) and thus producing wrong wall heat fluxes and consequently temperatures.

3) To some degree, the authors suffer from a quite common weakness in 'model validation studies', i.e. that they find everything 'well reproduced' (in the conclusions, for example, 'all is well represented' (lines, 840, 843, 848, 856, 862). Even if some 'exceptions' are usually mentioned, this is not really helpful. Most often, one learns much more from the cases, in which the model fails. Examples of issues that could/should be addressed are: the 'opposite wind direction' (major comment 5); the strange peaks in wall temperatures (e.g. Figs 13, 14, 15, 18) – which might be related to the wall heat flux (Fig. 30); the wrong attribution to surface characteristics (see the above insulations

issue); and the reflection (non-Lambertian reflection missing in the parameterization). In all cases, the problem is 'mentioned' (sometimes with a hypothesis why it occurs, sometimes not). If possible the origin of the failure should be discussed – and also some statistics should be provided on how often (how strongly etc.) this occurs.

4) The authors provide vertical profiles in comparison to a radio sounding – but the sounding is not within the modeling domain. This, in principle, is a 'no go' in a model validation study. Either I have data that is appropriate for a comparison, then I compare it - or I leave it. In the present case, I have no clue, whether the differences are due to a model deficiency or just due to different locations being compared (and the differences are substantial, e.g. Fig. 9 for wind speed 20.7. 00:00). Thus the options are: only show the model in order to characterize the situation. Alternatively, the coarser model results (WRF) could be used to estimate the height, above which the grid point-to-grid point variability 'vanishes' (i.e., the model results are more or less spatially representative over the larger 'urban' area) – and then make the comparison only down to that height.

5) Comparison to wind speed at the highest building ( referring to l. 510: 'This confirms the disagreement of wind speed'): if I understand Section 3.5 correctly, this analysis uses the nearest grid point in the PALM model (and we would be interested to learn which domain, see minor comments). When I check the wind direction at the times when PALM has what the authors call an extraneous (but probably mean extraordinary) peak (e.g., afternoon of 21.7., morning of 4.12., etc.) it is always when the wind direction is plus/minus opposite (or at least largely deviating). This brings us back to the (minor) comment to l. 232: even if wind speed was measured on the tallest building, if the observation is close to roof level, the flow will be influenced by the building itself. The flow impinging on the building will detach, form a wake, re-attach etc. – the details of which depend on the building structure, the flow of course – and the direction (see, e.g. Oke et al. / Climates of Cities / Chapter 4, e.g. Fig, 4.3). If the grid spacing is 2 m, all this is resolved by the model. If the overall (synoptic) wind direction is correct, the

model might still not capture the details of the near-roof flow structure. I consequently think that i) this type of situation deserves a more in-depth analysis and ii) the reason for the wrong wind direction needs to be found. If we take, e.g., Fig S17, 27.11.: the wind direction is almost constantly 'opposite'. If this is not wrong already in the WRF simulation (what I cannot imagine), it probably has to be a local (very local) type of reversed flow or similar close to the roof. A first step therefore would be to compare the WRF wind direction (and compare to the synoptic) over these periods of 'wrong wind direction' in PALM.

Minor comments

l. 11 for certain...

l. 36 the UHI effect....

l. 83 The Prague Dejvice....

l. 116 The majority...

l. 117 The lancover map...

l. 143 ...by the infrared camera: only one? Or rather 'by infrared cameras'? (if only one: by an infrared camera....)

l. 144 Tables S2 and S3: after having found those tables in the myriad of files provided in the supplemental material (it would probably be helpful to somewhere state that all the supplemental figures and tables can be assessed by following the 'index link '), the tables contain a number of unexplained acronyms (what is MV, HF, AQ, etc., also LCZ is not explained [can be guessed when reading the caption....])

l. 167 what is the TYRSY01 system (not introduced)

l. 229 proposed? Rather 'positioned'

l. 232 60 m high: more important is the height above roof. Even if the building where

the sensor is mounted is high (the highest), it will by itself impact the flow. Also, in Fig. 1 the location is not labelled 'FSv' (but 'anemometer'). See major comment 5

l. 235 of the inner domain

l. 236 in Fig 1 the 'drone site' is labelled 'autumn only'

l. 246 The radiosonde...

l. 254 The PALM...

l. 272 the stand-alone...

l. 297 in Fig. S1

l. 318 wherever possible

l. 322 The Prague...

l. 326 using data from the terrain mapping campaign

l. 336 described in Tab S6

l. 345 initial and boundary conditions. The authors only provide the source (WRF) for the BC, but not the type of BC (also: at the domain top). Furthermore: how are the 49 WRF levels mapped to the PALM levels (especially within the canopy)? For example, the lowest WRF level (probably at 10 m [?]), is in the NOAH LSM meant to be above the canopy (and a bulk canopy approach is used), while in the lowest PALM level it is 'between the buildings'. The same probably is true for the second lowest level. Finally: how are the BCs interpolated in time?

Fig. 3 caption: what are the green and red rectangles?

l. 423 the 'weather balloons' are usually referred to as 'radiosondes' (later occurrences)

l. 462 (i.e., caption Fig. S16): first, it should be mentioned that 'modelled' refers to the CAMx simulation (that's at least what I assume). The reference to Fig. Supp13

should likely read Fig S10. How is the 95% confidence interval determined (from the distribution of all the sites and days?)

l. 465 morning and evening peaks are. . . and appear. . ..

l. 466 than in observations. . . .. But the CAMx model. . .

l. 468 play a more. . .

l. 473 at the southern edge. . . . . . with the model simulation

l. 474 necessarily to match

Fig. 9 apparently, the modelled results are averages from the parent domain (caption). When explaining (in the text) that the comparison is 'not fair' (because the sounding is from a different location) this should be mentioned there, too.

l. 482 is much smaller: indeed, not only much smaller, but also has a completely different shape. For the midnight sounding, the maximum wind speed is 10 ms-1 at about 800 m height, while PALM has very low wind speeds. This must be detectable in the WRF simulation (PALM can likely increase the drag, but I wouldn't expect this to be so dramatic. Can the authors comment on this?

l. 486 these discrepancies (or this discrepancy)

Fig 11 caption: state, which side is the summer and which the winter episode. Also, it should be stated which simulation (parent, child) is used here.

l. 504 'extraneous': the authors probably mean extraordinary? (same on l. 508, 509, ...)

l. 509 is somewhat overpredicted

Fig. 12 caption: south-west (not west-south). Also, the colour scale is hard to even detect (nothing to say about being able to read the numbers)

L 516 of about 290 K

Fig. 13ff what is the grey shading? (day/night?)

l. 551 when the surface temperature. . .

l. 554 two sharp peaks: these are 10 to 15 K (which is quite substantial) – and it also occurs on the day before. Can the authors offer an explanation for these peaks? (if we look at Fig 8, it does not seem to be a radiation effect. . ..)

l. 555 it is striking to note. . .. Here the horizontal surfaces are probably referred to.

Fig. 17 soil moisture panel, inlet: this is definitely too small to be distinguished (in fact, all the inlets are too small)

l. 595 was adjusted: based on what?

l. 666 the first two examples: in the caption of the corresponding figures (also Fig 24) it would be helpful to explain what influence is being shown (the identification of the site number is not extremely helpful for the reader who does not know all the sites).

l. 690 The terrain. . .

l. 712 Two further consequences. . .. are an altered. . .

l. 720 due to their local nature

l. 731 with the modelling episode

l. 736 the wall temperature: shows again – as in Fig. 14 for site 12 – these stratage peaks. These are also visible in the heat flux. Could this be a starting point to evaluate the origin of those peaks?

Fig 32 caption: the yellow band. . .: this is only visible in a few of the sub-plots. Also, there seems to be a black dashed line associated with the yellow band, which is not explained (all the same for Fig. 33)

l. 791 in accordance. . ..: overall, the most relevant observation, i.e. that (diagnosed) wind speed from WRF largely overestimates magnitude and amplitudes in the time

series' (at all sites, all episodes) is not even made.

l. 800 when concentrations. . .

l. 806 . . .aerological soundings at 06:00 UTC 21 July (Fig. 33): I don't think Fig. 33 shows results from the aerological soundings – nor does it show a (particularly strong) underprediction.

l. 817 . . .fulfil the criteria for dispersion models. . .: it must be noted, however, that the 'dispersion models' of Chang and Hanna are simple 'Gaussian' or hybrid plume dispersion models. Thus with an immense increase in computing time and effort, the results are not convincingly better. Can the authors comment on that?

Fig. 35 what are the numbers in the inlet (identifying the '+' signs) referring to?

---

## Referee Comment (RC2) · Anonymous Referee #2 · 14 Dec 2020

I would like to commend the authors for this comprehensive study and the effort that went in this experimental campaign for model validation. The manuscript is interesting and useful. However, in my opinion there are several points that the authors should consider for improving the presentation of their results/comparison and, also, the reproducibility of their simulations.

Main comments

1) The authors correctly point out line 901-910 that the domain may be too small to develop proper turbulence conditions. Indeed, this is quite certain in my opinion based on my personal experience with 3D nesting in LES. They have an external grid of 4x4km^2

and an internal grid of about 1.5x1.5km^2. The outer grid resolution in the PALM model is 10meters while the internal grid is 2m. There are two issues: 1.25km is not sufficient to create an organized and realistic turbulent flow from the outer edges (where turbulence is synthetically generated) and the transition to the inner grid has a refinement factor of 5 and will take a lot before turbulence at smaller scales is generated. The authors seem aware of these issues. My point is that they do not try in any way to evaluate the turbulent state nor the relevance of the error induced by their simulation setting.

The authors write that within the urban canopy the turbulence will adjust faster due to the obstacles. This is true but did they try to quantify if they have developed turbulence?

The authors should add and discuss the spectrum of turbulence and its evolution in space e.g. calculate it right before moving from coarse to fine grid (considering wind direction) and in some reference locations moving away from the leading edge of the fine grid towards the trailing edge. Ideally for a position relatively near the ground and one more elevated.

It would be also interesting to see vertical profiles of horizontally averaged statistics over the inner portion of the inner grid, this to see if we observe typical canopy layer, mixing layer and boundary layer profiles.

2) A general comment is that the paper is too long and some of the comparison do not really add new information. On the contrary some discrepancies in the model results with respect to the measurements are not discussed but just mentioned. Below some more detailed comments.

2.1) Figure 5, 6, and 7 are not necessary. Move it to the supplement and simply give a short description. Instead it is very important in my opinion to include WRF model output in figure 9, 10 and 11. In this way the reader can appreciate if there is any significant difference between PALM and WRF. Moreover, WRF simulation results should be included both for the location of the PALM profile and the location of the

measured profile. In this way the reader can also evaluate how meaningful is to include the measured vertical profiles (that are for a location outside PALM coarse domain) in the comparison.

2.2) The peaks in the vertical walls temperature, figure 13 (winter), figure 14(winter) should be discussed and it should also be discussed why the peak does not appear in figure 15. Are these peaks related to the radiative transfer model and its interaction with the wall at a particular sun angle under clear sky conditions? This may be an explanation of these peaks appearing at the same time of the day, and also the reason that they do not appear in summer. I noticed that in figure 18 a similar peak appears also in the observation (location 06_4_V). Did the specific wall geometry in relation to sun angle generate the peak? However, it is my opinion that these spikes in temperature and heat fluxes are very local surface effects with little or no influence on the overall heat exchange of the walls and even less on the atmospheric flow (see also point 3 below).

2.3) Fig 11. The discrepancy in wind direction should be discussed in more details, and the WRF results should also be included both for wind speed and direction. I think that adding WRF wind direction will help in interpreting the results. Also, please make the shorter arrows longer (or simply make all the arrows of the same length since the magnitude is already reported in the lower panel). Please, do the same in the supplement.

2.4) figure 13, 14, 15, 16, 17, 22, 23, 24. I think that the WRF surface temperature should be included in the comparison for the "horizontal" surface. This should be just a single extra line for each location and allows to appreciate the difference/similarity with PALM, and this difference/similarity should also be discussed.

2.5) Why not using the heat flux measurement locations (Sinkule and Zelena) also in the discussion in section 4.3 (i.e. as typical urban locations). This may allow discarding some of the figures.

3) The authors discuss the error induced by discretization (section 4.3.5) and also its possible solution (section 5.2. lines 873-884). I think that these discussions are partially misleading and must be modified.

I think that step-like discretization errors at this resolution can create local discrepancies (local in space and/or in time, see point 2.2 above) of limited overall significance; they seem significant only for the very local surface heat flux and temperature. Moreover, by increasing the resolution (e.g. 1m or less) the discretization errors become of the same order of the neglected SGS wall features. Therefore, there is no point in avoiding these errors if the building is not represented in all its many details.

Furthermore, it seems to me that the error, being mainly related to the radiative transfer model, can be fixed irrespective of the representation of the building in the fluid dynamic solver (i.e. I think that there is no need to have a more sophisticated immersed boundary method in the fluid solver to fix the behavior of the radiative transfer model since for the latter it is enough to keep track of the local actual surface angle).

Finally, I think that the influence of the current discretization method on the flow is quite limited and local at the considered resolution, and become quickly negligible (and likely comparable with the neglected SGS wall elements) if further increasing the grid resolution.

4) line 845-847. The authors correctly point out that the PALM results are often very similar to the WRF results.

This is true for the close to surface air temperature, where WRF seems to have a better behavior in winter and a similar performance in summer compared to PALM. This should be discussed, and some statistical indices should be included comparing WRF and PALM performances.

On the contrary the wind speed is generally significantly different, and this is obviously because the buildings are simulated explicitly. The explicit building representation also

allows a better agreement between PALM and the observation for wind speed with respect to WRF (within the urban canopy). This should also be discussed, and some statistical indices should be included comparing WRF and PALM performances.

In my opinion this behavior demonstrate that the role of a very fine scale heat surface description is minor compared to having the obstacle represented and can perhaps be not so important. For the correct modeling of the street level urban atmospheric flow It seems sufficient to fairly reproduce the overall integrated heat exchange of the building wall. Very local temperature and heat flux effects (those investigate here with point wise sensors) seems quite negligible for the simulation of the mean flow and mean pollutant dispersion at street level.

5) The author should improve the reproducibility of their simulations. This is a GMD manuscript and I think that all the necessary input files for performing the simulations should be provided for download, not just a generic reference to the open source model code. If I am not wrong for PALM these should be, the namelist file e.g. "prague_p3d", form both grid levels, the dynamic input file, e.g. "prague_dynamic", the static input file for both grid levels, e.g. "prague_static". The author should also include the file for the shortwave and long wave radiation that are obtained from WRF and used in the simulation, and briefly explain how they are generated.

5.1) The authors write that the details for obtaining the WRF (interpolated) wind field on the PALM grid goes beyond the scope of this manuscript (lines 3675-376). I think this detail should be well explained, also because they did not refer to any previously published full validation of this WRF-PALM coupling.

5.2) The coupling between WRF and PALM includes the specification of the horizontal pressure gradients in the two orthogonal directions at any level. The typical profiles applied for the simulations in summer and winter should be included in the supplement to the manuscript.

5.3) It seems that buildings are not resolved explicitly in the outer PALM grid. What

local parametrizations have been used to describe the urban canopy?

Other comments

Page 3, I suggest removing lines 76-83 they seem redundant.

Page 4, line 116. What does it mean (+-20m), about 20m?

Figure 12. There is no color bar showing relations between temperature and colors.

Figure 32, 33, what does it mean "locations_V" or "location_Z"? I could not find the definition for the nomenclature "_V" "_Z".

Figure 34. The light green band is almost invisible.

Figure S13, S14, S15, add the vertical axis label

Figure S16, caption. What is Fig supp13? should it be Figure S10?

---

## Author Comment (AC1) · 17 Feb 2021

**General considerations**

*In this contribution, the authors use the recently developed PALM-4U urban LES model at very fine grid spacing (2 m) to model details of atmospheric properties, wall/surface temperatures etc., and compare the results to a detailed (dedicated) data set from the city of Prague. This is, first of all, a tremendous achievement – with respect to the modeling effort, as well as the dedicated observational campaign. It constitutes a major challenge for a numerical model to 'reproduce' various point measurements in an environment as complicated as an urban canopy layer. Overall, the observations and also the modeling approach are well and comprehensively described (there are a number of editorial issues, though. See major comment 1). With respect to the description of the obtained results, however, the paper looses its conciseness. By showing too much material (major comment 2), the 35 (!) figures are spent on just showing and describing the outcome of the simulation, rather than trying to identifying the true potential of PALM-4U – and also its limitations. According to my judgement, one may indeed conclude from the present results that, provided the correct wall and surface, etc. parameters are available, one can reproduce even the details of the wall temperatures, etc. If it comes to the flow characteristics (within the canopy!), temperature is pretty well reproduced (rms's likely below 1 K even for the bad cases). The magnitude of wind speed is largely improved over the driving (WRF) diagnosed wind speed, but the details (timing of local maxima etc.) are essentially not captured. As for air pollution (which of course is the greatest challenge), we get no more than what we can expect from a 'good dispersion model' (minor comment to l. 817). If the major problems were addressed (major comments 3-5) rather than just describing the output, one could further assess the potential of these fascinating simulations. For example, one of the relevant questions would be to decide what has more potential: to further improve the model (parameterizations) or to assess all the necessary surface parameters at the best possible detail.*

First of all, we would like to thank very much the reviewer for this very valuable detailed review with a lot of helpful suggestions.

We understand that the presentation of our results may have suffered from the fact that our analysis was a summary of two years of model development and improvement in which focusing on the details was often the key to identification of model issues and subsequent model improvement. In this revision, we tried to restructure the content of the manuscript to focus more

on condensed summarised outputs according to the suggestions of both reviewers. We also tried to extend and strengthen the discussions of our findings. We hope the revised version is much more attractive for an "interested researcher". We updated all parts of the manuscript, and chapter 4 was fully restructured and rewritten. We tried to address all suggestions of both reviewers as described in the answers to the particular comments.

Considering the question of interest given as an example, the question of the potential of assessing the surface parameters in the greatest possible detail is very complex and we attempt to tackle this problem in a companion paper (gmd-2020-126: Sensitivity analysis of the PALM model system 6.0 in the urban environment). To address it in this manuscript, we added a short discussion to the end of section 6.2.

**Major comments**

*1) Editorial issues. The material is not very well described. Most of the figure captions are not complete (I have added a number of specifically missing explanations in the detailed comments – but certainly not all). The authors should carefully go through all the figure captions and assess whether all the lines, symbols etc. are explained. The most important missing information for most of the figures is, which simulation (parent / child domain) is shown. Furthermore, essentially all the figures having a color code, this is not large enough to allow to distinguish the numbers (e.g., Fig, 17, 26, 28, ..). Finally, a lot of material is provided in the supplementary material (which is good in principle) but should be better identified (see detailed comment to l. 144)*

We went thoroughly through all figures and their descriptions and we considered again the graphical issues (fonts, lines, colors) as well as the completeness and comprehensibility of the description and fixed all issues we found. The supplements were restructured to provide better orientation in their content (see detailed answer after comment l. 144).
To improve the quality of the presentation, the revised manuscript also underwent language revision by a native speaker with a good knowledge of the topic.

*2) In fact, the authors are showing much too much material. Even if they have already put quite substantial part of the examples to the supplemental material, it remains much too much what is being presented. The problem with this is that it produces lengthy descriptions of each example (where and when which variable shows under/overestimation etc.). Even an interested reader will get lost with all the unnecessary information (e.g., on the date, site, etc.), which of course has to be provided when showing all the individual examples. But as an 'interested researcher' I am not interested in learning whether PALM underestimated the daily cycle on July 15 (and not on July 16) in the afternoon, at site 11 while the underestimation was not so great at site 07 on the same day, but with slight differences on the next). I am interested in learning i) the typical (average) behavior and ii) the exceptions. Thus, rather than showing so many different sites (e.g., for wall surfaces it is Fig 13-15 and associated text, and Fig 18-21 plus associated text), one should seek a form to best characterize the mean performance and show one example and then provide statistics on the other sites (the reader could then see what is presented in Fig. xx*

*and judged 'average performance', and learn that this figure is associated with a mean bias of yy and an rms of zz (whatever the statistics are to best characterize the behavior). This would leave more room (and attention) for the exceptions – e.g., for the wall temperatures there is this example with 'old buildings' having been insulated in the meantime (since the data base has been produced) and thus producing wrong wall heat fluxes and consequently temperatures.*

We are aware that there is too much material for one paper. It represents an important part of more than two years of work of a sizable group of researchers. We considered splitting the material into two papers at the beginning of our work on this manuscript but we finally decided to publish it all together as we considered all this information interconnected, and we felt that the publication as one body of work allows the relationships to come into view..

To address this comment, we completely restructured former chapter 4. We split the chapter into two chapters "Evaluation of model simulation setup" and "Results". The evaluation of the correct model setup was improved and extended by analysis of spatial development of the urban boundary layer. In the results, we removed most of the information about individual locations, with temporal graphs of quantities in individual points from the manuscript and we moved them to the supplements, as we still consider them valuable information, especially  for the model developers. We also removed most of the corresponding text in the manuscript and we kept only carefully selected examples. Instead, in order to show the average performance, we supplemented the statistical metrics of these quantities in the form of tables and scatter plots. We also added corresponding discussion of the findings following from these metrics. The examples were selected as cases which require either special analysis or which highlight specific model issues, also taking into account the detailed suggestions in both reviewers comments and we have rewritten the discussion of those cases. In the case of surface temperatures, the analysis and discussion include modelling of grass surfaces, modelling of walls of contemporary office buildings and the issue of glass walls, the analysis of the "strange peaks" of modelled surface temperature, and discussion of the surfaces influenced by the plant canopy. We restructured, condensed, and clarified the section about the effects of the discretization. Similar changes were done for other evaluations (wall heat fluxes, street canyon observation, roof observation) with the goal of providing a more condensed and focused text. This way we hope to show both the average performance as well as some interesting individual cases.

As for the issue of insulation in old buildings, this was an improper formulation on our part. The buildings were insulated at some point in history before the data collection. The problem in the results was probably that the insulating efficiency of this layer was underestimated in the model input data. We reformulated the text to express it clearly.

*3) To some degree, the authors suffer from a quite common weakness in 'model validation studies', i.e. that they find everything 'well reproduced' (in the conclusions, for example, 'all is well represented' (lines, 840, 843, 848, 856, 862). Even if some 'exceptions' are usually mentioned, this is not really helpful. Most often, one learns much more from the cases, in which the model fails. Examples of issues that could/should be addressed are: the 'opposite wind direction' (major comment 5); the strange peaks in wall temperatures (e.g. Figs 13, 14, 15, 18) –*

*which might be related to the wall heat flux (Fig. 30); the wrong attribution to surface characteristics (see the above insulations issue); and the reflection (non-Lambertian reflection missing in the parameterization). In all cases, the problem is 'mentioned' (sometimes with a hypothesis why it occurs, sometimes not). If possible the origin of the failure should be discussed – and also some statistics should be provided on how often (how strongly etc.) this occurs.*

We agree with the reviewer that this formulation isn't too helpful to the reader. While not wishing to excuse this error, in our defence we would mention that we were working hard on model development and improvement, along with finding the proper configuration and obtaining input data and parameters for nearly two years, and many of the "well reproduced" quantities were not well reproduced at the start of the project; but this is looking at things from the perspective of a model developer. To address this comment, we went through the text and we tried to restructure it and to remove unnecessary elements and statements. As outlined in the answer to the previous comment, we added parts where the mean performance of the model is discussed and averaged statistics indicating the bias or the scatter are provided. The presentations of special cases (grass, walls with complex structure, plant canopy effects) were improved and condensed. We also refined the text about non-Lambertian reflection and improved it by discussion of its practical influence on the results. This particular observation campaign does not provide sufficient data for "hard" results supported by corresponding statistics in this particular regard, so we added only an estimation based on the available observation data. We also added an analysis and discussion of the "strange peaks" in section 5.1.5 to address this part of the reviewer's comments. Our revisions address the presentation of the wind on FSv and its discussion, as detailed in our answer to comment 5). We added statistics measures for the FSv rooftop observations, as well as for street canyon measurement vehicle observations.

*4) The authors provide vertical profiles in comparison to a radio sounding – but the sounding is not within the modeling domain. This, in principle, is a 'no go' in a model validation study. Either I have data that is appropriate for a comparison, then I compare it - or I leave it. In the present case, I have no clue, whether the differences are due to a model deficiency or just due to different locations being compared (and the differences are substantial, e.g. Fig. 9 for wind speed 20.7. 00:00). Thus the options are: only show the model in order to characterize the situation. Alternatively, the coarser model results (WRF) could be used to estimate the height, above which the grid point-to-grid point variability 'vanishes' (i.e., the model results are more or less spatially representative over the larger 'urban' area) – and then make the comparison only down to that height.*

We agree with the reviewer that the different location of the sounding and modelling domain (even just a few km) is a serious problem. On the other hand, we still consider the information provided by the soundings valuable to some extent, as the distance between the locations is small and the surface conditions are similar because both locations are situated in a similar urban complex. In the revised manuscript we emphasize this fact and discuss possible implications so that the reader is aware of this issue. Moreover, we connected this comment with suggestion 2.1) of RC2 and we supplemented the graphs with the WRF profiles in both

locations, that is the sounding location as well as the location of the modelling domain. Further, we unified the different sets of comparison graphs - sounding vs. WRF profiles (Fig. 6, 7) and sounding vs. PALM profiles (Fig. 9, 10, S13, S14, S15) - into one series of graphs (FIg. 5 and 6 in the revised manuscript) which provides complex information about vertical profiles; we hope they give the reader valuable and clearly arranged information. We updated the text description accordingly.

*5) Comparison to wind speed at the highest building ( referring to l. 510: 'This confirms the disagreement of wind speed'): if I understand Section 3.5 correctly, this analysis uses the nearest grid point in the PALM model (and we would be interested to learn which domain, see minor comments). When I check the wind direction at the times when PALM has what the authors call an extraneous (but probably mean extraordinary) peak (e.g., afternoon of 21.7., morning of 4.12., etc.) it is always when the wind direction is plus/minus opposite (or at least largely deviating). This brings us back to the (minor) comment to l. 232: even if wind speed was measured on the tallest building, if the observation is close to roof level, the flow will be influenced by the building itself. The flow impinging on the building will detach, form a wake, re-attach etc. – the details of which depend on the building structure, the flow of course – and the direction (see, e.g. Oke et al. / Climates of Cities / Chapter 4, e.g. Fig, 4.3). If the grid spacing is 2 m, all this is resolved by the model. If the overall (synoptic) wind direction is correct, the model might still not capture the details of the near-roof flow structure. I consequently think that i) this type of situation deserves a more in-depth analysis and ii) the reason for the wrong wind direction needs to be found. If we take, e.g., Fig S17, 27.11.: the wind direction is almost constantly 'opposite'. If this is not wrong already in the WRF simulation (what I cannot imagine), it probably has to be a local (very local) type of reversed flow or similar close to the roof. A first step therefore would be to compare the WRF wind direction (and compare to the synoptic) over these periods of 'wrong wind direction' in PALM.*

As in the original version we considered that the wind observation on FSv building is only a supporting information and not part of the main focus of the study, we did not discuss it in detail; the focus of the study is the street canyon level comparison. To attempt to address this issue, we  have dived deeper into the problem. At first, we tried to decide whether this effect (which appears only during winter episodes) is a technical problem or some microscale dynamic effect. We went through the entire process of the measurement and data collection with the technicians, and we checked all our postprocessing tools. According to the technicians, the placement of the sensor and the settings of the equipment were identical for summer and winter episodes. However, we found no photo documentation of the sensor for the winter episode similarly as for the summer. We thus were not able to conclusively check and prove the actual orientation of the sensor in the winter study. We also tried to compare wind direction data from nearby synoptic stations as well as from WRF and PALM 3D fields. Despite all this effort, we came to no solid conclusions above some speculations. As we cannot exclude the possibility of a technical issue (the wrong orientation of the sensor) during the winter episode, which would reverse the wind direction, we decided to exclude the wind direction from the presented FSv

wind observation and we updated the description. The potential misorientation of the sensor would not influence the observed wind speed so we consider this approach as the correct one.

We are aware that there are recirculation zones present behind sharp corners on bluff bodies. The grid resolution in the inner domain was too coarse to simulate them accurately but the sensor location at 2 m above the roof in the centre of the building should be well outside of them. Still, the flow is certainly strongly affected by the  building and an accurate simulation of the details of the local flow would require higher grid resolution.

We consider this issue a very interesting and important problem and in new projects we are starting, we intend to repeat observations of similar type using an improved design.

To improve the presentation of the wind on FSv and its discussion, we first reworked the wind graphs themselves; the wind magnitude graphs are taller and the wind values from WRF and nearby synoptic station Ruzyne were added. We added statistics for this observation and we also improved the description of this part of the study to express clearly that we do not consider these observations as the observations of the free flow but as the observations of a flow influenced by surface effects, but at a higher level than the street canyon observations. To emphasize this, we moved this part from section 4.2 to section 5.3.3 "Wind speed on the roof" in the revised version.

**Minor comments**

*l. 11 for certain. . .*

Accepted.

*l. 36 the UHI effect. . ..*

Accepted.

*l. 83 The Prague Dejvice. . ..*

Accepted.

*l. 116 The majority. . .*

Accepted.

*l. 117 The lancover map. . .*

Accepted.

*l. 143 . . .by the infrared camera: only one? Or rather 'by infrared cameras'? (if only one: by an infrared camera. . ..)*

In fact, measurements were done with two IR cameras as a precaution, but we utilized the results only from one camera in this study. We adjusted the text in the l.143 accordingly.

*l. 144 Tables S2 and S3: after having found those tables in the myriad of files provided in the supplemental material (it would probably be helpful to somewhere state that all the supplemental figures and tables can be assessed by following the 'index link '), the tables contain a number of unexplained acronyms (what is MV, HF, AQ, etc., also LCZ is not explained [can be guessed when reading the caption. . ..])*

We looked for the optimal way to organize the supplements and we found the html approach better than all other approaches considered for this particular case. To ensure that the reader does not overlook the index.html file, we restructured the supplements in such a way that the main directory contains only this index.html file and the directory ".content". We also refined the structure of the sections and we improved the captions. We added explanations of MV, HF, AQ, LCZ, and other abbreviations in supplement tables. For LCZ, the explanation cites a reference to a paper paper with a more detailed description (Stewart and Oke, 2012).

*l. 167 what is the TYRSY01 system (not introduced)*

The previous sentence referred to the next subsection, where heat flux measurements and TRSYS01 were described. We reformulated the first sentence to make this more clear.

*l. 229 proposed? Rather 'positioned'*

The text of the paragraph was reformulated.

*l. 232 60 m high: more important is the height above roof. Even if the building where the sensor is mounted is high (the highest), it will by itself impact the flow. Also, in Fig. 1 the location is not labelled 'FSv' (but 'anemometer'). See major comment 5*

The text was reformulated - we no longer claim that the measurement represented above roof wind flow. Information on anemometer height above the rooftop (2 m) was added. The label of anemometer in Fig. 1 was changed to "rooftop wind FSv." Corresponding changes were made in Table S1: "FSv (above-roof wind)" -> "FSv rooftop wind", table S2 and S3 "Wind (above roof)" -> "FSv rooftop wind.

*l. 235 of the inner domain*

Accepted.

*l. 236 in Fig 1 the 'drone site' is labelled 'autumn only'*

Accepted. It was supposed to inform the reader that drone measurements were done in autumn only. The same remark is made for some mobile measurement locations e.g. "18 (12 July)". But we renamed the drone location in the map from "drone (autumn only)" to "drone".

*l. 246 The radiosonde. . .*

Accepted.

*l. 254 The PALM. . .*

Accepted

*l. 272 the stand-alone. . .*

Accepted.

*l. 297 in Fig. S1*

Accepted.

*l. 318 wherever possible*

Accepted.

*l. 322 The Prague. . .*

Accepted.

*l. 326 using data from the terrain mapping campaign*

Accepted.

*l. 336 described in Tab S6*

Accepted.

*l. 345 initial and boundary conditions. The authors only provide the source (WRF) for the BC, but not the type of BC (also: at the domain top). Furthermore: how are the 49 WRF levels mapped to the PALM levels (especially within the canopy)? For example, the lowest WRF level (probably at 10 m [?]), is in the NOAH LSM meant to be above the canopy (and a bulk canopy approach is used), while in the lowest PALM level it is 'between the buildings'. The same probably is true for the second lowest level. Finally: how are the BCs interpolated in time?*

While this review process has been ongoing, our scripts for the transformation of the WRF and CAMx outputs to the PALM initial and boundary conditions ("PALM dynamic driver") were included in the PALM official distribution and they are now available in the PALM SVN repository under the UTIL/WRF_interface directory together with a brief description of the principle and utilization. We added this reference to the text together with a brief description and we added a more detailed description of the transformation process to the supplements. We also were able to add a reference to the companion GMD discussion paper about the PALM mesoscale nesting (Kadasch et al.(2020), in review) which has meanwhile been published. Moreover, we added further details regarding temporal interpolation in the mesoscale nesting, as well as which quantities are nested.

*Fig. 3 caption: what are the green and red rectangles?*

Figure 3 and its caption were updated to provide all needed information.

*l. 423 the 'weather balloons' are usually referred to as 'radiosondes' (later occurrences)*

Accepted.

*l. 462 (i.e., caption Fig. S16): first, it should be mentioned that 'modelled' refers to the CAMx simulation (that's at least what I assume). The reference to Fig. Supp13 should likely read Fig S10. How is the 95% confidence interval determined (from the distribution of all the sites and days?)*

The caption was corrected. The 95% confidence interval of the mean is calculated by the bootstrap technique as described in the timeVariation function from the openAir package (https://davidcarslaw.github.io/openair/reference/timeVariation.html). Data from all the stations corresponding to the particular hour are used for its calculation (the same holds for model data). The main text was reformulated slightly to make it clear that also graphs of diurnal variation were made by the openair package.

*l. 465 morning and evening peaks are. . . and appear. . ..*

Corrected.

*l. 466 than in observations. . . .. But the CAMx model. . .*

*Corrected.*

*l. 468 play a more. . .*

Corrected.

*l. 473 at the southern edge. . .. . . with the model simulation*

Corrected.

*l. 474 necessarily to match*

Reformulated.

*Fig. 9 apparently, the modelled results are averages from the parent domain (caption). When explaining (in the text) that the comparison is 'not fair' (because the sounding is from a different location) this should be mentioned there, too.*

Information was added to the introduction of section 4.1 and to the figure captions (FIg. 5 and 6 in the revised manuscript). The figures were reorganized and reworked using both WRF and PALM profiles (see answer to major comment 4) and the corresponding discussion was updated.

*l. 482 is much smaller: indeed, not only much smaller, but also has a completely different shape. For the midnight sounding, the maximum wind speed is 10 ms-1 at about 800 m height, while PALM has very low wind speeds. This must be detectable in the WRF simulation (PALM can likely increase the drag, but I wouldn't expect this to be so dramatic. Can the authors comment on this?*

This problem was due to a technical issue in processing PALM output profile files and was corrected in the revised version. The profile from the WRF simulation was added to the figures for comparison (original Fig. 9 and 10 correspond to Fig. 5 and 6 in revised manuscript) and the text was restructured. In the corrected version the PALM simulation shows better agreement with WRF higher above the surface.

*l. 486 these discrepancies (or this discrepancy)*

Accepted.

*Fig 11 caption: state, which side is the summer and which the winter episode. Also, it should be stated which simulation (parent, child) is used here.*

Caption updated accordingly.

*l. 504 'extraneous': the authors probably mean extraordinary? (same on l. 508, 509, ...)*

Removed.

*l. 509 is somewhat overpredicted*

Missing word added.

*Fig. 12 caption: south-west (not west-south). Also, the colour scale is hard to even detect (nothing to say about being able to read the numbers)*

Text fixed, scale and numbers enlarged.

*L 516 of about 290 K*

Accepted.

*Fig. 13ff what is the grey shading? (day/night?)*

"The grey areas denote the night time." was added to the figure caption.

*l. 551 when the surface temperature. . .*

This text disappeared during the restructuring of the chapter.

*l. 554 two sharp peaks: these are 10 to 15 K (which is quite substantial) – and it also occurs on the day before. Can the authors offer an explanation for these peaks? (if we look at Fig 8, it does not seem to be a radiation effect. . ..)*

The analysis and discussion of these peaks were added to the chapter 5.1.5 "Rapid changes of surface temperature"

*l. 555 it is striking to note. . .. Here the horizontal surfaces are probably referred to.*

This text disappeared during the restructuring of the chapter.

*Fig. 17 soil moisture panel, inlet: this is definitely too small to be distinguished (in fact, all the inlets are too small)*

All graphs of the surface temperature were reworked, the inlets were enlarged and the corresponding figures were recreated.

*l. 595 was adjusted: based on what?*

At this point we must confess that we were quite unspecific. Soil moisture values at this level of detail were not available. Hence, we categorized grass surfaces into natural grass surfaces and urban-like grass surfaces with and without irrigation. Adjustment factors are only based on a best guess resulting from the survey of the locations and our personal experience. In the revised manuscript we explicitly note this.

*l. 666 the first two examples: in the caption of the corresponding figures (also Fig 24) it would be helpful to explain what influence is being shown (the identification of the site number is not extremely helpful for the reader who does not know all the sites).*

Figure 22 disappeared during revision, figures 23 and 24 were combined into one figure and the caption of this new figure was enhanced by the description of the location.

*l. 690 The terrain. . .*

This text disappeared during the restructuring of the chapter.

*l. 712 Two further consequences. . .. are an altered. . .*

Text of section 4.3.5 was restructured, the comment was taken into account.

*l. 720 due to their local nature*

Accepted.

*l. 731 with the modelling episode*

Accepted.

*l. 736 the wall temperature: shows again – as in Fig. 14 for site 12 – these strange peaks. These are also visible in the heat flux. Could this be a starting point to evaluate the origin of those peaks?*

The analysis and discussion of these peaks were added to the new subsection 5.1.5 "Rapid changes of the surface temperature" (compare also the answer to RC2 p. 2.2). This analysis also explains the peaks in wall heat fluxes.

*Fig 32 caption: the yellow band. . .: this is only visible in a few of the sub-plots. Also, there seems to be a black dashed line associated with the yellow band, which is not explained (all the same for Fig. 33)*

The thin black dashed line shows the original 10-minutes averages of the value in the gridbox corresponding to the point of the observation. This was explained in the figure caption but the word "dashed" was omitted by mistake. This omission was fixed.
The yellow band represents the spatial variability of the 10-minutes averages over adjacent grid boxes. The problem of its absence in some graphs lies in the fact that the local spatial variability of the variable in some places is very low and its low spread can lead to the situation that the corresponding band is overlaid by the black dashed line or by the thick solid line representing one-hour moving average of the values. This seems to be also valid information. The band is

only supporting information which gives the reader an estimation how much the value can be influenced by the spatial error caused e.g. by imperfection of the discretized position. To make the band morevisible, we reworked the graphs selecting another colour scheme and thickness for graph variables. We also mentioned the small spread of temperature values in the corresponding text to give the reader relevant information.

*l. 791 in accordance. . ..: overall, the most relevant observation, i.e. that (diagnosed) wind speed from WRF largely overestimates magnitude and amplitudes in the time series' (at all sites, all episodes) is not even made.*

This fact is connected with the configuration of WRF without special urban parameterization. It is noted and discussed in the revised manuscript.

*l. 800 when concentrations. . .*

Accepted.

*l. 806 . . .aerological soundings at 06:00 UTC 21 July (Fig. 33): I don't think Fig. 33 shows results from the aerological soundings – nor does it show a (particularly strong) underprediction.*

Correctly should refer to Figure 5. The reference was fixed.

*l. 817 . . .fulfil the criteria for dispersion models. . .: it must be noted, however, that the 'dispersion models' of Chang and Hanna are simple 'Gaussian' or hybrid plume dispersion models. Thus with an immense increase in computing time and effort, the results are not convincingly better. Can the authors comment on that?*

There are many sources of uncertainties in a real urban atmospheric dispersion scenario. COST ES1006 Model evaluation case studies: approach and results (Baumann-Stanzer, Trini-Castelli, Stenzel (eds.), 2015) thoroughly examined the differences in the accuracy of dispersion modelling (for accidental releases) for well-controlled (wind-tunnel) dispersion scenarios and showed that statistical metrics consistently improve with more complex models (Gaussian, Langrangian with diagnostic flow, CFD = RANS and LES). However, for real outdoor dispersion problems other sources of uncertainty can dominate as demonstrated by other scenarios in the same document. In such cases the advantage of accurate flow simulation by a more complex model can be diminished.
We should stress that the analysis in our paper compares spatio-temporally paired values in geometrically-complex locations. The original requirements on sufficient models by Chang and Hanna mostly originate in an intercomparison by Hanna (2000) which considers several scenarios, where however either arc maxima, or one-hour mean concentrations are compared from releases with well-known emissions, to lessen the effects of natural variability in turbulent dispersion. In our study the comparison is much more strict as 10-minute averages in the original submission and now 1-hour averages in the revised manuscript are paired in time and space and there are significant uncertainties in the emissions.

More complex models, such as LES, can reveal more information about the contaminant dispersion, as also shown by, e.g., COST ES1006 Model evaluation case studies, which also considered other concentration percentiles and peak concentrations or the statistical distribution in puff dispersion. These advanced quantities were not evaluated in our study, but they could be obtained from the simulation results.

In short: although similar criteria of success were used, the quantities compared are more difficult to simulate and make larger demands on the more complex model. The real outdoor dispersion without a single well-controlled emission source contributes to the uncertainty that has to be expected.

We added this short statement to the manuscript: "Although these criteria were developed for simpler models, they are applied to a more complex problem here and are good indicators of usefulness for purpose."

*Fig. 35 what are the numbers in the inlet (identifying the '+' signs) referring to?*

Numbers refer to mobile measurement locations in Fig. 1. This description was added to the caption of Fig. 35.

---

## Author Comment (AC2) · 17 Feb 2021

*I would like to commend the authors for this comprehensive study and the effort that went in this experimental campaign for model validation. The manuscript is interesting and useful. However, in my opinion there are several points that the authors should consider for improving the presentation of their results/comparison and, also, the reproducibility of their simulations.*

First of all, we would like to thank very much the reviewer for this very valuable detailed review with a lot of helpful suggestions. We tried to carefully follow all comments and suggestions of both reviewers to improve the manuscript, making it better structured, more condensed and focused on the important findings, and more attractive for an "interested reader". The details about our changes are provided in the responses to the individual comments.

**Main comments**
*1) The authors correctly point out line 901-910 that the domain may be too small to develop proper turbulence conditions. Indeed, this is quite certain in my opinion based on my personal experience with 3D nesting in LES. They have an external grid of 4x4km^2 and an internal grid of about 1.5x1.5km^2. The outer grid resolution in the PALM model is 10meters while the internal grid is 2m. There are two issues: 1.25km is not sufficient to create an organized and realistic turbulent flow from the outer edges (where turbulence is synthetically generated) and the transition to the inner grid has a refinement factor of 5 and will take a lot before turbulence at smaller scales is generated. The authors seem aware of these issues. My point is that they do not try in any way to evaluate the turbulent state nor the relevance of the error induced by their simulation setting.*
*The authors write that within the urban canopy the turbulence will adjust faster due to the obstacles. This is true but did they try to quantify if they have developed turbulence? The authors should add and discuss the spectrum of turbulence and its evolution in space e.g. calculate it right before moving from coarse to fine grid (considering wind direction) and in some reference locations moving away from the leading edge of the fine grid towards the trailing edge. Ideally for a position relatively near the ground and one more elevated.*
*It would be also interesting to see vertical profiles of horizontally averaged statistics over the inner portion of the inner grid, this to see if we observe typical canopy layer, mixing layer and boundary layer profiles.*

We agree with the reviewer's comment. The transition of the flow from a mesoscale model, where the turbulent transport is parametrized, to an LES model, where the relevant turbulence scales are resolved, requires a sufficiently large fetch length. The required fetch length depends strongly on the mean wind speed, boundary-layer depth and stability, and this is discussed in detail in a companion paper by Kadasch et al. (2020) in this special issue. The presence of obstacles and the orography also play a role as shown by Lee et al. (2018).

The same considerations are also valid for the LES-LES nesting where the turbulent flow undergoes a transition from the coarse to the fine grid. A detailed discussion about this can be found in a companion paper by Hellsten et al. (2020).

In order to strengthen the manuscript in this regard, we have decided to present a proper analysis of this. For this we re-run parts of the winter- and the summer-simulation at specific points in time in order to present results from different atmospheric conditions, i.e. for a neutrally-stratified to weakly stable situation represented by the winter case and convective conditions represented by the summer case. We have added a new subsection 4.1.3 "Spatial development of the urban boundary layer" where the adjustment of the flow in the parent domain is discussed in terms of the TKE at different heights. As expected, this analysis shows that the TKE has not been fully adjusted when the flow enters the child domain, especially at higher levels. We explicitly note that a larger outer coase-grid model domain would be desirable, and this would also account for mixing processes at higher levels. However, this analysis also shows that turbulence has already been developed and has similar strength compared to locations significantly further downstream, so that we do expect that the error made by this insufficient adjustment does not significantly affect simulation results on the street canyon scale.

With respect to the transition from the coarse-grid to the fine-grid domain, we now present frequency spectra at 50 m (mainly above the building roofs) evaluated at locations with different distances to the inflow boundary. Especially in the winter case the spectra indicate that a fetch length of a few hundreds of meters is required to avoid adjustment effects. In the summer case under convective conditions the flow transition is even faster and less than 50 m of fetch length are required, which is in agreement with findings presented in Hellsten et al. (2020) for buoyancy-driven boundary layers.

*2) A general comment is that the paper is too long and some of the comparison do not really add new information. On the contrary some discrepancies in the model results with respect to the measurements are not discussed but just mentioned. Below some more detailed comments.*

We are aware of this fact which partly follows from the quantity of the material but the problem is also caused by deficiencies in our presentation. This point about quantity is also discussed in RC 1 p. 2) and in our answer to that, To address this comment, we went through all the text and we tried to restructure and condense it and to remove all unnecessary parts and improve and strengthen description and discussion of the findings. Chapter 4 has undergone the most major restructuring, but also other parts have also been substantially improved. Detailed descriptions of the various restructurings are given in our answers to individual comments of both reviewers.

*2.1) Figure 5, 6, and 7 are not necessary. Move it to the supplement and simply give a short description. Instead it is very important in my opinion to include WRF model output in figure 9, 10 and 11. In this way the reader can appreciate if there is any significant difference between PALM and WRF. Moreover, WRF simulation results should be included both for the location of the PALM profile and the location of the measured profile. In this way the reader can also evaluate how meaningful is to include the measured vertical profiles (that are for a location outside PALM coarse domain) in the comparison.*

Thank you for this suggestion (compare also RC1 p.4) ). To address this issue, we supplemented the graphs with the WRF profiles for both locations, the sounding location as well as the location of the modelling domain. Moreover, we unified the different sets of comparison graphs - sounding vs. WRF profiles (Fig. 6, 7) and sounding vs. PALM profiles (Fig. 9, 10, S13, S14, S15) -  into one series of graphs (FIg. 5, 6, and S15, S16, S17 in the revised manuscript) which provides complex information about vertical profiles; we hope they give the reader valuable and clearly arranged information. We updated the text description accordingly.

*2.2) The peaks in the vertical walls temperature, figure 13 (winter), figure 14(winter) should be discussed and it should also be discussed why the peak does not appear in figure 15. Are these peaks related to the radiative transfer model and its interaction with the wall at a particular sun angle under clear sky conditions? This may be an explanation of these peaks appearing at the same time of the day, and also the reason that they do not appear in summer. I noticed that in figure 18 a similar peak appears also in the observation (location 06_4_V). Did the specific wall geometry in relation to sun angle generate the peak? However, it is my opinion that these spikes in temperature and heat fluxes are very local surface effects with little or no influence on the overall heat exchange of the walls and even less on the atmospheric flow (see also point 3 below).*

We added an analysis and discussion of the peaks in a new section 5.1.5 "Rapid changes of surface temperature". The analysis examines the situation at location 11-1_V where these peaks are very strong and are not properly reflected in the observations but the discussion is easily applicable to any other location. The peaks are in fact the real reactions of the surface skin layer temperature to the strong "binary" changes of the direct radiation forcing due to changing of the shading of the sun rays during the course of the day. There are two reasons why these changes are more visible during the winter episode. Firstly, the radiation forcing by the direct incoming radiation to the vertical walls can be as strong or stronger in the winter as during the summer in the right conditions, and the remainder of the incoming radiative energy  is lower which makes the  corresponding changes of the surface temperature more "visible". Next, the lower sun elevation angles causes the effects of the shading by other buildings or terrain to

occur more frequently. Moreover, this effect is more pronounced on walls than on the ground as the thermal conductivity of the walls is usually lower than the conductivity of ground surfaces which means the ground heat flux is able to dampen the changes of the skin temperature according to the energy balance equation. The magnitude of the changes of the surface temperature at location 11-1_V are similar in the model as in the observations according to Fig. 20 while the precise shape slightly differs due to geometrical imperfections in the digital elevation model used, discretization effects, etc. More details are given in the discussion in the manuscript. A brief mention of this effect and a reference to this analysis were added to the general description of the surface temperature results (section 5.1.1).

*2.3) Fig 11. The discrepancy in wind direction should be discussed in more details,
and the WRF results should also be included both for wind speed and direction. I
think that adding WRF wind direction will help in interpreting the results. Also, please
make the shorter arrows longer (or simply make all the arrows of the same length
since the magnitude is already reported in the lower panel). Please, do the same in
the supplement.*

As in the original version we considered the wind observation on FSv building as only supporting information and not part of the main focus of the study, we did not discuss it in detail (the focus of the study is the street canyon level comparison). To attempt to address this issue, we have dived deeper into the problem. At first, we tried to decide whether this effect (which appears only during winter episodes) is a technical problem or some microscale dynamic effect. We went through the entire process of measurement and data collection with the technicians, and we checked all our postprocessing tools. According to the technicians, the placement of the sensor and the settings of the equipment were identical for summer and winter episodes. However, we found no photo documentation of the sensor for the winter episode as exists in the summer photos. We thus were not able to conclusively check and prove the actual real orientation of the sensor in the winter study. We also tried to compare wind direction data from nearby synoptic stations as well as from WRF and PALM 3D fields. Despite all this effort, we came to no solid conclusions besides some speculations. As we cannot exclude the possibility of a technical issue (the wrong orientation of the sensor) during the winter episode, which would affect the wind direction, we decided to exclude the wind direction from the presented FSv wind observation and we updated the description. The potential missorientation of the sensor would not influence the observed wind speed so we consider this approach as the correct one. We consider this issue a very interesting and important problem and in new projects we are starting, we intend to repeat observations of similar type using an improved design.
To improve the presentation of the wind on FSv and its discussion, we first reworked the wind graphs themselves; the wind magnitude graphs are taller and the wind values from WRF and nearby synoptic station Ruzyne were added. We added statistics for this observation and we also improved the description of this part of the study to express clearly that we do not consider these observations as the observations of the free flow but as the observations of a flow influenced by surface effects, but at a higher level than the street canyon observations. To

emphasize this, we moved this part from section 4.2 to section 5.3.3 "Wind speed on the roof" in the revised version.

*2.4) figure 13, 14, 15, 16, 17, 22, 23, 24. I think that the WRF surface temperature should be included in the comparison for the "horizontal" surface. This should be just a single extra line for each location and allows to appreciate the difference/similarity with PALM, and this difference/similarity should also be discussed.*

Thank you for this suggestion. We added WRF values to all horizontal surface graphs and we updated the corresponding text and discussion.

*2.5) Why not using the heat flux measurement locations (Sinkule and Zelena) also in the discussion in section 4.3 (i.e. as typical urban locations). This may allow discarding some of the figures.*

This is a natural idea. We do not use them as they do not cover the same time period and/or locations as the surface temperature observations: the heat flux observations at Sinkule started later, at the beginning of the second summer episode, due to delayed delivery of the instrument; observations in Zelena followed two weeks after that, and IR camera measurements were not taken at this location. Moreover, the HF observations require there to be no direct sunlight on the wall while the "interesting" cases of the surface temperature examination occur in interactions of the wall with the solar radiation of all kinds, including the direct radiation. For that reason, we decided to examine and present the surface temperature observations and the heat flux observations separately.
Nevertheless, we restructured all the chapter 4 and the "typical urban locations" are not presented anymore. Instead, location 11-1_V is used as the only complete example, and it is also utilized later in the discussion of the peaks.

*3) The authors discuss the error induced by discretization (section 4.3.5) and also its possible solution (section 5.2. lines 873-884). I think that these discussions are partially misleading and must be modified.*
*I think that step-like discretization errors at this resolution can create local discrepancies (local in space and/or in time, see point 2.2 above) of limited overall significance; they seem significant only for the very local surface heat flux and temperature. Moreover, by increasing the resolution (e.g. 1m or less) the discretization errors become of the same order of the neglected SGS wall features. Therefore, there is no point in avoiding these errors if the building is not represented in all its many details. Furthermore, it seems to me that the error, being mainly related to the radiative transfer model, can be fixed irrespective of the representation of the building in the fluid dynamic solver (i.e. I think that there is no need to have a more sophisticated immersed boundary method in the fluid solver to fix the behavior of the radiative transfer model since for the latter it is enough to keep track of the local actual surface angle). Finally, I think that the influence of the current discretization method on the flow is*

*quite limited and local at the considered resolution, and become quickly negligible (and likely comparable with the neglected SGS wall elements) if further increasing the grid resolution.*

We would like to thank the reviewer for raising this question which is indeed a tricky problem. We agree with the reviewer that the effect of step-like surfaces on the flow is mainly via the surface heat flux and thus in the radiation and the surface energy balance, while the direct effect on the flow is probably only of local nature;  we added this to the text. However, to our knowledge there is no study available which directly compares the influence of step-like vs. sloped-surface representation on the flow. Due to step-like representation of the surface the surface area is artificially increased, meaning more surface friction, which may in turn also affect the mean flow above the urban boundary layer. But frankly speaking, we do not know for sure.

Also, we agree with the reviewer that the effects of the discretization on the radiation are most strongly manifested locally and their effects on larger scales partially mutually compensate. This fact is in our opinion clearly stated in the text, mainly in the last paragraph of the original section 4.3.5 (5.1.7 in the revised manuscript). As this study focuses mainly on the very local "point" comparison of the model and observations (one of its purposes was validation of the newly developed and improved modules RTM, USM, LSM), it is necessary to keep these effects in mind all the time, otherwise the conclusions could be wrong for point-to-point comparisons. The discussion of these effects inside this paper thus seems to be reasonable and it can be helpful for everybody who will attempt to use modelling results for assessment of local quantities.

Moreover, we are convinced that the global effects of the discretization cannot be neglected in some cases as the step-wise representation of the surface causes e.g. significant change of the surface area, roughness length, normal angles, mutual visibility, etc. To roughly estimate the level of the global effects of the discretization of a wall, we performed an idealized experiment which utilizes PALM's ability for grid rotation. We ran two simulations of the same idealized urban area with one west-east oriented street canyon. Both simulations modelled the same real geometry and properties of the street canyon, they had the same configurations with the exception that the first simulation was configured with no grid rotation while the other simulation had the grid rotated by 45°. We chose the configuration of the grid in a way that the effective width of the discretized street canyon stayed unchanged. The results for the south-facing wall were averaged over the central one third of the street canyon to avoid potential near boundary effects. The results show that the difference in the wall surface temperature can reach above 3 °C,  over 100 $Wm^{-2}$ in case of shortwave irradiance and about 80 $Wm^{-2}$ in case of net radiation. These differences in the radiation fluxes and surface temperature consequently alter ground heat flux and turbulent sensible heat flux which can afterwards (in suitable conditions) alter the pattern of the flow inside the street canyon as we experienced in many of our simulations (one example is mentioned in our earlier GMD paper (https://doi.org/10.5194/gmd-10-3635-2017, chapter 3.4.1, fig. 16). We are convinced that this issue needs further research and better quantification.

To address this comment, we restructured section 4.3.5 (5.1.7 in the revised manuscript) as well as the corresponding part of section 6.2. We decreased the number of examples and we condensed the text. We tried to strengthen the formulations and reasoning to avoid any possible misinterpretations. We also added the estimate of the averaged effects of the discretization on

the wall radiation balance and surface temperature to the end of the section 4.3.5 (5.1.7 currently) together with supporting material in supplements. Please, reconsider this revised version.

*4) line 845-847. The authors correctly point out that the PALM results are often very similar to the WRF results.*
*This is true for the close to surface air temperature, where WRF seems to have a better behavior in winter and a similar performance in summer compared to PALM.*
*This should be discussed, and some statistical indices should be included comparing WRF and PALM performances.*
*On the contrary the wind speed is generally significantly different, and this is obviously because the buildings are simulated explicitly. The explicit building representation also allows a better agreement between PALM and the observation for wind speed with respect to WRF (within the urban canopy). This should also be discussed, and some statistical indices should be included comparing WRF and PALM performances.*
*In my opinion this behavior demonstrate that the role of a very fine scale heat surface description is minor compared to having the obstacle represented and can perhaps be not so important. For the correct modeling of the street level urban atmospheric flow It seems sufficient to fairly reproduce the overall integrated heat exchange of the building wall. Very local temperature and heat flux effects (those investigate here with point wise sensors) seems quite negligible for the simulation of the mean flow and mean pollutant dispersion at street level.*

The behaviour of the WRF model simulations can be expected to be in accordance to e.g. Halenka et al (2019, https://www.inderscience.com/info/inarticle.php?artid=101840), in which it is shown that even mesoscale model with no urban parameterization, only what is referred to as "bulk representation" (only changing properties like albedo, emissivity, roughness, etc. in the surface parameterization), can simulate average temperatures with sufficient accuracy. For wind or PBL height it shows quite a poor performance. We added this statement and reference to the text.
However, in our study the focus is on the very local street canyon level features, not the mean flow or average temperatures. The purpose of these modeling studies is also to create a modeling system capable of providing very local information that can be used in urban studies such as UHI and AQ mitigation scenarios. Also, as we show in the companion paper (Belda et al. (2020), https://doi.org/10.5194/gmd-2020-126), testing sensitivity to parameter settings in the PALM model, the very fine scale heat surface description is essential as the local features can have unexpected results (e.g. increased albedo can lead to increase in surface temperature due to reflections from opposite surfaces). The example of the influence of the distribution of the surface heat flow in the street canyon is shown also in our earlier paper (Resler et.al. 2017, https://doi.org/10.5194/gmd-10-3635-2017) in Fig. 16 and 17.

*5) The author should improve the reproducibility of their simulations. This is a GMD*

*manuscript and I think that all the necessary input files for performing the simulations
should be provided for download, not just a generic reference to the open source model
code. If I am not wrong for PALM these should be, the namelist file e.g. "prague_p3d",
form both grid levels, the dynamic input file, e.g. "prague_dynamic", the static input
file for both grid levels, e.g. "prague_static". The author should also include the file for
the shortwave and long wave radiation that are obtained from WRF and used in the
simulation, and briefly explain how they are generated.*

The complete configurations and input data of all PALM simulations were published in the ASEP
repository (http://hdl.handle.net/11104/0315416, for information about the repository see
https://asep-portal.lib.cas.cz/basic-information); this reference was added to the text and to the
list of references in the manuscript.

*5.1) The authors write that the details for obtaining the WRF (interpolated) wind field
on the PALM grid goes beyond the scope of this manuscript (lines 3675-376). I think
this detail should be well explained, also because they did not refer to any previously
published full validation of this WRF-PALM coupling.*

The 3-D fields from WRF outputs (T, Q, U/V/W) were horizontally and vertically interpolated (in
that order) to the PALM model grid. Because the PALM model used a higher-resolution terrain
that would differ from the coarse terrain in WRF by as much as tens of meters, the vertical
interpolation had to include stretching of the atmospheric columns.

At the bottom, the atmospheric columns were shifted to match the PALM terrain, therefore there
were no missing data below the original terrain and the surface effects from WRF were
preserved. However, at higher altitudes, the atmospheric columns could not be shifted by the
same amount, as that would introduce unrealistic horizontal gradients mimicking the terrain shift
below. In order to avoid this, the atmospheric columns were stretched heterogeneously. The
WRF model uses either sigma or hybrid vertical coordinates, our simulations use the hybrid
option where the lowest level is terrain-following and the highest level is isobaric. For each
column, the geopotential height of each level in the WRF data was recalculated using the same
formula and parameters used in WRF for calculating the heights of the hybrid levels, however
with the surface pressure altered to match the PALM terrain. The recalculated level heights
were then used for linear vertical interpolation into the PALM Cartesian vertical coordinate
system.

The interpolated 3-D fields were used as initial conditions for the first timestep and their top and
lateral boundaries were used as boundary conditions for all timesteps. For the velocity fields,
the total volumetric flux disbalance was calculated for each timestep as a sum of the volumetric
inflow minus outflow for all boundaries. This residual volumetric flux was then divided by the
total area of the five boundaries and subtracted from the respective inwards-directed velocity
component for each boundary in order to make the inflow and outflow perfectly balanced, as is
required by the incompressible equations used in PALM.

The Python code used for processing the WRF and CAMx data into the PALM dynamic driver file has been included into the official PALM distribution and published in the PALM SVN repository in the directory UTIL/WRF_interface since revision 4766.

To address the reviewer comment, we added a brief description of the transformation process and the reference to the transformation source code in the text of the chapter 3.3. of the manuscript. We also added a more detailed description of the transformation of WRF and CAMx data to PALM dynamic driver to Sect. S5 of the supplements with reference in the manuscript.

*5.2) The coupling between WRF and PALM includes the specification of the horizontal pressure gradients in the two orthogonal directions at any level. The typical profiles applied for the simulations in summer and winter should be included in the supplement to the manuscript.*

In the WRF-PALM coupling we do not consider pressure gradients explicitly. In the mesoscale nesting approach boundary values of the velocity components, potential temperature and mixing ratio are provided at discrete points in time, no horizontal pressure gradients are prescribed in this approach. This is because the horizontal pressure gradients are already implicitly considered in the solution of the Poisson equation for the perturbation pressure, which in turn includes the lateral boundary values for u and v. In other words, the pressure solver (to maintain incompressibility of the flow) creates the large-scale pressure gradient itself.

After several internal discussions within the PALM development group, we decided to completely omit large-scale pressure gradients in the mesoscale nesting approach, even though tests showed that it does not make a significant difference if it is considered or not. At this point we would like to refer to Kadasch et al. (2020, GMD discussion paper, https://doi.org/10.5194/gmd-2020-285) where the mesoscale nesting approach is presented in detail, including a discussion about the large-scale pressure forcing at the end of section 2.2.4.

*5.3) It seems that buildings are not resolved explicitly in the outer PALM grid. What local parametrizations have been used to describe the urban canopy?*

The buildings in the parent domain are explicitly resolved and they follow quite well the general structure of the buildings in the domain. The input data for the parent domain were prepared from the same DEM source (the Prague 3D model based on photogrammetric aerial mapping) as the child domain. The resolution of the parent domain is 10 m and thus all buildings with height higher than 5 m are explicitly resolved. This represents the absolute majority of the buildings in the domain. We thus did not utilize any additional parameterization except the standard roughness length utilized in LSM and USM via roughness length of individual surfaces. To address this comment, we provide two additional figures of the buildings in the parent domain (3D view of the domain and the grid footprint with the resolved buildings) in the supplements (S11 and S12) and a brief remark about explicit resolution of the buildings in the parent domain in Sect. 3.1.

**Other comments**
*Page 3, I suggest removing lines 76-83 they seem redundant.*

The text was not formulated well, but it gives important information about the focus of the study which influences its design. It was reformulated and condensed in the revised version.

*Page 4, line 116. What does it mean (+-20m), about 20m?*

Yes. This formulation should mean "about". The sentence was reformulated.

*Figure 12. There is no color bar showing relations between temperature and colors.*

The figure already contained the color bar in the right bottom corner but it was too small and poorly visible. We changed the format of this color bar (size, font size, shadowing of the fonts) to make it more readable.

*Figure 32, 33, what does it mean "locations_V" or "location_Z"? I could not find the definition for the nomenclature "_V" "_Z".*

We considered it only as part of the location indicator and we did not assign it to any nomenclature in the case of the locations of vehicle meteorological stations. In fact, this naming was assigned by the crew of the observation vehicles and it originates from Východ/Západ (East/West in Czech language). In order to make labels of the graphs more intuitive, we changed it to the full description (e.g. "Bubeneč house East").

*Figure 34. The light green band is almost invisible.*

The spread of the 10-minutes values is small for observations of the concentrations in many parts of the campaign and the area of this spread is almost hidden by the value line (in contrast with e.g. graphs of the wind). This also gives information in our opinion. To improve the visibility of this spread, we changed the colors of the graphs 32, 33, and 34 (26, 27, and 29 in revised manuscript). Moreover, we restructured these graphs in the way that graphs are wider to make lines better visible and we reduced graphs in the manuscript to episodes summer e1, summer e2, and winter e3; the complete graphs for all episodes and variables were moved to supplements to section S5 "Street canyon quantities".

*Figure S13, S14, S15, add the vertical axis label*

Corrected

*Figure S16, caption. What is Fig supp13? should it be Figure S10?*

Yes, thank you for noticing it. Fixed.

---

## Author Response (AR2)

***Topical Editor Decision: Publish subject to minor revisions***
*(review by editor) (26 Apr 2021) by Ignacio Pisso*

***Comments to the Author:***
*Please address the two minor comments of the reviewer:*

Dear topical editor,

We are sending you our answers to the reviewer comments below in the form of individual points. We have also uploaded the revised manuscript updated according to the reviewers comments and the track-changes file. The changes are located on lines: 78, 129-130, 140-141, 597, 857-860, 1006-1008, 1263, and in Figure 9 and its caption. We would like to thank you for all your time and effort dedicated to this review process.

With kind regards,

Jaroslav Resler
(on behalf of all co-authors)

*I find the revised manuscript improved, easier to read and follow. The authors have considered my comments and I have now only two minor comments below for the authors and some corrections to the text.*

*1) The authors answered to my comment n.3 writing that "…we are convinced that the global effects of the discretization cannot be neglected in some cases as the step-wise representation of the surface causes e.g. significant change of the surface area, roughness length, normal angles, mutual visibility, etc.."*
*Their rotation test shows indeed that there is an effect on heat fluxes in the most extreme case of a rotation of 45 degree for a flat surface. However, in my opinion the rotation test included by authors show a very partial view and I remark what I expressed in my original comment:*
   *a) The error in the simulation is mainly induced by the radiative model not the LES flow model, i.e. not by the mask method in LES but by the stepwise representation in the radiative transfer model. The correction of geometry for the radiative transfer model can be done irrespective of the representation of the geometry in the LES. Therefore, the change in the representation by using IMB in the LES flow solver mentioned in 6.2, line 1006, is somewhat unrelated to the present issue.*
   *b) The effects discussed by the authors will become quickly negligible when the resolution of the model approaches the uncertainty in surface representation, i.e. what in the model appears as a flat surface is in reality characterized by protruding objects (e.g. balconies, windows frames etc.., see also e.g. figure 22 top-left) that have sizes of the order of a meter or so. The uncertainty in the representation of the surfaces is likely comparable to the error induced by the discretization as soon as the resolution in the model is of the order of about 1 meter.*
*I mention that the new detailed discussion in sect 5.1.5 explains that the unrealistic spikes in the modelled temperatures (e.g. Fig. 11) can be mainly attributed to the DEM used for the buildings not being sufficiently accurate, this seems to confirm the view that I expressed above in point (b).*

We consider our view on these matters not too distinct from the reviewer's one and we would not like to argue about this issue. We would like to mention the beginning of our answer to reviewer comment 3: „*We agree with the reviewer that the effect of step-like surfaces on the flow is mainly via the surface heat flux and thus in the radiation and the surface energy balance, while the direct effect on the flow is probably only of local nature; we added this to the text.*" On the other hand, the explicit quantification of the direct effects of the surface representation by the orthogonal grid structure on the LES flow neither follows from our work nor is it precisely done in the literature to our knowledge. We consider it a topic of future research (as we mention in our answer). For this reason we would like to avoid any speculations in this direction in the text of the current paper.

Point a): The PALM model is an integrated model where the individual processes share most of the data structures and we consider that as one of the strengths of this model. The data representation of the slanted surfaces thus also will be shared across all affected processes (RTM, USM, LSM, and LES). The reviewer is right that the formulation on line 1006 is unfortunate and can suggest that the central point of this enhancement is the representation of the slanted surfaces in the LES module. To avoid this possible misinterpretation, we reformulated this sentence and we replaced the citation of the particular IBM approach for LES by the general term "arbitrarily oriented surfaces" which is applicable for all affected processes.

Point b): If the grid spacing does not allow the resolution of small-scale surface objects, also reflections and mutual visibility on these scales cannot be represented either. In this regard, we agree with the reviewer that the error made by the step-like approach can be of the same order as the error made by non-representation of subgrid obstacles. To some degree we even expect that errors made by these effects may partly compensate for each other, though we emphasize that this is just speculation and we need to quantify this in the future.

However, when we look on relatively smooth facades, natural terrain, or slanted roofs without any significant objects, the error of non-representing subgrid obstacles is negligible, whereas the error made by the step-like surface representation is a pure misrepresentation of the model. To quantify all these effects, also in conjunction with the misrepresentation of small-scale surface objects, further research is required in the future.

As our study was not designed to investigate these issues and thus our results do not sufficiently support this issue, we would like to avoid direct statements of this type in the manuscript. To address this comment, we reformulated the corresponding paragraph of section 6.2 which is referred to at the end of section 5.1.7. As was mentioned in the answer to point a), we reformulated this part in a way which expresses better that the main goal for this model enhancement is to tackle discretization issues in the energy related processes which is consistent with the findings described in section 5.1.7.

The last paragraph of this reviewer comment touches the problem of the uncertainty of the input data which is another interesting issue. In our opinion, the main result presented in section 5.1.5 is the finding that these surface temperature spikes are realistic in their magnitude and general shape. The spatial and temporal discrepancies can be probably attributed to the impreciseness of the DEM used and caused mainly by the existence of objects on the roofs, which are not included in the DEM but which can cause shading (e.g. air conditions, balusters, and other "soft" objects). Some part of the discrepancies can be

also caused by the discretization process. The DEM model itself is based on the official Prague Open Data (see sect. 3.2 and https://opendata.praha.eu/en/), it has a resolution of about a half meter. The data have been improving so DEM imprecision should not be the main issue in the future and the main problem seems to be in the above mentioned "soft" objects. But this particular issue is a part of a more general problem of the uncertainty of most of the model input data and parameters. The influence of these uncertainties on the model results is systematically studied in another manuscript of this special issue - Belda at al.: Sensitivity analysis of the PALM model system 6.0 in the urban environment.

All these reviewer comments touch partial issues which are very interesting and worth studying individually deeper in the future. We consider other more detailed or specialised discussion of any of these particular problems outside the scope of this work and of this manuscript. Nevertheless, we thank the reviewer for this discussion and all his/her comments, which not only helped to improve the manuscript itself but also influenced the aims of our future research.

*2) Figure 9, please add the Kolmogorov inertial range scaling in the figure for reference.*

We added Kolmogorov's -5/3 law as a dashed line for reference. Within the inertial range the LES spectra agree well with Kolmogorov scaling, whereas we can observe a sharp drop-off at higher frequencies that correspond to smaller spatial scales. This is a well-known behaviour that can be mainly attributed to numerical dissipation accompanied by the advection scheme (acting up to scales smaller 10 times the grid spacing) as well as to the subgrid-scale scheme.

***Text corrections.***
*Line 78, remove "complex" does not seem appropriate.*
Done.

*Line 129, 140. Rewrite the sentences removing "Important", it seems unnecessary.*
Done.

*Line 597,"5.1.5" not "5.1.3".*
Fixed

*Line 860, I do not think that "expedient" is the correct word.*
The word "expedient" was replaced by the word "appropriate" and all this sentence was reformulated.

---

## Author Response (AR3)

Dear Sir or Madam,

I am submitting the latex text and the figures of our manuscript gmd-2020-175: Validation of the PALM model system 6.0 in a real urban environment; case study on Prague-Dejvice, Czech Republic.

The changes in the document from the last minor revision are:

1. All URLs of the web links in the document have been checked and updated in case the structure of the web pages changed in the meantime.

2. The references to the preprints of the accompanied papers of this GMD special issue were checked and if the final paper was published in the meantime, the reference has been changed to this final paper.

3. The affiliation "Institute of Computer Science, Czech Academy of Sciences" has been replaced by formulation "Institute of Computer Science of the Czech Academy of Sciences" which is required by the current rules of the Czech Academy of Sciences.

4. Due to longer than expected review process, an additional financial support grant has been added:
Financial support was also provided by the Norway Grants and Technology Agency of the Czech Republic project TO01000219: Turbulent-resolving urban modelling of air quality and thermal comfort.
Please, add this financial support also to the manuscript database. Thank you.
We provide financial support information in the first paragraph of the Acknowledgement section.

With regards,

Jaroslav Resler